# CAN REASONING LANGUAGE MODELS THINK MORE CREATIVELY? A STUDY OF REASONING ABILITY AND OVERCONFIDENCE

## ABSTRACT

Recent advances in Large Language Models (LLMs) have been largely attributed to improvements in reasoning abilities. LLMs trained with Supervised Fine-Tuning (SFT) and Reinforcemenet Learning (RL), such as Policy Optimization (PO), demonstrate significantly superior performance compared to base models. However, recent studies raise questions about the ability of these RL models to achieve creative thinking beyond that of the base model. In this study, we compare the creative problem-solving abilities in mathematics of two types of models: RL models and base models prior to SFT or RL, further trained on simple mathematical corpora. Our comparison spans two representative open-source LLM families, DeepSeek and Qwen. The results indicate that RL models are less effective in generating creative solutions. We attribute the RL models' limited ability to generate creative responses to Overconfidence (OC)—the tendency of models to exhibit excessive confidence in their own outputs. For example, within the DeepSeek family, RL models exhibit 15% higher OC compared to the base model, and within the Qwen family, the gap rises to 80%. Notwithstanding their heightened OC, they fail to generate creative responses as intended. Based on prior studies reporting overly aggressive probability shifts for certain tokens during SFT and PO, we hypothesize that the high OC may arise from such adjustments. To examine this hypothesis, we introduce the notion of a High Entropy Segment (HES), defined as a region in which entropy varies sharply. Within these segments, RL models tend to exhibit greater heterogeneity compared to other models. Lastly, we measure the proportion of time steps where the model does not generate the most probable token, and observe that RL models show a substantially lower rate than base models. This is largely because their distributions contain a substantially greater share of tokens whose probabilities exceed 80% at each step. Our findings will be of great help in understanding and improving RL models.

## 1 INTRODUCTION

Since the significant success of OpenAI's ChatGPT-o1 (Zhong et al., 2024; OpenAI, 2024; Wang, 2025), numerous studies have focused on enhancing the reasoning abilities of Large Language Models (LLMs). Among these, the most representative and successful approaches are instruct tuning with Supervised Fine-Tuning (SFT) and Reinforcement Learning (RL) using Policy Optimization (PO) (Schulman et al., 2017; Shao et al., 2024; Lambert et al., 2024; Yu et al., 2025). Models trained with both SFT and RL have demonstrated remarkable reasoning abilities, achieving performance close to or even surpassing human level in solving mathematical problems. More recently, studies analyzing the characteristics of these high-performing RL models have increased. However, some recent studies have reported that RL models often show an excessively high degree of confidence in their generations. This phenomenon, largely attributed to the SFT and RL processes, results in exceedingly low entropy in the outputs (Wang et al., 2025; Chu et al., 2025; Yue et al., 2025). Some studies refer to this phenomenon as the Over-Confidence (OC), and there have been efforts to address this issue by proposing new PO methods (Wen et al., 2024; Leng et al., 2024).

It is somewhat ironic that, despite the prevalence of OC in RL models, relatively little attention has been given to one of its most intuitive consequences: a lack of diversity and creativity in RL model

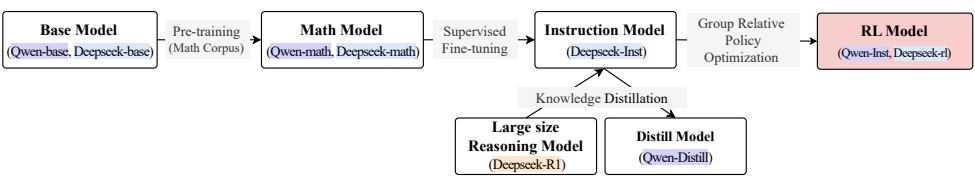

Figure 1: Training pipeline of the model families. Base models are adapted via math-specific pre-training, supervised fine-tuning, knowledge distillation, and reinforcement learning with group relative policy optimization.

generations. For example, while extensive research has explored creative writing with LLMs (Kim, 2006; Chakrabarty et al., 2024; Zhao et al., 2025), only few studies have investigated whether RL models can approach tasks such as mathematics or coding in diverse and creative ways. Although these models have demonstrated high accuracy in solving mathematical problems, it would be a significant limitation if they consistently produced similar solutions and failed to generate approaches that are novel and innovative.

In this study, we evaluate the creative mathematical problem-solving abilities of both base models and RL models, and analyze the correlation between creative reasoning ability and the OC ratio. Specifically, we compare models before and after training by SFT and PO, finding that models before training exhibit significantly stronger abilities to generate creative solutions. Furthermore, RL models overwhelmingly generate OC responses compared to base models, and such OC responses do not contribute positively to creative problem solving. In order to investigate the reasons why RL models exhibit lower creative abilities compared to base models (only further trained with mathematical corpora), we measured the entropy of outputs from both models across multiple model families that share the same base model (Deepseek, Qwen) (Shao et al., 2024; Qwen et al., 2025; Huang et al., 2025b). The results show that while base models display consistent entropy patterns across all model families, RL models exhibit unique entropy patterns to their family. To further investigate, we conducted two complementary experiments.

First, we defined segments exhibiting sudden entropy shifts as High Entropy Segments (HES). Interestingly, RL models—despite generally producing low-entropy outputs—exhibited a more HES than base models. Furthermore, the overlap of HES between RL models and other models was notably weaker. In contrast, base models demonstrated highly consistent HES across all families. Second, we measured the difference between the highest-probability token at each time step and the token actually generated. RL models showed the lowest rate of divergence, indicating a strong tendency to follow the most likely token. In addition, the proportion of time steps where the highest-probability token had a probability exceeding 80% was significantly higher in RL models than in other models. This suggests that the SFT and PO processes may assign disproportionately high weights to certain tokens and paths, resulting in OC generation. Based on these findings, our contributions in this study are as follows:

- We show that RL models exhibit lower creative mathematical problem-solving abilities than other models.
- We define *Overconfidence* phenomenon and prove that RL models generate significantly higher levels of overconfident responses than other models.
- We define the *High Entropy Segments* and reveal heterogeneity between RL models and other models.

## 2 RELATED WORK

### 2.1 HALLUCINATIONS FROM OVERCONFIDENT GENERATIONS

In various tasks, when the uncertainty of LLM generation is high, the generation is often hallucination, and a certain degree of correlation is recognized between the uncertainty of model generation and the accuracy of task (Ji et al., 2023b; Chen et al., 2024; Sriramanan et al., 2024; Huang et al., 2025a). However, detecting or mitigating hallucinations becomes particularly challenging when

generation uncertainty is low (Ji et al., 2023a). Although methods such as Expected Calibration Error (ECE) (Guo et al., 2017; Wang et al., 2022; Wang, 2023) have been proposed to address the issue of models being overconfident in their outputs despite low actual accuracy, these approaches are essentially black-box methods that query the model itself about the confidence of its generation. As such, they do not sufficiently investigate the causes of overconfident and wrong generations.

## 2.2 OVERCONFIDENCE IN RL MODELS

It has been revealed that models that have undergone SFT and RL (recently PO-based methods have become a trend) tend to generate responses with very low entropy. This overconfident (low-entropy) generation is the result of faithfully executing learned patterns and often shows high accuracy. However, recent studies suggest that base models can achieve performance comparable to RL models when given sufficient inference opportunity. Moreover, some works have identified that certain tokens—referred to as fork tokens, which play critical roles in branching the generation—appear infrequently but exhibit high entropy (Wang et al., 2025; Yue et al., 2025) and that performance can be improved by manipulating these tokens. This raises the question of how the presence and behavior of such fork tokens affect generation quality in OC.

## 2.3 CREATIVITY IN MATHEMATICAL PROBLEM SOLVING

The reason why research on the creativity of RL models is rarely conducted is because evaluating 'creativity' relies entirely on human evaluations, and because the task requires finding the 'correct answer', constraining diverse thought processes becomes beneficial for assessing performance. Recent studies have attempted to evaluate typical and novel solutions separately by utilizing multiple solutions solved by humans for recognized mathematical problems (Yue et al., 2024; Ye et al., 2025). By providing reference solutions and inducing the model to generate distinct responses, the creative generation ability of the model could be evaluated without human intervention.

# 3 EXPERIMENTAL SETUP

## 3.1 EXPERIMENTAL MODEL

**Deepseek (DS) Family**   The Deepseek family is based on the Deepseek-Coder-Base-v1.5-7B (Daya Guo, 2024) and comprises the following four models:

- **Deepseek-Base** (Daya Guo, 2024): Base model (Deepseek-Coder-Base-v1.5-7B)
- **Deepseek-Math** (Shao et al., 2024): Models further pre-trained on mathematical corpora (Deepseek-math-7b-base)
- **Deepseek-Inst** (Shao et al., 2024): Models instruct-tuned with SFT on Deepseek-Math (Deepseek-math-7b-instruct)
- **Deepseek-RL** (Guo et al., 2025): Models that apply Group Relative Policy Optimization (GRPO) to Deepseek-Inst (Deepseek-math-7b-rl)

**Qwen (QW) Family**   The Qwen family uses Qwen2.5-7B (Qwen et al., 2025) as its base model, and a model with both SFT and GRPO applied was released as the Qwen-Inst model, so there is no corresponding model for Deepseek-Inst (To avoid confusion, we note that Deepseek-RL and Qwen-Inst are equivalent models). So instead, we added the Qwen-distill model, which is a distillation of the Deepseek-R1 model into Qwen2.5-Math-7B, to the experimental group.

- **Qwen-Base** (Qwen et al., 2025): Base model (Qwen2.5-7B)
- **Qwen-Math** (Qwen et al., 2025): Models further pre-trained on mathematical corpora (Qwen2.5-Math-7B)
- **Qwen-Inst** (Qwen et al., 2025): Models fine-tuned with SFT and GRPO on Qwen-Math (Qwen2.5-Math-7B-Instruct)
- **Qwen-Distill** (Guo et al., 2025): Models trained by distilling the outputs of the teacher model, Deepseek-R1, into the student model, Qwen-Math (Deepseek-R1-Distill-Qwen-7B)

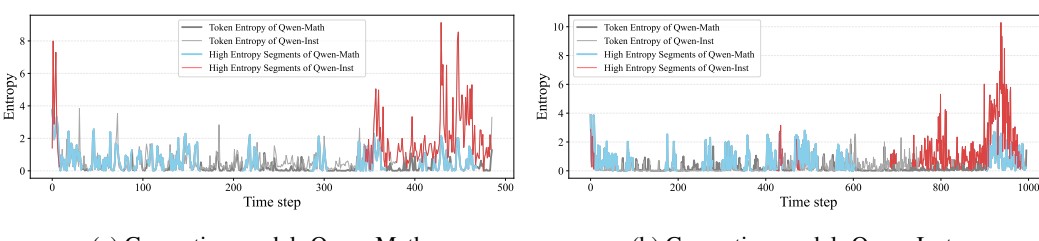

(a) Generation model: Qwen-Math        (b) Generation model: Qwen-Inst

Figure 2: Visualization of entropy and High Entropy Segments (HES) for model responses. Each subplot visualizes token-level entropy: dark gray for the Math model and light gray for the RL/Inst model. Blue and red regions highlight the HES detected for the Math and RL/Inst models.

To compare the differences between various models, from base models to RL models, we formed an experimental group based on the Deepseek and Qwen families. Given the presence of distinct base models, we refer to DeepSeek-Math and Qwen-Math as math models, and DeepSeek-RL and Qwen-Inst as RL models throughout this study.

## 3.2 DATASET CONSTRUCTION

We use CreativeMath (Ye et al., 2025), a publicly available math dataset, to analyze the creative generation abilities of both math and RL models. It comprises 400 problems, with 50 sampled from each of eight mathematics competitions: AMC8, AMC10, AMC12, A(J)HSME, AIME, USAJMO, USAMO, and IMO. We construct a new dataset of model-generated responses by prompting each model to generate creative solutions, which serve as the input to our creativity analysis. Details are provided in Appendix B.

## 3.3 EVALUATOR FOR CREATIVE ASSESSMENT

We employ two frontier-level LLMs—GPT-o4-mini and Gemini 1.5 Pro—to evaluate the solutions generated by experimental model. These two models have demonstrated outstanding performance on mathematical benchmarks and have already been recognized as LLM Evaluators replacing humans in several previous studies (Ye et al., 2025). Please, see Appendix B.2 for detail creativity assessment criteria and reliability. Because this is a very recent research finding and may be unfamiliar, the basis for this is presented in detail separately in the Appendix B.2.3 and B.2.4.

## 4 METHOD

### 4.1 MEASURING ENTROPY & OVERCONFIDENCE

To observe the entropy at each time step rather than over the entire response, we compute the entropy at time step $t$ using simple Token-level Entropy (TE), as shown in Equation 1.

$$\text{TE}_t = -\sum_{v=1}^{V} p_t^{(v)} \log p_t^{(v)}, \quad V = |\text{vocabulary}| \tag{1}$$

In a prior work, (Lee, 2023; Quevedo et al., 2024), it was both mathematically and empirically demonstrated that the Minimum Token Probability (MTP) is superior to the average token probability for detecting hallucinations. In addition, it was implied that it is stable to use a quantile lower probability token than the minimum probability token to prevent some token probabilities extremely low. Building on prior work, we propose $\text{Conf}_q$, which maps the $q$-th quantile of tokens with the lowest log probabilities from a model-generated sequence into the probability space. A generation is classified as OC if its corresponding $\text{Conf}_q$ exceeds a threshold $\theta$. In results section of main text, we use $q = 5\%$ and $\theta = 0.35$, and results for varying values of $q$ and $\theta$ are presented in the Appendix C.

$$\text{OC} = \begin{cases} \text{True,} & \text{if } \text{Conf}_q > \theta \\ \text{False,} & \text{otherwise} \end{cases}, \text{ where } \text{Conf}_q = \exp\left(\text{Quantile}_q(\{\log p_t\}_{t=1}^T)\right) \tag{2}$$

Table 1: Evaluation results of the CreativeMath dataset. Metrics include Correctness ratio (Cor), Creativity ratio (Cre), and the Creativity-to-Correctness ratio (Cre/Cor). The MATH column reports 4-shot CoT accuracy on the MATH benchmark.

| Source | Model | Cor (%) ↑ | Cre (%) ↑ | Cre/Cor (%) ↑ | MATH (%) ↑ |
|--------|-------|-----------|-----------|---------------|------------|
| Closed-Source | Gemini-1.5-Pro | 69.92 | 66.94 | 95.75 | 67.7 Reid et al. (2024) |
| | Claude-3-Opus | 59.84 | 44.63 | 74.59 | 61.0 Anthropic (2024) |
| | GPT-4o | 60.83 | 30.08 | 49.46 | 76.6 OpenAI (2024) |
| Open-Source | DeepSeek-math-7b-base | 34.68 | 19.92 | 57.44 | 32.3 Shao et al. (2024) |
| | DeepSeek-math-7b-rl | 44.04 | 11.42 | 25.93 | 51.7 Shao et al. (2024) |
| | Qwen2.5-Math-7B | 56.65 | 22.92 | 40.46 | 55.4 Qwen et al. (2025) |
| | Qwen2.5-Math-7B-Instruct | 58.07 | 15.81 | 27.22 | 83.6 Qwen et al. (2025) |

## 4.2 MEASURING HIGH ENTROPY SEGMENTS

We propose a method to measure the change in TE per time step using a sliding window method to observe the extreme probability changes that may occur in some tokens (typically fork tokens) due to SFT or PO, although the average entropy of the entire generation was low and thus not noticed. Our goal is to measure, for each model, the ratio of the mean TE of a window of a given size to the mean TE of the entire generation. We define a window $w^{(i)} = \{\text{TE}_t\}_{t=i}^{i+W-1}$, starting at time step $i$, as a **High Entropy Segment (HES)** if its average entropy $\mu^{(i)} = \frac{1}{W} \sum_{t=i}^{i+W-1} \text{TE}_t$ exceeds the overall average entropy $\mu = \frac{1}{T} \sum_{t=1}^{T} \text{TE}_t$. In our experiments, we use a window size of $W = 10$ and a stride of $S = 5$. The formal definition of HES is as follows:

$$\text{HES}(w^{(i)}) = \begin{cases} 1, & \text{if } \mu^{(i)} > \mu \\ 0, & \text{otherwise} \end{cases} \qquad (3)$$

Since the sliding window moves with a stride $S$, multiple HES values can be assigned to a single time step $t$. Accordingly, the HES mask over the entire sequence, denoted as HESM, is defined such that $\text{HESM}(t) = 1$ if any window containing $t$ is identified as a HES (i.e., HES = 1).

**High Entropy Segments Ratio (HESR)**: This metric represents the proportion of tokens that fall within the high entropy regions out of the total number of response tokens $T$.

$$\text{HESR} = \frac{1}{T} \sum_{t=1}^{T} \text{HESM}[t] \qquad (4)$$

**Jaccard Similarity**: This metric measures the degree of overlap in high entropy regions between the generation model $g$ and the evaluation model $e$.

$$\text{Jaccard}(\text{HESM}_g, \text{HESM}_e) = \frac{|\text{HESM}_g \cap \text{HESM}_e|}{|\text{HESM}_g \cup \text{HESM}_e|} \qquad (5)$$

To compare HES differences across evaluation models, we used the metrics described above. We generate responses using the math and RL models of the Deepseek and Qwen families, and evaluate the HES of four models of each family. Figure 2 visualizes the HES for a single sample generated by each generation model, as evaluated by both the math and RL models. For the same generation, the HES varies depending on the evaluation model, and even in the RL models—well known to generate low-entropy outputs—some tokens exhibit notably high entropy.

## 5 RESULTS

### 5.1 EVALUATING CORRECTNESS & CREATIVITY

As shown in Table 1, the results indicate that the math models in both Qwen and Deepseek families generated siginifically more creative solutions than their RL model. Qwen-Math achieved a Creativity-to-Correctness ratio (Cre/Cor) of 40.46%, while Qwen-Inst, which underwent SFT and

Table 2: Results of evaluating Overconfidence (OC). The hyperparameters for determining OC are set to $q = 5\%$ and $\theta = 0.35$. **Gen** refers to the generation model, **Eval** refers to the evaluation model. **Ratio** indicates the proportion of OC (or NOC) samples. **Correct** and **Creative** are the correctness and creativity ratios, respectively, and **Cre/Cor** is the creativity-to-correctness ratio. **Bold** indicates the highest OC ratio within the same generation model.

| Gen | Eval | OC | | | | NOC | | | |
|---|---|---|---|---|---|---|---|---|---|
| | | **Ratio** | **Correct** | **Creative** | **Cre/Cor** | **Ratio** | **Correct** | **Creative** | **Cre/Cor** |
| DS Math | DS-Base | 9.9% | 65.1% | 24.2% | 37.1% | 90.1% | 32.2% | 15.6% | 51.5% |
| | DS-Math | **13.2%** | 60.8% | 23.6% | 38.8% | 86.8% | 31.6% | 16.4% | 51.8% |
| | DS-Inst | 9.3% | 58.0% | 23.6% | 37.1% | 90.7% | 31.6% | 16.7% | 51.3% |
| | DS-RL | 8.1% | 66.7% | 24.4% | 37.1% | 91.9% | 32.7% | 16.7% | 51.1% |
| DS RL | DS-Base | 0.6% | 70.0% | 0.0% | 0.0% | 9.4% | 44.0% | 11.5% | 26.2% |
| | DS-Math | 0.7% | 54.5% | 0.0% | 0.0% | 99.3% | 44.1% | 11.6% | 26.2% |
| | DS-Inst | 7.4% | 45.3% | 7.7% | 17.0% | 92.6% | 44.1% | 11.8% | 26.7% |
| | DS-RL | **28.9%** | 46.3% | 8.8% | 19.0% | 71.1% | 43.3% | 12.6% | 29.0% |
| QW Math | QW-Base | 0.6% | 55.6% | 0.0% | 0.0% | 99.4% | 56.7% | 23.1% | 40.7% |
| | QW-Math | **9.0%** | 73.0% | 26.3% | 36.0% | 91.0% | 55.0% | 32.4% | 41.0% |
| | QW-Inst | 3.1% | 75.0% | 22.9% | 30.6% | 96.9% | 56.1% | 22.9% | 40.9% |
| | QW-Distill | 0.2% | 100% | 0.0% | 0.0% | 99.8% | 56.6% | 23.0% | 40.6% |
| QW Inst | QW-Base | 3.6% | 85.5% | 25.5% | 29.8% | 96.4% | 57.1% | 15.4% | 27.0% |
| | QW-Math | 8.8% | 81.6% | 26.9% | 33.3% | 91.2% | 56.0% | 14.7% | 26.3% |
| | QW-Inst | **83.6%** | 63.9% | 17.0% | 26.6% | 16.4% | 28.9% | 9.6% | 33.3% |
| | QW-Distill | 1.5% | 73.7% | 30.4% | 43.8% | 98.5% | 52.3% | 15.5% | 26.8% |

GRPO, achieved only 27.22%. In the DeepSeek family, DeepSeek-Math recorded a Cre/Cor of 57.44%, whereas DeepSeek-RL reached only 25.93%, highlighting a substantial gap. Among open-source models Qwen-Inst, and DeepSeek-RL reported Cre of 15.81%, and 11.42%, respectively, indicating lower records compared to math models. However, the Qwen-Math achieved 56.65% on CreativeMath and 55.4% on the MATH, while Qwen-Inst achieved 58.07% and 83.6%. DeepSeek-Math recorded 34.68% and 32.3%, whereas DeepSeek-RL achieved 44.04% and 51.7%. While RL models show limitations in generating creative solutions that differ from reference solutions, they are proficient in solving problems with typical solutions.

## 5.2 EVALUATING OVERCONFIDENCE

We present the results of the OC evaluation in Table 2. The RL model (DS-RL, QW-Inst) showed notably high OC ratios of 28.9% and 83.6%, respectively, when evaluated by themselves, even under a strict threshold (0.35). These results indicate that the RL models were highly overconfident in their own generations. However, when the same generations were evaluated by other models, the OC ratio dropped significantly. DS-RL's generations yield OC ratios of 0.6%, 0.7%, and 7.4% when measured by the Base, Math, and Inst models, respectively. Similarly, QW-Inst's showed OC ratios of 3.6%, 8.8%, and 1.5% when measured by Base, Math, and Distill models.

Also, RL models did not exhibit remarkable correctness or creativity when they were overconfident: DS-RL had 8.8% Cre and 19% Cre/Cor when self-assessed as OC, which was lower than the 12.6% and 29% for non-OC; QW-Inst had 17% Cre and 26.6% Cre/Cor when self-assessed as OC, which was not significantly different from non-OC (9.6% and 33.3%), and it was difficult to judge because it generated OC at a very high rate. On the other hand, the generation of math models had a very low OC ratio overall. In DS-Math, both Cor and Cre were higher when the model was overconfident in its own generation than when it was not overconfident. In QW-Math, the percentage of OCs was less than 1%, making it difficult to draw a reliable conclusion.

In Table 1, we observed differences in the creative generation ability between the RL model and the math model. In Table 2, we measured the OC ratio and found that the RL model was overwhelmingly higher than the math model. This suggests that the phenomenon of excessive OC in the RL model may hinder its ability to generate a wide variety of outputs.

Table 3: Evaluation results of Qwen (QW) and Deepseek (DS) families. **Generation** indicates the model used to generate solutions; **Eval** denotes the evaluation model. **Hall** denotes a Hallucinated Solution; **Crea** a Creative Solution; **Typi** a Typical Solution. **Length** is the average output token lengths. **Entropy**, **HESR**, and **Jaccard** are reported as $\mu \pm \sigma$.

| Generation: DS-Math (Hall: 64.30%, Crea: 16.75%, Typi: 18.94%) | | | | | | | | | | | | | | | | |
|---|---|---|---|---|---|---|---|---|---|---|---|---|---|---|---|
| Eval | DS-Base | | | | DS-Math | | | | DS-Inst | | | | DS-RL | | | |
| Label | ALL | Hall | Crea | Typi | ALL | Hall | Crea | Typi | ALL | Hall | Crea | Typi | ALL | Hall | Crea | Typi |
| Length | 532.8 | 535.9 | 516.7 | 536.3 | 532.8 | 535.9 | 516.7 | 536.3 | 532.8 | 535.9 | 516.7 | 536.3 | 532.8 | 535.9 | 516.7 | 536.3 |
| Entropy | 0.73 ± 0.38 | 0.78 ± 0.37 | 0.74 ± 0.42 | 0.56 ± 0.33 | 0.69 ± 0.37 | 0.73 ± 0.36 | 0.70 ± 0.40 | 0.52 ± 0.32 | 0.47 ± 0.25 | 0.50 ± 0.25 | 0.47 ± 0.27 | 0.35 ± 0.21 | 0.43 ± 0.23 | 0.46 ± 0.23 | 0.44 ± 0.25 | 0.33 ± 0.20 |
| HESR | 0.43 ± 0.12 | 0.46 ± 0.11 | 0.41 ± 0.13 | 0.37 ± 0.13 | 0.43 ± 0.12 | 0.45 ± 0.11 | 0.41 ± 0.13 | 0.36 ± 0.13 | 0.44 ± 0.13 | 0.46 ± 0.12 | 0.42 ± 0.13 | 0.37 ± 0.14 | 0.44 ± 0.13 | 0.46 ± 0.11 | 0.43 ± 0.13 | 0.37 ± 0.14 |
| Jaccard | 0.90 ± 0.07 | 0.90 ± 0.07 | 0.92 ± 0.06 | 0.90 ± 0.09 | 1.00 ± 0.00 | 1.00 ± 0.00 | 1.00 ± 0.00 | 1.00 ± 0.00 | 0.85 ± 0.09 | 0.85 ± 0.08 | 0.86 ± 0.08 | 0.84 ± 0.10 | 0.83 ± 0.09 | 0.82 ± 0.09 | 0.83 ± 0.09 | 0.83 ± 0.11 |

| Generation: DS-RL (Hall: 55.96%, Crea: 11.42%, Typi: 32.62%) | | | | | | | | | | | | | | | | |
|---|---|---|---|---|---|---|---|---|---|---|---|---|---|---|---|---|
| Eval | DS-Base | | | | DS-Math | | | | DS-Inst | | | | DS-RL | | | |
| Label | ALL | Hall | Crea | Typi | ALL | Hall | Crea | Typi | ALL | Hall | Crea | Typi | ALL | Hall | Crea | Typi |
| Length | 426.8 | 486.1 | 378.7 | 342.0 | 426.8 | 486.1 | 378.7 | 342.0 | 426.8 | 486.1 | 378.7 | 342.0 | 426.8 | 486.1 | 378.7 | 342.0 |
| Entropy | 0.77 ± 0.26 | 0.76 ± 0.26 | 0.83 ± 0.27 | 0.75 ± 0.26 | 0.68 ± 0.25 | 0.67 ± 0.25 | 0.73 ± 0.26 | 0.67 ± 0.24 | 0.37 ± 0.15 | 0.37 ± 0.15 | 0.40 ± 0.14 | 0.36 ± 0.14 | 0.32 ± 0.13 | 0.32 ± 0.14 | 0.34 ± 0.13 | 0.31 ± 0.12 |
| HESR | 0.52 ± 0.08 | 0.53 ± 0.07 | 0.53 ± 0.07 | 0.51 ± 0.09 | 0.50 ± 0.08 | 0.51 ± 0.07 | 0.51 ± 0.08 | 0.49 ± 0.09 | 0.50 ± 0.08 | 0.51 ± 0.07 | 0.50 ± 0.08 | 0.49 ± 0.09 | 0.50 ± 0.08 | 0.51 ± 0.07 | 0.50 ± 0.08 | 0.50 ± 0.10 |
| Jaccard | 0.79 ± 0.09 | 0.80 ± 0.08 | 0.78 ± 0.10 | 0.80 ± 0.11 | 0.81 ± 0.10 | 0.81 ± 0.08 | 0.80 ± 0.10 | 0.80 ± 0.11 | 0.86 ± 0.08 | 0.87 ± 0.07 | 0.86 ± 0.09 | 0.85 ± 0.10 | 1.00 ± 0.00 | 1.00 ± 0.00 | 1.00 ± 0.00 | 1.00 ± 0.00 |

| Generation: QW-Math (Hall: 43.35%, Crea: 22.92%, Typi: 33.73%) | | | | | | | | | | | | | | | | |
|---|---|---|---|---|---|---|---|---|---|---|---|---|---|---|---|---|
| Eval | QW-Base | | | | QW-Math | | | | QW-Inst | | | | QW-Distill | | | |
| Label | ALL | Hall | Crea | Typi | ALL | Hall | Crea | Typi | ALL | Hall | Crea | Typi | ALL | Hall | Crea | Typi |
| Length | 651.7 | 684.5 | 634.3 | 621.3 | 651.7 | 684.5 | 634.3 | 621.3 | 651.7 | 684.5 | 634.3 | 621.3 | 651.7 | 684.5 | 634.3 | 621.3 |
| Entropy | 0.62 ± 0.21 | 0.69 ± 0.24 | 0.57 ± 0.15 | 0.58 ± 0.17 | 0.46 ± 0.19 | 0.52 ± 0.22 | 0.42 ± 0.14 | 0.42 ± 0.15 | 1.00 ± 0.61 | 1.28 ± 0.69 | 0.77 ± 0.41 | 0.79 ± 0.45 | 0.46 ± 0.17 | 0.51 ± 0.19 | 0.42 ± 0.12 | 0.41 ± 0.14 |
| HESR | 0.48 ± 0.07 | 0.49 ± 0.07 | 0.48 ± 0.06 | 0.47 ± 0.07 | 0.44 ± 0.07 | 0.45 ± 0.08 | 0.44 ± 0.07 | 0.43 ± 0.07 | 0.36 ± 0.10 | 0.39 ± 0.09 | 0.35 ± 0.09 | 0.34 ± 0.09 | 0.48 ± 0.06 | 0.50 ± 0.06 | 0.47 ± 0.06 | 0.46 ± 0.06 |
| Jaccard | 0.72 ± 0.10 | 0.73 ± 0.10 | 0.72 ± 0.10 | 0.72 ± 0.10 | 1.00 ± 0.00 | 1.00 ± 0.00 | 1.00 ± 0.00 | 1.00 ± 0.00 | 0.56 ± 0.14 | 0.54 ± 0.13 | 0.59 ± 0.13 | 0.57 ± 0.14 | 0.73 ± 0.09 | 0.72 ± 0.10 | 0.74 ± 0.09 | 0.74 ± 0.10 |

| Generation: QW-Inst (Hall: 41.93%, Crea: 15.81%, Typi: 42.26%) | | | | | | | | | | | | | | | | |
|---|---|---|---|---|---|---|---|---|---|---|---|---|---|---|---|---|
| Eval | QW-Base | | | | QW-Math | | | | QW-Inst | | | | QW-Distill | | | |
| Label | ALL | Hall | Crea | Typi | ALL | Hall | Crea | Typi | ALL | Hall | Crea | Typi | ALL | Hall | Crea | Typi |
| Length | 641.9 | 746.5 | 625.5 | 544.3 | 641.9 | 746.5 | 625.5 | 544.3 | 641.9 | 746.5 | 625.5 | 544.3 | 641.9 | 746.5 | 625.5 | 544.3 |
| Entropy | 0.44 ± 0.12 | 0.46 ± 0.12 | 0.43 ± 0.12 | 0.42 ± 0.12 | 0.36 ± 0.11 | 0.37 ± 0.11 | 0.35 ± 0.12 | 0.36 ± 0.12 | 0.40 ± 0.30 | 0.53 ± 0.33 | 0.31 ± 0.25 | 0.29 ± 0.22 | 0.30 ± 0.11 | 0.33 ± 0.11 | 0.28 ± 0.11 | 0.28 ± 0.10 |
| HESR | 0.52 ± 0.06 | 0.53 ± 0.05 | 0.53 ± 0.06 | 0.51 ± 0.06 | 0.49 ± 0.06 | 0.50 ± 0.06 | 0.50 ± 0.06 | 0.48 ± 0.06 | 0.43 ± 0.08 | 0.45 ± 0.07 | 0.42 ± 0.07 | 0.40 ± 0.08 | 0.50 ± 0.06 | 0.51 ± 0.06 | 0.50 ± 0.06 | 0.49 ± 0.06 |
| Jaccard | 0.56 ± 0.12 | 0.56 ± 0.12 | 0.55 ± 0.12 | 0.55 ± 0.11 | 0.58 ± 0.12 | 0.57 ± 0.12 | 0.58 ± 0.12 | 0.58 ± 0.12 | 1.00 ± 0.00 | 1.00 ± 0.00 | 1.00 ± 0.00 | 1.00 ± 0.00 | 0.58 ± 0.11 | 0.58 ± 0.11 | 0.59 ± 0.12 | 0.59 ± 0.11 |

## 5.3 MEASURING ENTROPY & OUTPUT TOKEN LENGTH

As shown in Table 3, although the RL model was additionally trained from the same foundation model, it consistently exhibited lower average entropy across all evaluation models compared to the math model, a finding that has already been widely reported (Yue et al., 2025). The DS-Math generation recorded average entropies of 0.73, 0.69, 0.47, and 0.43 across the four evaluation models, while the DS-RL generation showed slightly lower values of 0.77, 0.68, 0.37, and 0.32, generally exhibiting lower entropy in most cases. In particular, the RL models displayed very low entropy for their own generations but produced much higher entropy when evaluating outputs from the math models. In comparison, QW-Math generation recorded average entropies of 0.62, 0.46, 1.00, and 0.46, whereas QW-Inst generation showed consistently lower values of 0.44, 0.36, 0.40, and 0.30.

In addition, we found that the RL models applied non-standard evaluation criteria. When DS-RL was used as the evaluation model, the entropy values were 0.43 and 0.32 for DS-Math and DS-RL generations, respectively. When QW-Inst was used, it showed relatively large differences of 1.00 and 0.40 for QW-Math and QW-Inst generations. By contrast, models that did not undergo SFT or PO did not show large differences in entropy between math and RL outputs. For example, when DS-Base and DS-Math were used as evaluation models, they showed almost no difference between the generations of DS-Math and DS-RL, respectively (0.73, 0.77 and 0.69, 0.68). Similarly, when QW-Base and QW-Math were used as evaluation models, they showed relatively small differences between QW-Math and QW-Inst generations (0.62, 0.44 and 0.46, 0.36).

When examining entropy differences by class, the RL model generations did not show significant variation across classes (hallucinated, creative, typical) when entropy was measured by other models. For example, QW-Inst generations recorded entropy values of (0.46, 0.43, 0.42) on QW-Base, and (0.37, 0.35, 0.36) on QW-Math. Similarly, DS-RL generations recorded (0.76, 0.83, 0.75) on DS-Base, and (0.67, 0.73, 0.67) on DS-Math. By contrast, the math models consistently showed higher entropy for hallucinated solutions across all evaluation models, suggesting that methods previously used to detect hallucinations through uncertainty in output probabilities—such as entropy or perplexity—may not be effective for RL models that have undergone SFT and PO.

As shown in Table 3, the math models generated outputs of similar length across classes, whereas the RL models produced substantially different lengths depending on the class. This suggests that the RL models are optimized for typical solutions and may attempt to induce creativity by increasing token length. By contrast, the math models demonstrated consistent token lengths across all classes while naturally exhibiting creative generation.

## 5.4 Measuring High Entropy Segments

As shown in Table 3, RL models produced generations with lower average entropy than math models. However, their HESR values were consistently higher, a trend observed across all models within each family. Given the definition of HESR, this indicates that generations from RL models contained a larger proportion of tokens whose entropy exceeded the generation's average entropy, compared to those from math models. Recent research has shown that a small number of fork tokens—crucial to generation—possess high entropy, and that RL models focus on these tokens during the RL process (Shao et al., 2024; Wang et al., 2025). While this process adjusts the probabilities of fork tokens, it can also lead RL models into unstable paths. Therefore, we argue that the high HESR observed in RL model arises from their active use of fork tokens for creative generation, which can sometimes lead to paths with high uncertainty. It is also noteworthy that HESR is a new independent variable largely unaffected by the length or overall entropy of the generation. In Table 3, the generations of DS-Math and DS-RL exhibit different entropy across all four models of the DeepSeek family but show almost same HESR. Similarly, the generations of QW-Math and QW-Inst display comparable HESR across the three models in the Qwen family, excluding QW-Inst itself.

Moreover, we measured the HES of both math and RL models within each of the families and calculated their overlap using Equation 5. The results showed that the generations of RL models had lower similarity with other models compared to those of math models. For example, when we measured the Jaccard similarity between the HES of QW-Math and the other three models in the Qwen family, the similarities were 0.72 with QW-Base and 0.73 with QW-Distill, but only 0.56 with QW-Inst. In contrast, when comparing the HES of QW-Inst with the other models, the similarities were consistently low at 0.56, 0.58, and 0.58. The DeepSeek family exhibited a similar trend: DS-Math showed Jaccard similarities of 0.90, 0.85, and 0.83 with the other three models, whereas DS-RL recorded only 0.79, 0.81, and 0.86. Overall, RL models demonstrated relatively heterogeneous HES, with significant regions of uncertainty.

Lastly, QW-Distill was found to be highly stable in evaluating the generations of QW-Inst, with an average entropy of 0.30, while sharing a similar HES with QW-Math. As shown in Table 2, it even evaluated the generations of QW-Inst as having only 1.5% OC. This suggests that, although QW-Inst evaluates generations from other models in a highly heterogeneous manner—for instance, assigning an entropy of 1.00 to QW-Math's generations—QW-Distill may have the potential to perform well while remaining relatively unaffected by SFT or PO. A previous study reported that, when given sufficient opportunities for generation, the performances of the base, inst, and RL models converged,

Table 4: The table reports token-level Off-Argmax Ratios (OAR) between predicted and highest-probability token. The hyperparameters for determining OC are set to $q = 5\%$ and $\theta = 0.35$.

| Generation (total tokens) | DS-Math (804,574) | | | | DS-RL (676,570) | | | | QW-Math (995,140) | | | | QW-Inst (982,893) | | | |
|---|---|---|---|---|---|---|---|---|---|---|---|---|---|---|---|---|
| Evaluation | Base | Math | Inst | RL | Base | Math | Inst | RL | Base | Math | Inst | Distill | Base | Math | Inst | Distill |
| OC Ratio (%) | 9.9 | 13.2 | 9.3 | 8.1 | 0.6 | 0.7 | 7.45 | 28.9 | 0.6 | 0.9 | 3.1 | 0.2 | 3.6 | 8.8 | 83.6 | 1.5 |
| OAR (%) | 13.01 | 11.10 | 14.17 | 15.01 | 16.04 | 13.99 | 9.54 | 7.32 | 12.85 | 9.10 | 11.02 | 15.93 | 10.35 | 8.80 | 4.02 | 11.94 |
| OC OAR (%) | 2.78 | 2.86 | 3.16 | 3.07 | 3.40 | 3.55 | 4.82 | 4.77 | 4.55 | 4.96 | 4.64 | 4.00 | 5.01 | 4.93 | 3.39 | 5.17 |
| NOC OAR (%) | 14.33 | 12.59 | 15.51 | 16.24 | 16.15 | 14.12 | 10.15 | 8.74 | 12.90 | 9.56 | 11.27 | 15.96 | 10.54 | 9.21 | 6.78 | 12.07 |
| % tokens ≥ 75% | 73.84 | 74.85 | 79.66 | 80.52 | 68.81 | 71.63 | 81.36 | 83.34 | 72.36 | 78.17 | 80.08 | 76.88 | 78.92 | 82.23 | 89.11 | 83.36 |
| with top-1 ≥ **80%** | 71.31 | 72.53 | 77.49 | 78.38 | 65.46 | 68.68 | 78.92 | 81.11 | 69.14 | 75.47 | 77.11 | 74.07 | 75.93 | 79.74 | 87.01 | 80.87 |
| prob ≥ 85% | 68.40 | 69.70 | 74.86 | 75.81 | 61.77 | 65.06 | 75.93 | 78.36 | 65.58 | 72.54 | 73.43 | 71.14 | 72.57 | 76.90 | 84.38 | 78.34 |

whereas only the Distill model continued to perform better (Yue et al., 2025). We believe the Distill model has potential to generate creative solutions while demonstrating strong RL abilities.

## 5.5 MEASURING OFF-ARGMAX RATIOS

$$\text{Off-Argmax Ratios (OAR)} = \frac{1}{T} \sum_{t=1}^{T} \mathbf{1}\{y_t \neq \hat{y}_t\}, \quad \text{where } \hat{y}_t = \arg\max_{v \in V} P(v \mid y_{<t}) \quad (6)$$

We examined how the outputs of math and RL models differed from those generated by greedy decoding. As shown in Table 4, each of the four generators produced the lowest Off-Argmax Ratio (OAR) on its own generations. However, DS-Math recorded an OAR of 11.10%, whereas DS-RL recorded 7.32%, indicating that the RL model had a lower ratio. Similarly, QW-Math recorded 9.10%, while QW-Inst recorded only 4.02%. More specifically, we measured the OAR for both OC and non-OC cases, with all OC cases showing very small OAR that can be regarded as nearly greedy generations. As shown in Table 2, the RL model demonstrates a high rate of OC. Such a prevalence of OC suggests that the model likely fails to generate creative solutions.

We measured the proportion of tokens with a probability of top tokens greater than 80% to analyze the cause of OC, which can occur when tokens receive excessively high probabilities. As shown in Table 4, DS-Math generations recorded 72.53% for the Math model and 78.38% for the RL model (a difference of 5.85%), whereas DS-RL generations recorded 68.68% and 81.11% (a difference of 12.43%). Similarly, QW-Math generations recorded 75.47% and 77.11% (a difference of only 1.64%), while QW-Inst generations recorded 79.74% and 87.01% (a difference of 7.27%).

These results suggest that RL models contain a much higher proportion of high-probability tokens, whereas such tokens are relatively rare in other models. The problem is that the predominance of these tokens may hinder creative generation, as shown in Table 5, where increasing the temperature from 0.7 to 1.4 caused the RL model's correctness to drop sharply, reaching that of the math model. This implies that the common strategy of increasing temperature to generate creativity in model generation may be limited in RL models.

## 6 CONCLUSION

In this study, we focused on analyzing the creative abilities of the reasoning model. Despite their strengths in mathematical tasks, they did not outperform math models in creative generation. Our experimental results showed that, although reasoning models often exhibited (over)confidence in the creativity of their generations, the actual proportion of creative solutions was relatively low. To investigate the underlying reasons, we employed models from the Qwen and DeepSeek families as experimental groups, measuring and comparing metrics such as entropy, length, high-entropy segments, and off-argmax ratio. Our results suggest that current approaches to improving reasoning ability, such as instruction tuning and reinforcement learning, may inadvertently limit a model's capacity for creative generation. Thus, it is necessary to design training methods, and applying reasoning models to tasks where creativity is crucial still warrants further investigation.

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

# A  LIMITATIONS

We compared and analyzed the generations of the math and RL models, but we did not perform any SFT or GRPO processes on the models themselves. Because of the substantial resources required for fine-tuning, it was not feasible in our setting. Additionally, because we added reference solutions to the input prompt and set an output length of up to 1024 tokens to encourage a wide range of creative solutions, using LLMs larger than 7B was limited to due to resource constraints. Although there is a limitation in that math models did not generate the equal attempt (Please refer to Figure 4 in the Appendix), many prior studies have demonstrated that a sufficiently large number of trials k using the pass@k metric. However, if the same number of trials is guaranteed or evaluation is performed using greedy decoding, the RL model can clearly outperform the math model.

# B  DETAILS OF DATASET CONSTRUCTION

## B.1  RESPONSE GENERATION

### B.1.1  PROMPT FOR RESPONSE GENERATION

> Criteria for evaluating the difference between two mathematical solutions include:
> i). If the methods used to arrive at the solutions are fundamentally different, such as algebraic manipulation versus geometric reasoning, they can be considered distinct;
> ii). Even if the final results are the same, if the intermediate steps or processes involved in reaching those solutions vary significantly, the solutions can be considered different;
> iii). If two solutions rely on different assumptions or conditions, they are likely to be distinct;
> iv). A solution might generalize to a broader class of problems, while another solution might be specific to certain conditions. In such cases, they are considered distinct;
> v). If one solution is significantly simpler or more complex than the other, they can be regarded as essentially different, even if they lead to the same result.
>
> Given the following mathematical problem:
> **{problem}**
>
> And some typical solutions:
> **{reference_solutions}**
>
> Please output a novel solution distinct from the given ones for this math problem.

Figure 3: Input prompt used for generating responses on the CreativeMath dataset

Figure 3 shows the input prompt used to generate responses for the four models. This prompt, originally proposed by (Ye et al., 2025) to elicit creative response generation, includes a description of the criteria that define a creative solution, the problem statement, and a set of reference solutions (i.e., more than one), each written by humans. These reference solutions are provided in the CreativeMath dataset and consist of diverse and creative approaches.

### B.1.2 IMPLEMENTATION DETAILS

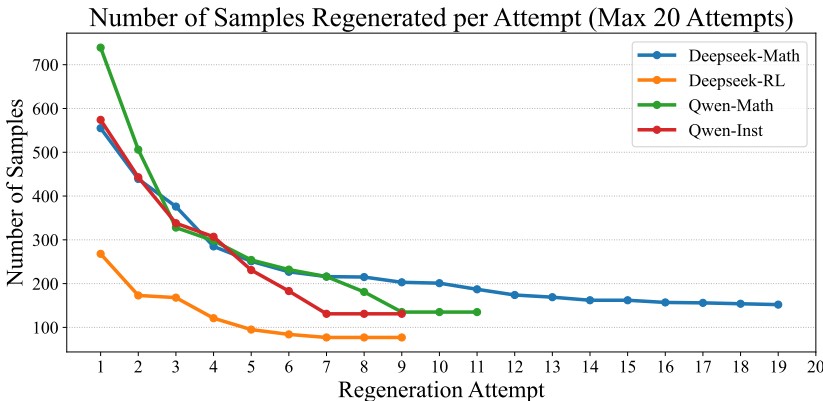

Figure 4: Number of samples requiring regeneration at each attempt for Deepseek-Math, Deepseek-RL, Qwen-Math, and Qwen-Inst. The maximum number of regeneration attempts was limited to 20.

The maximum number of reference solutions $k$ used in the prompts is set to two. The sampling settings were fixed at temperature = 0.7, top-p = 1.0, and top-k = 50 for all models. For each problem, the model generated three responses, with a maximum output token length constrained to 1024. If there is a problem with model response, such as when the response exceeds the maximum output token length or contains code that attempts to call an API key, regeneration was performed up to 20 times to mitigate imbalances in the number of response data samples across models. All generations were conducted on six NVIDIA RTX A5000 (24GB) GPUs.

### B.2 LABELING OF GENERATED RESPONSES

### B.2.1 LLM EVALUATOR

We employed two LLM evaluators to classify each response generated by the four models into one of three classes: *Hallucinated_Solution*, *Creative_Solution*, or *Typical_Solution*.

- **Gemini-1.5-pro** (models/gemini-1.5-pro-002)
- **GPT-o4-mini** (o4-mini-2025-04-16)

### B.2.2 EVALUATION OF GENERATED RESPONSES

> Given the following mathematical problem:
> **{problem}**
>
> Reference solutions:
> **{reference_solutions}**
>
> New solution:
> **{new_solution}**
>
> Please output "YES" if the new solution leads to the same result as the reference solutions; otherwise, output "NO".
> YES or NO?

Figure 5: Prompt for correctness evaluation of generated responses

Criteria for evaluating the novelty of a new mathematical solution include:
1. If the new solution used to arrive at the solutions is fundamentally different from reference solutions, such as algebraic manipulation versus geometric reasoning, it can be considered novel;
2. Even if the final results are the same, if the intermediate steps or processes involved in reaching those solutions vary significantly, the new solution can be considered novel;
3. If the new solution relies on different assumptions or conditions, it should be considered novel;
4. A solution might generalize to a broader class of problems, while another solution might be specific to certain conditions. In such cases, they are considered distinct;
5. If the new solution is significantly simpler or more complex than the others, it can be regarded as essentially novel, even if they lead to the same result.

Given the following mathematical problem:
**{problem}**

Reference solutions:
**{reference_solutions}**

New solution:
**{new_solution}**

Please output "YES" if the new solution is a novel solution; otherwise, output "NO".
YES or NO?

Figure 6: Prompt for creativity evaluation of generated responses

As previously mentioned, each response generated by the models is classified into one of three categories based on the judgments of two LLM evaluators. Figure 5 presents the prompt used to determine whether a response is a *Hallucinated_Solution*. Only if both evaluators agree that the response is not a hallucination, an additional evaluation is conducted for creativity using the prompt shown in Figure 6.

For creativity judgment, if either evaluator labels the response as a *Creative_Solution*, it is categorized as such. All responses generated by the four models are labeled according to these criteria.

### B.2.3    RELIABILITY AND JUSTIFICATION OF LLM-BASED LABELING

We conducted labeling based on the creativity evaluation framework proposed by (Ye et al., 2025). While the prior study employed three LLM evaluators and determined creativity through majority voting, we used only two LLM evaluators, including o4-mini, which demonstrates superior mathematical problem-solving capabilities, rather than the GPT-4o used in prior work. Consequently, it was necessary to establish a separate criterion to handle cases in which the two evaluators disagreed in their assessments.

To assess the consistency between the two evaluators, we computed Cohen's kappa coefficient, which yielded a value of 0.713—indicating a statistically substantial level of agreement. This suggests that the evaluators were generally well-aligned in their judgments of creativity. However, because creativity evaluation inherently involves a degree of subjectivity, requiring both evaluators to assess a generated response as a *Creative_Solution* could lead to the exclusion of genuinely creative responses. To address this, we adopted a more inclusive criterion: a response was labeled as a *Creative_Solution* if either evaluator judged it to be creative.

| Case 1: LLM Evaluator Output — Creative | Case 2: LLM Evaluator Output — Not Creative |
|---|---|
| The new solution uses a completely different approach (Lifting The Exponent Lemma) compared to the reference solutions (difference of squares factorization). This satisfies criterion 1 for novelty. While the final result is the same, the intermediate steps and the mathematical tools used are entirely distinct, satisfying criterion 2. The LTE lemma also relies on different underlying principles and conditions than simple algebraic manipulation, satisfying criterion 3. Therefore, the new solution is considered novel. | The new solution is not novel. It uses the same logical deductions as the reference solution. Both solutions arrive at the same answer through the same constraints. The core logic—Carl not being next to Bret, and Abby not being between Bret and Carl—is identical. The new solution merely restates these conditions and elaborates on the placement possibilities without introducing any new mathematical techniques, different assumptions, or a more generalized approach. Therefore, it does not meet any of the criteria for novelty. |

Figure 7: Example outputs from the LLM evaluators for determining creativity. The evaluators not only provide a binary judgment (Creative or Not Creative) but also justify their decision with a textual rationale based on the predefined criteria.

Furthermore, to assess how well the LLM evaluator understood the creativity criteria and provided consistent judgments, we modified the prompts to elicit not only labels but also supporting rationales for each decision. Specifically, we appended the sentence "Additionally, explain the reason for your judgment." to the end of the prompt, as shown in Figure 6. Figure 7 presents an example of the actual output generated by the LLM during this validation process. This procedure allowed us to verify that the LLM evaluator delivered reliable and well-justified assessments of creativity.

### B.2.4 RATIONALE FOR EXCLUDING HUMAN EVALUATION

Following the creative response generation and evaluation framework proposed by (Ye et al., 2025), we conducted fully automated evaluations of responses from all four models using only LLM evaluators, without any human involvement. Nevertheless, we acknowledge that human evaluation remains important for thoroughly assessing the creativity of generated responses. However, the CreativeMath dataset used in this study comprises highly challenging mathematical problems, which poses a practical limitation: evaluating the creativity of the generated solutions requires substantial mathematical expertise and time resources.

Nevertheless, as mentioned in Appendix B.1, the LLM evaluator receives as input a set of diverse and creative human-written solutions (i.e., reference solutions), which serve as a basis for evaluating the creativity of generated responses. Consequently, although direct human evaluation was not performed, the human-written creative solutions serve as a key comparative basis in the creativity assessment process. In this way, human creativity and judgment criteria are indirectly reflected in the overall assessment.

### B.3 STATISTICS OF GENERATED RESPONSES

Table 5 presents the evaluation results of model-generated responses under two different temperature ($T$) settings (0.7 and 1.4), following the setup described in Section 3.3. We compared the two temperature settings, considering that they can influence creative generation by modulating the diversity of model outputs.

As shown in Table 5, the $T = 1.4$ setting resulted in significantly lower correctness compared to $T = 0.7$, and the number of creative solutions dropped sharply, leading to a severe class imbalance problem. In addition, upon inspecting the generated responses, we observed that many were incomplete or contained Unicode characters, indicating that the overall response quality was poorer than under the $T = 0.7$ setting. As a result, the total number of usable samples in the dataset of model-generated responses was reduced, and we ultimately constructed the experimental dataset using only the responses generated with the $T = 0.7$ setting.

Table 5: Evaluation results of generated responses under different temperature ($T$) settings ($T = 0.7$ and $T = 1.4$) for four models. Metrics include Correctness (Cor), Creativity (Cre), and the Creativity-to-Correctness ratio (Cre/Cor), with higher values indicating better performance. Compared to $T = 0.7$, increasing temperature to $T = 1.4$ leads to a drastic reduction in the number of Creative solutions, resulting in severe class imbalance and degraded creativity.

| $T$ | Model | Cor (%) | Cre (%) | Cre/Cor (%) |
|---|---|---|---|---|
| 0.7 | QW-Math | 56.65 | 22.92 | 40.46 |
| | QW-Inst | 58.07 | 15.81 | 27.22 |
| | DS-Math | 34.68 | 19.92 | 57.44 |
| | DS-RL | 44.04 | 11.42 | 25.93 |
| 1.4 | QW-Math | 21.87 | 6.90 | 31.57 |
| | QW-Inst | 23.92 | 5.56 | 23.23 |
| | DS-Math | 19.84 | 6.81 | 34.32 |
| | DS-RL | 22.47 | 5.42 | 24.11 |

Table 6: Distribution of generation classes for each model. **Hall**, **Crea**, and **Typi** refer to the labels *Hallucinated_Solution*, *Creative_Solution*, and *Typical_Solution*, respectively.

| Generation Model | Hall | Crea | Typi | Total |
|---|---|---|---|---|
| QW-Math | 662 | 350 | 515 | 1527 |
| QW-Inst | 642 | 242 | 647 | 1531 |
| DS-Math | 895 | 276 | 339 | 1510 |
| DS-RL | 887 | 181 | 517 | 1585 |

Table 6 summarizes the distribution of model-generated responses under the $T = 0.7$ setting, categorized into three classes: *Hallucinated_Solution*, *Creative_Solution*, and *Typical_Solution*.

## C    RESULTS UNDER VARYING HYPERPARAMETERS FOR OC CLASSIFICATION

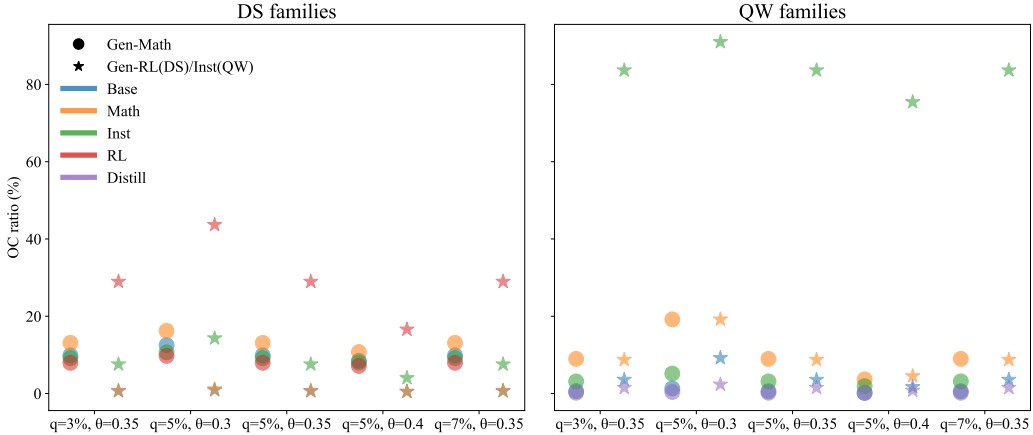

Figure 8: OC ratio results for DS and QW families across different $(q, \theta)$ pairs. Gen denotes the generation model: datasets generated by DS-Math and QW-Math are shown with circles, while those generated by DS-RL and QW-Inst are shown with stars. Colors indicate the evaluation models. Across various hyperparameter settings used to define OC, RL models tend to exhibit relatively higher OC ratios when evaluated on data generated by themselves compared to those generated by Math models.

We used the definition in Equation 2 to classify each generated response into one of two categories: OverConfidence (OC) and Non-OverConfidence (Non-OC). Figure 8 illustrates the OC ratios of

the two model families across different hyperparameter settings $(q, \theta)$ for this classification. Appendix C.1 and Appendix C.2 report additional evaluation results using alternative metrics, including Entropy, Token Length, HESR, and Jaccard Similarity. Specifically, Appendix C.1 provides results for the QW family, while Appendix C.2 presents those for the DS family.

## C.1 RESULTS OF QWEN FAMILIES

Tables 8–12 present the evaluation results obtained using the QW family of models on responses generated by QW-Math.

Among these, Tables 8–10 show the results when the hyperparameter $q$ is fixed at 5% and $\theta$ is varied across 0.30, 0.35, and 0.40. In contrast, Tables 11–12 show the results when $\theta$ is fixed at 0.35 and $q$ is varied across 3% and 7%.

Tables 13–17 report the results of evaluating responses generated by QW-Inst using the same QW family of models.

Specifically, Tables 13–15 correspond to the case where $q$ is fixed at 5% and $\theta$ is varied across 0.30, 0.35, and 0.40, while Tables 16–17 correspond to the case where $\theta$ is fixed at 0.35 and $q$ is set to 3% and 7%.

## C.2 RESULTS OF DEEPSEEK FAMILIES

Tables 18– 22 present the evaluation results obtained using the DS family of models on responses generated by DS-Math.

Among these, Tables 18–20 show the results when the hyperparameter $q$ is fixed at 5% and $\theta$ is varied across 0.30, 0.35, and 0.40. In contrast, Tables 21–22 report the results when $\theta$ is fixed at 0.35 and $q$ is varied across 3% and 7%.

Tables 23–27 present the evaluation results for responses generated by DS-RL, evaluated using the same DS family of models.

Specifically, Tables 23–25 correspond to the case where $q$ is fixed at 5% and $\theta$ is varied across 0.30, 0.35, and 0.40, while Tables 26–27 correspond to the case where $\theta$ is fixed at 0.35 and $q$ is set to 3% and 7%.

## D COMPARED WITH SELF-ASSIGNED CONFIDENCE AND OVERCONFIDENCE

Recently, there was an attempt to ask LLMs about their own confidence and calibrate the gap between confidence and accuracy. To verify the superiority of our proposed overconfidence definition, we asked the DS-math, DS-RL, QW-Math, and QW-Inst models to assign confidence scores to the responses they generated. And the relationship between confidence score, which is the degree to which one believes one's own creative generation, and creative generation is presented in Figure 9. As a result, there was almost no correlation between the two. As claimed in previous studies, it was confirmed that models that went through PO gave high confidence scores for their own response, but there was no significant correlation with creative generation in either the math model or the RL model.

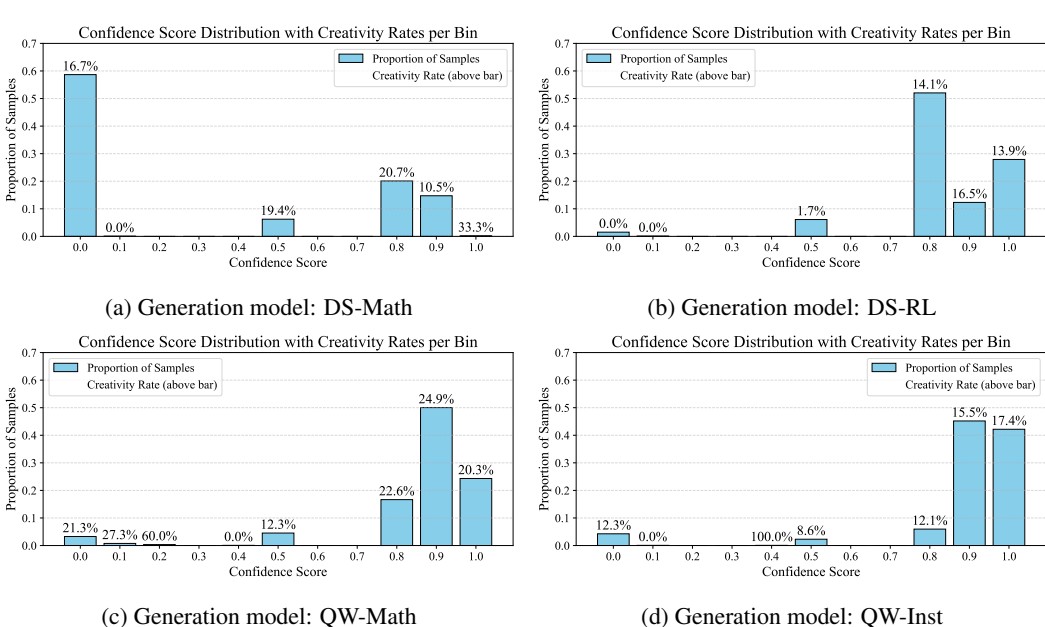

(a) Generation model: DS-Math      (b) Generation model: DS-RL

(c) Generation model: QW-Math      (d) Generation model: QW-Inst

Figure 9: Confidence score distribution and creativity rate analysis for responses generated by each generation model. Each subfigure corresponds to one generation model, and shows the distribution of confidence scores (x-axis) as the proportion of samples falling into each confidence bin (y-axis). The confidence score is computed by the model that generated the response. Bars are annotated with the percentage of samples in each bin that are labeled as creative solutions.

# E   ANALYZING TOKEN-LEVEL CHANGES IN SFT AND RL

Table 7: Average entropy differences between the Math model and the RL model (i.e., Math – RL) computed over the top 20% entropy and bottom 80% entropy token groups. Here, **Inc** denotes the proportion of tokens whose normalized entropy increased, **Dec** denotes the proportion whose normalized entropy decreased, **Norm △ Avg** is the average difference in min–max normalized entropy, and **Raw △ Avg** is the average difference in raw entropy values.

| Generation Model / Split | Inc (%) | Dec (%) | Norm △ Avg | Raw △ Avg |
|---|---|---|---|---|
| Deepseek-Math (Top 20%) | 37.69 | 62.31 | -0.04 | -0.79 |
| Deepseek-Math (Bottom 80%) | 19.84 | 80.12 | -0.00 | -0.09 |
| Qwen-Math (Top 20%) | 26.52 | 73.48 | -0.09 | 1.04 |
| Qwen-Math (Bottom 80%) | 68.41 | 31.58 | 0.02 | 0.43 |

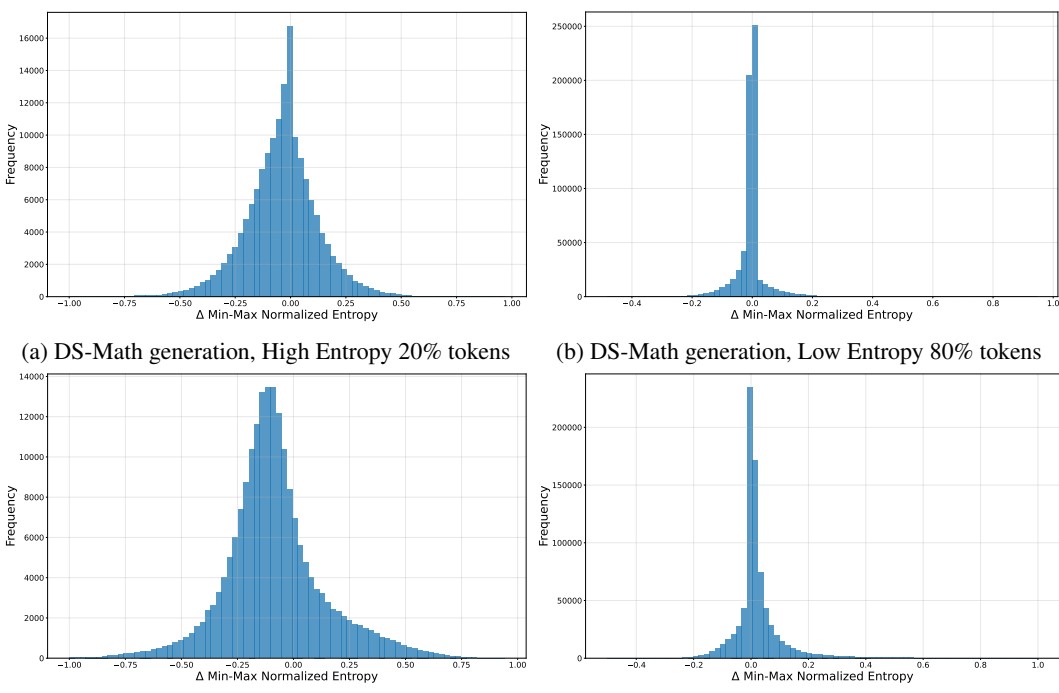

(a) DS-Math generation, High Entropy 20% tokens      (b) DS-Math generation, Low Entropy 80% tokens

(c) QW-Math generation, High Entropy 20% tokens      (d) QW-Inst generation, Low Entropy 80% tokens

Figure 10: Average token-level entropy differences obtained by feeding the generations produced by the Math model into the Math model and the RL model. Panels (a) and (c) show the entropy changes for the top 20% highest-entropy tokens (based on the Math model), while panels (b) and (d) show the changes for the bottom 80% tokens. Panels (a) and (c) use generations from the DS-Math model, and panels (b) and (d) use generations from the QW-Math model.

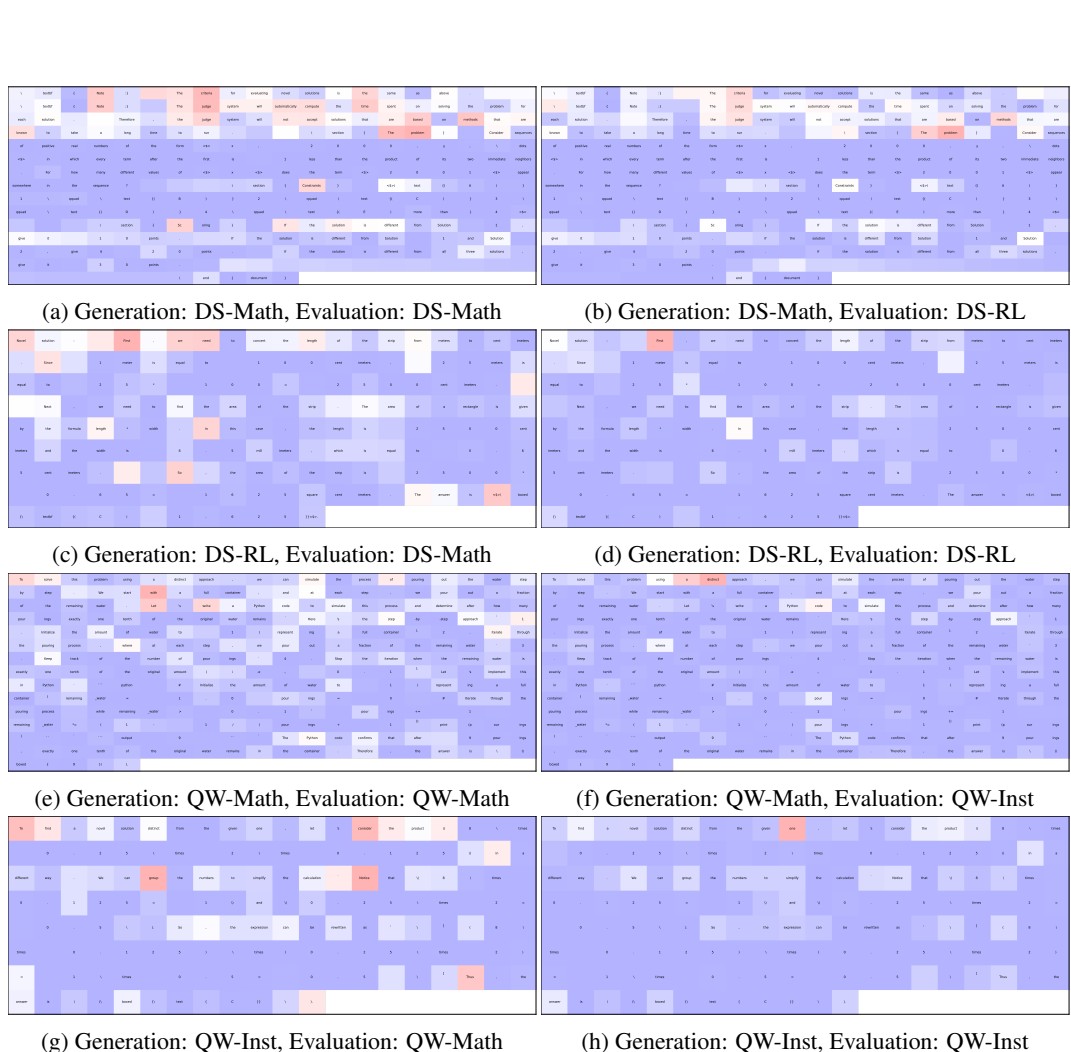

(a) Generation: DS-Math, Evaluation: DS-Math

(b) Generation: DS-Math, Evaluation: DS-RL

(c) Generation: DS-RL, Evaluation: DS-Math

(d) Generation: DS-RL, Evaluation: DS-RL

(e) Generation: QW-Math, Evaluation: QW-Math

(f) Generation: QW-Math, Evaluation: QW-Inst

(g) Generation: QW-Inst, Evaluation: QW-Math

(h) Generation: QW-Inst, Evaluation: QW-Inst

Figure 11: Visualization of entropy changes across evaluation models for the same generations. Blue indicates lower entropy, and red indicates higher entropy.

Table 8: Evaluation results on responses generated by the QW-Math model, categorized into two response types—Overconfidence (OC) and Non-Overconfidence (Non-OC)—across four QW family variants. The hyperparameters for determining response types are set to $q = 5\%$ and $\theta = 0.30$, as defined in Equation 2. Model denotes the evaluation model. ALL, Hall, Crea, and Typi refer to the full dataset, *Hallucinated_Solution*, *Creative_Solution*, and *Typical_Solution* samples, respectively. Length indicates the average token length of generated responses. Entropy, HESR, and Jaccard are reported as $\mu \pm \sigma$.

| Type | Model | Label (%) | | Length | Entropy | HESR | Jaccard |
|------|-------|------|------|--------|---------|------|---------|
| OC | QW-Base | ALL | 100.00 | 736.10 | 0.30 ± 0.05 | 0.47 ± 0.06 | 0.64 ± 0.13 |
| | | Hall | 20.00 | 650.00 | 0.28 ± 0.02 | 0.47 ± 0.03 | 0.76 ± 0.07 |
| | | Crea | 25.00 | 724.20 | 0.32 ± 0.03 | 0.47 ± 0.06 | 0.61 ± 0.10 |
| | | Typi | 55.00 | 772.82 | 0.29 ± 0.06 | 0.47 ± 0.07 | 0.60 ± 0.14 |
| | QW-Math | ALL | 100.00 | 725.00 | 0.28 ± 0.05 | 0.44 ± 0.07 | 1.00 ± 0.00 |
| | | Hall | 29.69 | 821.41 | 0.28 ± 0.05 | 0.45 ± 0.07 | 1.00 ± 0.00 |
| | | Crea | 27.30 | 718.36 | 0.28 ± 0.05 | 0.45 ± 0.06 | 1.00 ± 0.00 |
| | | Typi | 43.00 | 662.64 | 0.27 ± 0.06 | 0.44 ± 0.07 | 1.00 ± 0.00 |
| | QW-Inst | ALL | 100.00 | 796.52 | 0.49 ± 0.32 | 0.40 ± 0.08 | 0.54 ± 0.14 |
| | | Hall | 29.11 | 841.65 | 0.51 ± 0.30 | 0.39 ± 0.07 | 0.52 ± 0.12 |
| | | Crea | 24.05 | 749.53 | 0.40 ± 0.17 | 0.38 ± 0.08 | 0.60 ± 0.12 |
| | | Typi | 46.84 | 792.59 | 0.53 ± 0.38 | 0.40 ± 0.08 | 0.51 ± 0.14 |
| | QW-Distill | ALL | 100.00 | 851.83 | 0.14 ± 0.04 | 0.44 ± 0.05 | 0.61 ± 0.11 |
| | | Hall | 16.67 | 936.00 | 0.17 ± 0.00 | 0.40 ± 0.00 | 0.72 ± 0.00 |
| | | Crea | - | - | - | - | - |
| | | Typi | 83.33 | 835.00 | 0.14 ± 0.04 | 0.45 ± 0.05 | 0.59 ± 0.11 |
| Non-OC | QW-Base | ALL | 100.00 | 650.58 | 0.63 ± 0.20 | 0.48 ± 0.07 | 0.72 ± 0.10 |
| | | Hall | 43.66 | 684.71 | 0.69 ± 0.24 | 0.49 ± 0.07 | 0.73 ± 0.10 |
| | | Crea | 22.89 | 633.07 | 0.58 ± 0.15 | 0.48 ± 0.06 | 0.72 ± 0.10 |
| | | Typi | 33.44 | 617.99 | 0.58 ± 0.16 | 0.47 ± 0.07 | 0.72 ± 0.10 |
| | QW-Math | ALL | 100.00 | 634.29 | 0.51 ± 0.18 | 0.44 ± 0.07 | 1.00 ± 0.00 |
| | | Hall | 46.60 | 663.79 | 0.55 ± 0.21 | 0.45 ± 0.08 | 1.00 ± 0.00 |
| | | Crea | 21.88 | 609.49 | 0.46 ± 0.13 | 0.44 ± 0.07 | 1.00 ± 0.00 |
| | | Typi | 31.52 | 607.90 | 0.47 ± 0.14 | 0.43 ± 0.07 | 1.00 ± 0.00 |
| | QW-Inst | ALL | 100.00 | 643.79 | 1.02 ± 0.61 | 0.36 ± 0.10 | 0.56 ± 0.13 |
| | | Hall | 44.13 | 678.85 | 1.31 ± 0.69 | 0.39 ± 0.09 | 0.54 ± 0.13 |
| | | Crea | 22.86 | 627.76 | 0.79 ± 0.41 | 0.35 ± 0.09 | 0.59 ± 0.13 |
| | | Typi | 33.01 | 608.04 | 0.81 ± 0.45 | 0.34 ± 0.09 | 0.58 ± 0.13 |
| | QW-Distill | ALL | 100.00 | 650.91 | 0.46 ± 0.17 | 0.48 ± 0.06 | 1.00 ± 0.00 |
| | | Hall | 43.46 | 684.12 | 0.51 ± 0.19 | 0.50 ± 0.06 | 1.00 ± 0.00 |
| | | Crea | 23.01 | 634.37 | 0.42 ± 0.12 | 0.47 ± 0.06 | 1.00 ± 0.00 |
| | | Typi | 33.53 | 619.20 | 0.42 ± 0.13 | 0.46 ± 0.06 | 1.00 ± 0.00 |

Table 9: Evaluation results of the QW family on responses generated by the QW-Math model ($q = 5\%, \theta = 0.35$).

| Type | Model | Label (%) | | Length | Entropy | HESR | Jaccard |
|---|---|---|---|---|---|---|---|
| OC | QW-Base | ALL | 100.00 | 741.78 | $0.27 \pm 0.04$ | $0.48 \pm 0.03$ | $0.64 \pm 0.15$ |
| | | Hall | 44.44 | 650.00 | $0.28 \pm 0.02$ | $0.47 \pm 0.03$ | $0.76 \pm 0.07$ |
| | | Crea | - | - | - | - | - |
| | | Typi | 55.56 | 815.20 | $0.27 \pm 0.05$ | $0.49 \pm 0.02$ | $0.54 \pm 0.13$ |
| | QW-Math | ALL | 100.00 | 715.29 | $0.25 \pm 0.05$ | $0.44 \pm 0.07$ | $1.00 \pm 0.00$ |
| | | Hall | 27.01 | 804.54 | $0.25 \pm 0.05$ | $0.46 \pm 0.07$ | $1.00 \pm 0.00$ |
| | | Crea | 26.28 | 691.94 | $0.25 \pm 0.04$ | $0.44 \pm 0.06$ | $1.00 \pm 0.00$ |
| | | Typi | 46.72 | 676.83 | $0.25 \pm 0.06$ | $0.43 \pm 0.08$ | $1.00 \pm 0.00$ |
| | QW-Inst | ALL | 100.00 | 785.44 | $0.40 \pm 0.24$ | $0.38 \pm 0.08$ | $0.55 \pm 0.12$ |
| | | Hall | 25.00 | 838.75 | $0.35 \pm 0.19$ | $0.36 \pm 0.06$ | $0.55 \pm 0.10$ |
| | | Crea | 22.92 | 725.27 | $0.35 \pm 0.10$ | $0.38 \pm 0.08$ | $0.61 \pm 0.10$ |
| | | Typi | 52.08 | 786.32 | $0.45 \pm 0.29$ | $0.39 \pm 0.09$ | $0.53 \pm 0.13$ |
| | QW-Distill | ALL | 100.00 | 841.00 | $0.12 \pm 0.04$ | $0.43 \pm 0.01$ | $0.57 \pm 0.12$ |
| | | Hall | - | - | - | - | - |
| | | Crea | - | - | - | - | - |
| | | Typi | 100.00 | 841.00 | $0.12 \pm 0.04$ | $0.43 \pm 0.01$ | $0.57 \pm 0.12$ |
| Non-OC | QW-Base | ALL | 100.00 | 651.16 | $0.63 \pm 0.21$ | $0.48 \pm 0.07$ | $0.72 \pm 0.10$ |
| | | Hall | 43.35 | 684.71 | $0.69 \pm 0.24$ | $0.49 \pm 0.07$ | $0.73 \pm 0.10$ |
| | | Crea | 23.06 | 634.37 | $0.57 \pm 0.15$ | $0.48 \pm 0.06$ | $0.72 \pm 0.10$ |
| | | Typi | 33.60 | 619.40 | $0.58 \pm 0.17$ | $0.47 \pm 0.07$ | $0.72 \pm 0.10$ |
| | QW-Math | ALL | 100.00 | 645.43 | $0.48 \pm 0.18$ | $0.44 \pm 0.07$ | $1.00 \pm 0.00$ |
| | | Hall | 44.96 | 677.40 | $0.53 \pm 0.22$ | $0.45 \pm 0.08$ | $1.00 \pm 0.00$ |
| | | Crea | 22.59 | 627.77 | $0.44 \pm 0.13$ | $0.44 \pm 0.07$ | $1.00 \pm 0.00$ |
| | | Typi | 32.45 | 613.42 | $0.45 \pm 0.14$ | $0.43 \pm 0.07$ | $1.00 \pm 0.00$ |
| | QW-Inst | ALL | 100.00 | 647.36 | $1.02 \pm 0.61$ | $0.36 \pm 0.10$ | $0.56 \pm 0.14$ |
| | | Hall | 43.95 | 681.66 | $1.30 \pm 0.69$ | $0.39 \pm 0.09$ | $0.54 \pm 0.13$ |
| | | Crea | 22.92 | 631.42 | $0.78 \pm 0.41$ | $0.35 \pm 0.09$ | $0.59 \pm 0.13$ |
| | | Typi | 33.13 | 612.88 | $0.81 \pm 0.45$ | $0.34 \pm 0.09$ | $0.58 \pm 0.14$ |
| | QW-Distill | ALL | 100.00 | 651.32 | $0.46 \pm 0.17$ | $0.48 \pm 0.06$ | $1.00 \pm 0.00$ |
| | | Hall | 43.44 | 684.50 | $0.51 \pm 0.19$ | $0.50 \pm 0.06$ | $1.00 \pm 0.00$ |
| | | Crea | 22.97 | 634.37 | $0.42 \pm 0.12$ | $0.47 \pm 0.06$ | $1.00 \pm 0.00$ |
| | | Typi | 33.60 | 620.01 | $0.42 \pm 0.13$ | $0.46 \pm 0.06$ | $1.00 \pm 0.00$ |

Table 10: Evaluation results of the QW family on responses generated by the QW-Math model ($q = 5\%, \theta = 0.40$).

| Type | Model | Label (%) | | Length | Entropy | HESR | Jaccard |
|---|---|---|---|---|---|---|---|
| OC | QW-Base | ALL | 100.00 | 811.67 | 0.25 ± 0.01 | 0.49 ± 0.02 | 0.58 ± 0.14 |
| | | Hall | 33.33 | 680.00 | 0.25 ± 0.00 | 0.48 ± 0.00 | 0.66 ± 0.00 |
| | | Crea | - | - | - | - | - |
| | | Typi | 66.67 | 877.50 | 0.24 ± 0.01 | 0.49 ± 0.02 | 0.54 ± 0.16 |
| | QW-Math | ALL | 100.00 | 754.55 | 0.21 ± 0.04 | 0.42 ± 0.08 | 1.00 ± 0.00 |
| | | Hall | 26.79 | 805.27 | 0.22 ± 0.03 | 0.46 ± 0.06 | 1.00 ± 0.00 |
| | | Crea | 23.21 | 736.46 | 0.23 ± 0.04 | 0.44 ± 0.07 | 1.00 ± 0.00 |
| | | Typi | 50.00 | 735.79 | 0.21 ± 0.05 | 0.40 ± 0.09 | 1.00 ± 0.00 |
| | QW-Inst | ALL | 100.00 | 817.00 | 0.41 ± 0.28 | 0.38 ± 0.09 | 0.51 ± 0.10 |
| | | Hall | 20.69 | 811.33 | 0.26 ± 0.10 | 0.35 ± 0.05 | 0.54 ± 0.11 |
| | | Crea | 17.24 | 767.60 | 0.33 ± 0.10 | 0.36 ± 0.09 | 0.55 ± 0.02 |
| | | Typi | 62.07 | 832.61 | 0.48 ± 0.32 | 0.39 ± 0.10 | 0.49 ± 0.10 |
| | QW-Distill | ALL | 100.00 | 820.50 | 0.10 ± 0.03 | 0.44 ± 0.01 | 0.53 ± 0.14 |
| | | Hall | - | - | - | - | - |
| | | Crea | - | - | - | - | - |
| | | Typi | 100.00 | 820.50 | 0.10 ± 0.03 | 0.44 ± 0.01 | 0.53 ± 0.14 |
| Non-OC | QW-Base | ALL | 100.00 | 651.38 | 0.62 ± 0.21 | 0.48 ± 0.07 | 0.72 ± 0.10 |
| | | Hall | 43.37 | 684.51 | 0.69 ± 0.24 | 0.49 ± 0.07 | 0.73 ± 0.10 |
| | | Crea | 22.97 | 634.37 | 0.57 ± 0.15 | 0.48 ± 0.06 | 0.72 ± 0.10 |
| | | Typi | 33.66 | 620.30 | 0.58 ± 0.17 | 0.47 ± 0.07 | 0.72 ± 0.10 |
| | QW-Math | ALL | 100.00 | 647.78 | 0.47 ± 0.18 | 0.44 ± 0.07 | 1.00 ± 0.00 |
| | | Hall | 43.98 | 681.70 | 0.52 ± 0.22 | 0.45 ± 0.08 | 1.00 ± 0.00 |
| | | Crea | 22.91 | 630.43 | 0.43 ± 0.14 | 0.44 ± 0.07 | 1.00 ± 0.00 |
| | | Typi | 33.11 | 614.71 | 0.43 ± 0.15 | 0.44 ± 0.07 | 1.00 ± 0.00 |
| | QW-Inst | ALL | 100.00 | 648.50 | 1.01 ± 0.61 | 0.36 ± 0.10 | 0.56 ± 0.14 |
| | | Hall | 43.79 | 683.34 | 1.29 ± 0.69 | 0.39 ± 0.09 | 0.54 ± 0.13 |
| | | Crea | 23.03 | 632.44 | 0.78 ± 0.41 | 0.35 ± 0.09 | 0.59 ± 0.13 |
| | | Typi | 33.18 | 613.64 | 0.80 ± 0.45 | 0.34 ± 0.09 | 0.58 ± 0.14 |
| | QW-Distill | ALL | 100.00 | 651.47 | 0.46 ± 0.17 | 0.48 ± 0.06 | 1.00 ± 0.00 |
| | | Hall | 43.41 | 684.50 | 0.51 ± 0.19 | 0.50 ± 0.06 | 1.00 ± 0.00 |
| | | Crea | 22.95 | 634.37 | 0.42 ± 0.12 | 0.47 ± 0.06 | 1.00 ± 0.00 |
| | | Typi | 33.64 | 620.52 | 0.42 ± 0.14 | 0.46 ± 0.06 | 1.00 ± 0.00 |

Table 11: Evaluation results of the QW family on responses generated by the QW-Math model ($q = 3\%, \theta = 0.35$).

| Type | Model | Label (%) | | Length | Entropy | HESR | Jaccard |
|---|---|---|---|---|---|---|---|
| OC | QW-Base | ALL | 100.00 | 873.00 | 0.25 ± 0.00 | 0.51 ± 0.00 | 0.38 ± 0.00 |
| | | Hall | - | - | - | - | - |
| | | Crea | - | - | - | - | - |
| | | Typi | 100.00 | 873.00 | 0.25 ± 0.00 | 0.51 ± 0.00 | 0.38 ± 0.00 |
| | QW-Math | ALL | 100.00 | 725.38 | 0.19 ± 0.05 | 0.42 ± 0.08 | 1.00 ± 0.00 |
| | | Hall | 7.69 | 1006.00 | 0.19 ± 0.00 | 0.49 ± 0.00 | 1.00 ± 0.00 |
| | | Crea | 15.38 | 816.00 | 0.17 ± 0.00 | 0.45 ± 0.01 | 1.00 ± 0.00 |
| | | Typi | 76.92 | 679.20 | 0.19 ± 0.06 | 0.40 ± 0.08 | 1.00 ± 0.00 |
| | QW-Inst | ALL | 100.00 | 799.57 | 0.51 ± 0.48 | 0.43 ± 0.08 | 0.51 ± 0.13 |
| | | Hall | 14.29 | 680.00 | 0.11 ± 0.00 | 0.37 ± 0.00 | 0.71 ± 0.00 |
| | | Crea | 14.29 | 725.00 | 0.17 ± 0.00 | 0.38 ± 0.00 | 0.55 ± 0.00 |
| | | Typi | 71.43 | 838.40 | 0.65 ± 0.50 | 0.45 ± 0.08 | 0.45 ± 0.11 |
| | QW-Distill | ALL | 100.00 | 873.00 | 0.07 ± 0.00 | 0.44 ± 0.00 | 0.39 ± 0.00 |
| | | Hall | - | - | - | - | - |
| | | Crea | - | - | - | - | - |
| | | Typi | 100.00 | 873.00 | 0.07 ± 0.00 | 0.44 ± 0.00 | 0.39 ± 0.00 |
| Non-OC | QW-Base | ALL | 100.00 | 651.55 | 0.62 ± 0.21 | 0.48 ± 0.07 | 0.72 ± 0.10 |
| | | Hall | 43.38 | 684.50 | 0.69 ± 0.24 | 0.49 ± 0.07 | 0.73 ± 0.10 |
| | | Crea | 22.94 | 634.37 | 0.57 ± 0.15 | 0.48 ± 0.06 | 0.72 ± 0.10 |
| | | Typi | 33.68 | 620.81 | 0.58 ± 0.17 | 0.47 ± 0.07 | 0.72 ± 0.10 |
| | QW-Math | ALL | 100.00 | 651.06 | 0.46 ± 0.19 | 0.44 ± 0.07 | 1.00 ± 0.00 |
| | | Hall | 43.66 | 684.02 | 0.52 ± 0.22 | 0.45 ± 0.08 | 1.00 ± 0.00 |
| | | Crea | 22.99 | 633.33 | 0.42 ± 0.14 | 0.44 ± 0.07 | 1.00 ± 0.00 |
| | | Typi | 33.36 | 620.15 | 0.43 ± 0.15 | 0.43 ± 0.07 | 1.00 ± 0.00 |
| | QW-Inst | ALL | 100.00 | 651.02 | 1.00 ± 0.61 | 0.36 ± 0.10 | 0.56 ± 0.14 |
| | | Hall | 43.49 | 684.51 | 1.28 ± 0.69 | 0.39 ± 0.09 | 0.54 ± 0.13 |
| | | Crea | 22.96 | 634.11 | 0.77 ± 0.41 | 0.35 ± 0.09 | 0.59 ± 0.13 |
| | | Typi | 33.55 | 619.17 | 0.79 ± 0.45 | 0.34 ± 0.09 | 0.58 ± 0.14 |
| | QW-Distill | ALL | 100.00 | 651.55 | 0.46 ± 0.17 | 0.48 ± 0.06 | 0.73 ± 0.09 |
| | | Hall | 43.38 | 684.50 | 0.51 ± 0.19 | 0.50 ± 0.06 | 0.72 ± 0.10 |
| | | Crea | 22.94 | 634.37 | 0.42 ± 0.12 | 0.47 ± 0.06 | 0.74 ± 0.09 |
| | | Typi | 33.68 | 620.81 | 0.42 ± 0.14 | 0.46 ± 0.06 | 0.74 ± 0.09 |

Table 12: Evaluation results of the QW family on responses generated by the QW-Math model ($q = 7\%, \theta = 0.35$).

| Type | Model | Label (%) | | Length | Entropy | HESR | Jaccard |
|---|---|---|---|---|---|---|---|
| OC | QW-Base | ALL | 100.00 | 719.98 | 0.32 ± 0.05 | 0.45 ± 0.06 | 0.65 ± 0.12 |
| | | Hall | 29.17 | 775.21 | 0.32 ± 0.05 | 0.47 ± 0.04 | 0.70 ± 0.12 |
| | | Crea | 27.08 | 603.54 | 0.33 ± 0.03 | 0.46 ± 0.06 | 0.63 ± 0.08 |
| | | Typi | 43.75 | 755.24 | 0.32 ± 0.06 | 0.44 ± 0.07 | 0.63 ± 0.13 |
| | QW-Math | ALL | 100.00 | 731.08 | 0.30 ± 0.06 | 0.44 ± 0.07 | 1.00 ± 0.00 |
| | | Hall | 31.85 | 823.82 | 0.31 ± 0.06 | 0.44 ± 0.07 | 1.00 ± 0.00 |
| | | Crea | 26.33 | 720.09 | 0.30 ± 0.06 | 0.44 ± 0.07 | 1.00 ± 0.00 |
| | | Typi | 41.83 | 667.39 | 0.30 ± 0.06 | 0.43 ± 0.07 | 1.00 ± 0.00 |
| | QW-Inst | ALL | 100.00 | 735.45 | 0.49 ± 0.25 | 0.37 ± 0.09 | 0.54 ± 0.13 |
| | | Hall | 26.09 | 828.04 | 0.56 ± 0.24 | 0.39 ± 0.08 | 0.52 ± 0.10 |
| | | Crea | 32.37 | 713.36 | 0.45 ± 0.18 | 0.37 ± 0.09 | 0.58 ± 0.11 |
| | | Typi | 41.55 | 694.53 | 0.48 ± 0.29 | 0.36 ± 0.09 | 0.53 ± 0.15 |
| | QW-Distill | ALL | 100.00 | 787.28 | 0.18 ± 0.04 | 0.46 ± 0.04 | 0.63 ± 0.11 |
| | | Hall | 38.89 | 770.86 | 0.21 ± 0.02 | 0.47 ± 0.05 | 0.67 ± 0.12 |
| | | Crea | 5.56 | 725.00 | 0.16 ± 0.00 | 0.47 ± 0.00 | 0.60 ± 0.00 |
| | | Typi | 55.56 | 805.00 | 0.16 ± 0.04 | 0.45 ± 0.04 | 0.60 ± 0.08 |
| Non-OC | QW-Base | ALL | 100.00 | 649.48 | 0.63 ± 0.20 | 0.48 ± 0.07 | 0.72 ± 0.10 |
| | | Hall | 43.81 | 682.54 | 0.70 ± 0.23 | 0.49 ± 0.07 | 0.73 ± 0.10 |
| | | Crea | 22.79 | 635.56 | 0.58 ± 0.15 | 0.48 ± 0.06 | 0.72 ± 0.10 |
| | | Typi | 33.40 | 615.60 | 0.59 ± 0.16 | 0.47 ± 0.07 | 0.72 ± 0.10 |
| | QW-Math | ALL | 100.00 | 616.29 | 0.53 ± 0.18 | 0.45 ± 0.07 | 1.00 ± 0.00 |
| | | Hall | 48.48 | 643.69 | 0.58 ± 0.21 | 0.45 ± 0.08 | 1.00 ± 0.00 |
| | | Crea | 21.40 | 587.34 | 0.49 ± 0.13 | 0.44 ± 0.07 | 1.00 ± 0.00 |
| | | Typi | 30.11 | 592.74 | 0.50 ± 0.14 | 0.43 ± 0.07 | 1.00 ± 0.00 |
| | QW-Inst | ALL | 100.00 | 638.56 | 1.08 ± 0.62 | 0.36 ± 0.10 | 0.57 ± 0.14 |
| | | Hall | 46.06 | 671.76 | 1.34 ± 0.68 | 0.39 ± 0.09 | 0.54 ± 0.14 |
| | | Crea | 21.44 | 615.67 | 0.85 ± 0.42 | 0.35 ± 0.09 | 0.59 ± 0.13 |
| | | Typi | 32.50 | 606.62 | 0.85 ± 0.45 | 0.34 ± 0.09 | 0.58 ± 0.13 |
| | QW-Distill | ALL | 100.00 | 650.08 | 0.46 ± 0.17 | 0.48 ± 0.06 | 0.73 ± 0.09 |
| | | Hall | 43.41 | 683.58 | 0.52 ± 0.19 | 0.50 ± 0.06 | 0.72 ± 0.10 |
| | | Crea | 23.13 | 634.11 | 0.42 ± 0.12 | 0.47 ± 0.06 | 0.74 ± 0.09 |
| | | Typi | 33.47 | 617.66 | 0.42 ± 0.13 | 0.46 ± 0.06 | 0.74 ± 0.09 |

Table 13: Evaluation results on responses generated by the QW-Inst model, categorized into two response types—Overconfidence (OC) and Non-Overconfidence (Non-OC)—across four QW family variants. The hyperparameters for determining response types are set to $q = 5\%$ and $\theta = 0.30$, as defined in Equation 2. Model denotes the evaluation model. ALL, Hall, Crea, and Typi refer to the full dataset, *Hallucinated_Solution*, *Creative_Solution*, and *Typical_Solution* samples, respectively. Length indicates the average token length of generated responses. Entropy, HESR, and Jaccard are reported as $\mu \pm \sigma$.

| Type | Model | Label (%) | | Length | Entropy | HESR | Jaccard |
|---|---|---|---|---|---|---|---|
| OC | QW-Base | ALL | 100.00 | 622.65 | 0.29 ± 0.06 | 0.50 ± 0.06 | 0.54 ± 0.09 |
| | | Hall | 22.70 | 735.72 | 0.29 ± 0.06 | 0.48 ± 0.06 | 0.54 ± 0.10 |
| | | Crea | 21.99 | 660.52 | 0.28 ± 0.04 | 0.51 ± 0.06 | 0.54 ± 0.11 |
| | | Typi | 55.32 | 561.22 | 0.29 ± 0.07 | 0.50 ± 0.06 | 0.54 ± 0.08 |
| | QW-Math | ALL | 100.00 | 686.34 | 0.24 ± 0.06 | 0.46 ± 0.06 | 0.57 ± 0.11 |
| | | Hall | 25.85 | 808.18 | 0.24 ± 0.06 | 0.45 ± 0.06 | 0.56 ± 0.11 |
| | | Crea | 21.09 | 699.52 | 0.23 ± 0.06 | 0.48 ± 0.06 | 0.57 ± 0.11 |
| | | Typi | 53.06 | 621.74 | 0.24 ± 0.06 | 0.46 ± 0.06 | 0.58 ± 0.10 |
| | QW-Inst | ALL | 100.00 | 632.24 | 0.33 ± 0.21 | 0.42 ± 0.08 | 1.00 ± 0.00 |
| | | Hall | 38.69 | 744.02 | 0.43 ± 0.22 | 0.45 ± 0.07 | 1.00 ± 0.00 |
| | | Crea | 16.87 | 623.00 | 0.29 ± 0.22 | 0.43 ± 0.07 | 1.00 ± 0.00 |
| | | Typi | 44.44 | 538.41 | 0.26 ± 0.17 | 0.40 ± 0.08 | 1.00 ± 0.00 |
| | QW-Distill | ALL | 100.00 | 752.72 | 0.14 ± 0.03 | 0.43 ± 0.06 | 0.52 ± 0.11 |
| | | Hall | 27.78 | 830.90 | 0.15 ± 0.02 | 0.44 ± 0.05 | 0.51 ± 0.11 |
| | | Crea | 22.22 | 875.25 | 0.12 ± 0.02 | 0.38 ± 0.04 | 0.46 ± 0.12 |
| | | Typi | 50.00 | 654.83 | 0.15 ± 0.03 | 0.45 ± 0.07 | 0.54 ± 0.10 |
| Non-OC | QW-Base | ALL | 100.00 | 643.96 | 0.46 ± 0.11 | 0.52 ± 0.06 | 0.56 ± 0.12 |
| | | Hall | 43.88 | 747.12 | 0.47 ± 0.11 | 0.53 ± 0.05 | 0.56 ± 0.12 |
| | | Crea | 15.18 | 620.43 | 0.45 ± 0.11 | 0.53 ± 0.06 | 0.56 ± 0.12 |
| | | Typi | 40.94 | 542.09 | 0.44 ± 0.11 | 0.51 ± 0.06 | 0.56 ± 0.12 |
| | QW-Math | ALL | 100.00 | 631.45 | 0.39 ± 0.10 | 0.50 ± 0.06 | 0.58 ± 0.12 |
| | | Hall | 45.76 | 738.27 | 0.38 ± 0.10 | 0.51 ± 0.06 | 0.57 ± 0.12 |
| | | Crea | 14.55 | 600.09 | 0.39 ± 0.11 | 0.51 ± 0.06 | 0.58 ± 0.12 |
| | | Typi | 39.69 | 519.82 | 0.40 ± 0.11 | 0.49 ± 0.06 | 0.59 ± 0.12 |
| | QW-Inst | ALL | 100.00 | 740.50 | 1.02 ± 0.35 | 0.44 ± 0.08 | 1.00 ± 0.00 |
| | | Hall | 74.64 | 759.80 | 1.04 ± 0.36 | 0.46 ± 0.07 | 1.00 ± 0.00 |
| | | Crea | 5.07 | 711.71 | 1.02 ± 0.37 | 0.40 ± 0.07 | 1.00 ± 0.00 |
| | | Typi | 20.29 | 676.71 | 0.95 ± 0.29 | 0.40 ± 0.09 | 1.00 ± 0.00 |
| | QW-Distill | ALL | 100.00 | 639.33 | 0.31 ± 0.11 | 0.50 ± 0.06 | 0.59 ± 0.11 |
| | | Hall | 42.27 | 745.21 | 0.34 ± 0.10 | 0.52 ± 0.06 | 0.58 ± 0.11 |
| | | Crea | 15.65 | 617.03 | 0.29 ± 0.11 | 0.50 ± 0.06 | 0.60 ± 0.12 |
| | | Typi | 42.07 | 541.23 | 0.28 ± 0.10 | 0.50 ± 0.06 | 0.59 ± 0.11 |

Table 14: Evaluation results of the QW family on responses generated by the QW-Inst model ($q = 5\%, \theta = 0.35$).

| Type | Model | Label (%) | | Length | Entropy | HESR | Jaccard |
|------|-------|------|------|--------|---------|------|---------|
| OC | QW-Base | ALL | 100.00 | 627.62 | 0.25 ± 0.05 | 0.49 ± 0.05 | 0.53 ± 0.09 |
| | | Hall | 14.55 | 746.50 | 0.24 ± 0.05 | 0.50 ± 0.03 | 0.57 ± 0.10 |
| | | Crea | 25.45 | 694.79 | 0.25 ± 0.04 | 0.50 ± 0.05 | 0.49 ± 0.10 |
| | | Typi | 60.00 | 570.30 | 0.25 ± 0.06 | 0.48 ± 0.05 | 0.54 ± 0.07 |
| | QW-Math | ALL | 100.00 | 701.62 | 0.21 ± 0.05 | 0.45 ± 0.06 | 0.57 ± 0.11 |
| | | Hall | 19.40 | 803.62 | 0.20 ± 0.04 | 0.43 ± 0.07 | 0.57 ± 0.11 |
| | | Crea | 26.87 | 750.92 | 0.21 ± 0.05 | 0.45 ± 0.06 | 0.56 ± 0.13 |
| | | Typi | 53.73 | 640.14 | 0.22 ± 0.05 | 0.45 ± 0.06 | 0.57 ± 0.10 |
| | QW-Inst | ALL | 100.00 | 623.93 | 0.30 ± 0.19 | 0.42 ± 0.08 | 1.00 ± 0.00 |
| | | Hall | 36.30 | 740.74 | 0.39 ± 0.19 | 0.45 ± 0.07 | 1.00 ± 0.00 |
| | | Crea | 17.02 | 622.48 | 0.27 ± 0.20 | 0.42 ± 0.07 | 1.00 ± 0.00 |
| | | Typi | 46.68 | 533.64 | 0.25 ± 0.15 | 0.40 ± 0.08 | 1.00 ± 0.00 |
| | QW-Distill | ALL | 100.00 | 776.96 | 0.13 ± 0.03 | 0.42 ± 0.06 | 0.48 ± 0.11 |
| | | Hall | 30.43 | 897.86 | 0.14 ± 0.01 | 0.43 ± 0.05 | 0.47 ± 0.11 |
| | | Crea | 30.43 | 860.71 | 0.11 ± 0.01 | 0.37 ± 0.04 | 0.44 ± 0.12 |
| | | Typi | 39.13 | 617.78 | 0.14 ± 0.03 | 0.44 ± 0.07 | 0.51 ± 0.10 |
| Non-OC | QW-Base | ALL | 100.00 | 642.53 | 0.45 ± 0.12 | 0.52 ± 0.06 | 0.56 ± 0.12 |
| | | Hall | 42.95 | 746.55 | 0.46 ± 0.12 | 0.53 ± 0.05 | 0.56 ± 0.12 |
| | | Crea | 15.45 | 621.32 | 0.44 ± 0.11 | 0.53 ± 0.06 | 0.56 ± 0.12 |
| | | Typi | 41.60 | 543.00 | 0.43 ± 0.11 | 0.51 ± 0.06 | 0.56 ± 0.11 |
| | QW-Math | ALL | 100.00 | 636.27 | 0.38 ± 0.11 | 0.50 ± 0.06 | 0.58 ± 0.12 |
| | | Hall | 44.09 | 744.14 | 0.37 ± 0.10 | 0.50 ± 0.06 | 0.57 ± 0.12 |
| | | Crea | 14.75 | 603.66 | 0.38 ± 0.11 | 0.51 ± 0.06 | 0.58 ± 0.12 |
| | | Typi | 41.16 | 532.40 | 0.38 ± 0.11 | 0.49 ± 0.06 | 0.59 ± 0.12 |
| | QW-Inst | ALL | 100.00 | 734.54 | 0.87 ± 0.33 | 0.44 ± 0.07 | 1.00 ± 0.00 |
| | | Hall | 70.80 | 761.81 | 0.91 ± 0.34 | 0.45 ± 0.07 | 1.00 ± 0.00 |
| | | Crea | 9.60 | 653.63 | 0.72 ± 0.32 | 0.44 ± 0.06 | 1.00 ± 0.00 |
| | | Typi | 19.60 | 675.63 | 0.83 ± 0.30 | 0.41 ± 0.09 | 1.00 ± 0.00 |
| | QW-Distill | ALL | 100.00 | 639.94 | 0.31 ± 0.11 | 0.50 ± 0.06 | 0.59 ± 0.11 |
| | | Hall | 42.11 | 744.88 | 0.34 ± 0.11 | 0.52 ± 0.06 | 0.58 ± 0.11 |
| | | Crea | 15.58 | 618.56 | 0.29 ± 0.11 | 0.50 ± 0.06 | 0.60 ± 0.12 |
| | | Typi | 42.31 | 543.36 | 0.28 ± 0.10 | 0.49 ± 0.06 | 0.59 ± 0.11 |

Table 15: Evaluation results of the QW family on responses generated by the QW-Inst model ($q = 5\%, \theta = 0.40$).

| Type | Model | Label (%) | | Length | Entropy | HESR | Jaccard |
|------|-------|------|------|--------|---------|------|---------|
| OC | QW-Base | ALL | 100.00 | 635.50 | 0.22 ± 0.04 | 0.48 ± 0.05 | 0.53 ± 0.09 |
| | | Hall | 15.38 | 701.75 | 0.21 ± 0.03 | 0.49 ± 0.05 | 0.55 ± 0.12 |
| | | Crea | 11.54 | 710.67 | 0.21 ± 0.02 | 0.49 ± 0.04 | 0.49 ± 0.12 |
| | | Typi | 73.08 | 609.68 | 0.22 ± 0.04 | 0.47 ± 0.05 | 0.53 ± 0.07 |
| | QW-Math | ALL | 100.00 | 724.48 | 0.19 ± 0.04 | 0.44 ± 0.07 | 0.57 ± 0.10 |
| | | Hall | 18.84 | 854.08 | 0.18 ± 0.04 | 0.40 ± 0.06 | 0.57 ± 0.09 |
| | | Crea | 27.54 | 743.26 | 0.18 ± 0.04 | 0.44 ± 0.07 | 0.55 ± 0.13 |
| | | Typi | 53.62 | 669.30 | 0.20 ± 0.04 | 0.45 ± 0.07 | 0.58 ± 0.09 |
| | QW-Inst | ALL | 100.00 | 614.28 | 0.27 ± 0.15 | 0.42 ± 0.08 | 1.00 ± 0.00 |
| | | Hall | 33.33 | 735.37 | 0.35 ± 0.17 | 0.44 ± 0.07 | 1.00 ± 0.00 |
| | | Crea | 17.49 | 614.36 | 0.24 ± 0.15 | 0.43 ± 0.07 | 1.00 ± 0.00 |
| | | Typi | 49.18 | 532.17 | 0.23 ± 0.12 | 0.40 ± 0.08 | 1.00 ± 0.00 |
| | QW-Distill | ALL | 100.00 | 847.67 | 0.11 ± 0.02 | 0.38 ± 0.04 | 0.43 ± 0.12 |
| | | Hall | 25.00 | 913.33 | 0.13 ± 0.01 | 0.42 ± 0.02 | 0.43 ± 0.14 |
| | | Crea | 50.00 | 919.83 | 0.11 ± 0.02 | 0.37 ± 0.04 | 0.44 ± 0.13 |
| | | Typi | 25.00 | 637.67 | 0.10 ± 0.01 | 0.36 ± 0.03 | 0.41 ± 0.08 |
| Non-OC | QW-Base | ALL | 100.00 | 642.11 | 0.44 ± 0.12 | 0.52 ± 0.06 | 0.56 ± 0.12 |
| | | Hall | 42.39 | 746.83 | 0.46 ± 0.12 | 0.53 ± 0.05 | 0.56 ± 0.12 |
| | | Crea | 15.88 | 624.50 | 0.43 ± 0.12 | 0.53 ± 0.06 | 0.56 ± 0.12 |
| | | Typi | 41.73 | 542.42 | 0.43 ± 0.11 | 0.51 ± 0.06 | 0.55 ± 0.11 |
| | QW-Math | ALL | 100.00 | 638.10 | 0.37 ± 0.11 | 0.50 ± 0.06 | 0.58 ± 0.12 |
| | | Hall | 43.02 | 744.33 | 0.37 ± 0.10 | 0.50 ± 0.06 | 0.57 ± 0.12 |
| | | Crea | 15.25 | 615.54 | 0.37 ± 0.11 | 0.51 ± 0.06 | 0.58 ± 0.12 |
| | | Typi | 41.72 | 536.82 | 0.37 ± 0.11 | 0.49 ± 0.06 | 0.58 ± 0.12 |
| | QW-Inst | ALL | 100.00 | 727.14 | 0.78 ± 0.33 | 0.44 ± 0.07 | 1.00 ± 0.00 |
| | | Hall | 68.35 | 763.30 | 0.81 ± 0.33 | 0.46 ± 0.07 | 1.00 ± 0.00 |
| | | Crea | 10.64 | 682.15 | 0.69 ± 0.32 | 0.42 ± 0.07 | 1.00 ± 0.00 |
| | | Typi | 21.01 | 632.29 | 0.72 ± 0.30 | 0.41 ± 0.08 | 1.00 ± 0.00 |
| | QW-Distill | ALL | 100.00 | 640.37 | 0.30 ± 0.11 | 0.50 ± 0.06 | 0.59 ± 0.11 |
| | | Hall | 42.07 | 745.77 | 0.34 ± 0.11 | 0.51 ± 0.06 | 0.58 ± 0.11 |
| | | Crea | 15.54 | 618.08 | 0.29 ± 0.11 | 0.50 ± 0.06 | 0.60 ± 0.12 |
| | | Typi | 42.40 | 543.96 | 0.28 ± 0.10 | 0.49 ± 0.06 | 0.59 ± 0.11 |

Table 16: Evaluation results of the QW family on responses generated by the QW-Inst model ($q = 3\%, \theta = 0.35$).

| Type | Model | Label (%) | | Length | Entropy | HESR | Jaccard |
|---|---|---|---|---|---|---|---|
| OC | QW-Base | ALL | 100.00 | 618 | 0.20 ± 0.02 | 0.47 ± 0.03 | 0.58 ± 0.10 |
| | | Hall | 20.00 | 666 | 0.22 ± 0.00 | 0.51 ± 0.00 | 0.74 ± 0.00 |
| | | Crea | - | - | - | - | - |
| | | Typi | 80.00 | 606 | 0.19 ± 0.02 | 0.46 ± 0.02 | 0.54 ± 0.07 |
| | QW-Math | ALL | 100.00 | 620.83 | 0.16 ± 0.03 | 0.44 ± 0.05 | 0.55 ± 0.12 |
| | | Hall | - | - | - | - | - |
| | | Crea | 50.00 | 682.83 | 0.14 ± 0.02 | 0.42 ± 0.04 | 0.57 ± 0.17 |
| | | Typi | 50.00 | 558.83 | 0.19 ± 0.03 | 0.46 ± 0.05 | 0.53 ± 0.05 |
| | QW-Inst | ALL | 100.00 | 597.55 | 0.22 ± 0.11 | 0.42 ± 0.07 | 1.00 ± 0.00 |
| | | Hall | 25.38 | 737.69 | 0.27 ± 0.13 | 0.44 ± 0.06 | 1.00 ± 0.00 |
| | | Crea | 19.51 | 610.63 | 0.20 ± 0.10 | 0.42 ± 0.07 | 1.00 ± 0.00 |
| | | Typi | 55.11 | 528.37 | 0.20 ± 0.09 | 0.40 ± 0.07 | 1.00 ± 0.00 |
| | QW-Distill | ALL | - | - | - | - | - |
| | | Hall | - | - | - | - | - |
| | | Crea | - | - | - | - | - |
| | | Typi | - | - | - | - | - |
| Non-OC | QW-Base | ALL | 100.00 | 642.07 | 0.44 ± 0.12 | 0.52 ± 0.06 | 0.55 ± 0.12 |
| | | Hall | 42.01 | 746.67 | 0.46 ± 0.12 | 0.53 ± 0.05 | 0.56 ± 0.12 |
| | | Crea | 15.86 | 625.57 | 0.43 ± 0.12 | 0.53 ± 0.06 | 0.55 ± 0.12 |
| | | Typi | 42.14 | 544.01 | 0.43 ± 0.12 | 0.51 ± 0.06 | 0.55 ± 0.11 |
| | QW-Math | ALL | 100.00 | 642.16 | 0.36 ± 0.11 | 0.49 ± 0.06 | 0.58 ± 0.12 |
| | | Hall | 42.26 | 746.55 | 0.37 ± 0.11 | 0.50 ± 0.06 | 0.57 ± 0.12 |
| | | Crea | 15.54 | 624.11 | 0.36 ± 0.12 | 0.50 ± 0.06 | 0.58 ± 0.12 |
| | | Typi | 42.20 | 544.26 | 0.36 ± 0.12 | 0.49 ± 0.06 | 0.59 ± 0.12 |
| | QW-Inst | ALL | 100.00 | 697.62 | 0.62 ± 0.32 | 0.44 ± 0.08 | 1.00 ± 0.00 |
| | | Hall | 62.65 | 751.04 | 0.66 ± 0.33 | 0.45 ± 0.07 | 1.00 ± 0.00 |
| | | Crea | 11.18 | 658.18 | 0.56 ± 0.31 | 0.43 ± 0.07 | 1.00 ± 0.00 |
| | | Typi | 26.18 | 586.60 | 0.53 ± 0.29 | 0.41 ± 0.09 | 1.00 ± 0.00 |
| | QW-Distill | ALL | 100.00 | 641.99 | 0.30 ± 0.11 | 0.50 ± 0.06 | 0.58 ± 0.11 |
| | | Hall | 41.93 | 746.55 | 0.33 ± 0.11 | 0.51 ± 0.06 | 0.58 ± 0.11 |
| | | Crea | 15.81 | 625.57 | 0.28 ± 0.11 | 0.50 ± 0.06 | 0.59 ± 0.12 |
| | | Typi | 42.26 | 544.39 | 0.28 ± 0.10 | 0.49 ± 0.06 | 0.59 ± 0.11 |

Table 17: Evaluation results of the QW family on responses generated by the QW-Inst model ($q = 7\%, \theta = 0.35$).

| Type | Model | Label (%) | | Length | Entropy | HESR | Jaccard |
|---|---|---|---|---|---|---|---|
| OC | QW-Base | ALL | 100.00 | 650.15 | 0.31 ± 0.06 | 0.50 ± 0.06 | 0.54 ± 0.10 |
| | | Hall | 26.95 | 790.22 | 0.31 ± 0.05 | 0.49 ± 0.06 | 0.54 ± 0.11 |
| | | Crea | 17.38 | 655.61 | 0.30 ± 0.06 | 0.52 ± 0.06 | 0.54 ± 0.10 |
| | | Typi | 55.67 | 580.64 | 0.31 ± 0.06 | 0.50 ± 0.06 | 0.54 ± 0.09 |
| | QW-Math | ALL | 100.00 | 680.34 | 0.27 ± 0.06 | 0.47 ± 0.06 | 0.57 ± 0.11 |
| | | Hall | 31.26 | 787.98 | 0.27 ± 0.06 | 0.47 ± 0.06 | 0.56 ± 0.12 |
| | | Crea | 19.38 | 686.12 | 0.26 ± 0.06 | 0.48 ± 0.06 | 0.56 ± 0.11 |
| | | Typi | 49.36 | 609.90 | 0.27 ± 0.06 | 0.47 ± 0.06 | 0.58 ± 0.11 |
| | QW-Inst | ALL | 100.00 | 636.00 | 0.35 ± 0.23 | 0.42 ± 0.08 | 1.00 ± 0.00 |
| | | Hall | 40.11 | 744.73 | 0.46 ± 0.24 | 0.45 ± 0.07 | 1.00 ± 0.00 |
| | | Crea | 16.53 | 623.31 | 0.30 ± 0.22 | 0.43 ± 0.07 | 1.00 ± 0.00 |
| | | Typi | 43.36 | 540.26 | 0.27 ± 0.18 | 0.40 ± 0.08 | 1.00 ± 0.00 |
| | QW-Distill | ALL | 100.00 | 688.36 | 0.17 ± 0.03 | 0.46 ± 0.06 | 0.55 ± 0.11 |
| | | Hall | 24.03 | 816.22 | 0.17 ± 0.03 | 0.45 ± 0.05 | 0.54 ± 0.13 |
| | | Crea | 18.83 | 761.52 | 0.16 ± 0.03 | 0.45 ± 0.06 | 0.56 ± 0.11 |
| | | Typi | 57.14 | 610.50 | 0.17 ± 0.03 | 0.47 ± 0.07 | 0.56 ± 0.10 |
| Non-OC | QW-Base | ALL | 100.00 | 640.15 | 0.47 ± 0.11 | 0.53 ± 0.06 | 0.56 ± 0.12 |
| | | Hall | 45.32 | 740.68 | 0.48 ± 0.11 | 0.54 ± 0.05 | 0.56 ± 0.12 |
| | | Crea | 15.45 | 617.94 | 0.46 ± 0.11 | 0.53 ± 0.06 | 0.56 ± 0.13 |
| | | Typi | 39.23 | 532.78 | 0.46 ± 0.11 | 0.51 ± 0.06 | 0.56 ± 0.12 |
| | QW-Math | ALL | 100.00 | 620.68 | 0.41 ± 0.10 | 0.51 ± 0.06 | 0.58 ± 0.12 |
| | | Hall | 47.87 | 731.51 | 0.40 ± 0.09 | 0.51 ± 0.05 | 0.57 ± 0.12 |
| | | Crea | 13.82 | 578.37 | 0.42 ± 0.10 | 0.52 ± 0.06 | 0.59 ± 0.13 |
| | | Typi | 38.31 | 497.47 | 0.43 ± 0.11 | 0.50 ± 0.06 | 0.59 ± 0.12 |
| | QW-Inst | ALL | 100.00 | 743.98 | 1.17 ± 0.33 | 0.44 ± 0.08 | 1.00 ± 0.00 |
| | | Hall | 72.94 | 763.58 | 1.21 ± 0.33 | 0.46 ± 0.08 | 1.00 ± 0.00 |
| | | Crea | 3.53 | 805.00 | 1.39 ± 0.18 | 0.37 ± 0.08 | 1.00 ± 0.00 |
| | | Typi | 23.53 | 674.05 | 1.03 ± 0.27 | 0.41 ± 0.09 | 1.00 ± 0.00 |
| | QW-Distill | ALL | 100.00 | 636.81 | 0.32 ± 0.10 | 0.51 ± 0.06 | 0.59 ± 0.11 |
| | | Hall | 43.94 | 742.29 | 0.34 ± 0.10 | 0.52 ± 0.06 | 0.58 ± 0.11 |
| | | Crea | 15.47 | 607.06 | 0.30 ± 0.11 | 0.50 ± 0.06 | 0.60 ± 0.12 |
| | | Typi | 40.60 | 533.99 | 0.30 ± 0.10 | 0.50 ± 0.06 | 0.59 ± 0.11 |

Table 18: Evaluation results on responses generated by the DS-Math model, categorized into two response types—Overconfidence (OC) and Non-Overconfidence (Non-OC)—across four DS family variants. The hyperparameters for determining response types are set to $q = 5\%$ and $\theta = 0.30$, as defined in Equation 2. Model denotes the evaluation model. ALL, Hall, Crea, and Typi refer to the full dataset, *Hallucinated_Solution*, *Creative_Solution*, and *Typical_Solution* samples, respectively. Length indicates the average token length of generated responses. Entropy, HESR, and Jaccard are reported as $\mu \pm \sigma$.

| Type | Model | Label (%) | | Length | Entropy | HESR | Jaccard |
|------|-------|------|--------|--------|---------|------|---------|
| OC | DS-Base | ALL | 100.00 | 621.83 | 0.21 ± 0.08 | 0.23 ± 0.08 | 0.88 ± 0.10 |
| | | Hall | 41.30 | 611.89 | 0.23 ± 0.08 | 0.24 ± 0.08 | 0.89 ± 0.09 |
| | | Crea | 19.13 | 613.60 | 0.22 ± 0.07 | 0.23 ± 0.06 | 0.92 ± 0.09 |
| | | Typi | 39.57 | 636.03 | 0.19 ± 0.08 | 0.22 ± 0.08 | 0.85 ± 0.11 |
| | DS-Math | ALL | 100.00 | 617.96 | 0.22 ± 0.09 | 0.25 ± 0.09 | 1.00 ± 0.00 |
| | | Hall | 48.37 | 623.68 | 0.24 ± 0.09 | 0.27 ± 0.10 | 1.00 ± 0.00 |
| | | Crea | 18.63 | 607.34 | 0.23 ± 0.07 | 0.24 ± 0.07 | 1.00 ± 0.00 |
| | | Typi | 33.01 | 617.26 | 0.20 ± 0.09 | 0.24 ± 0.09 | 1.00 ± 0.00 |
| | DS-Inst | ALL | 100.00 | 623.86 | 0.13 ± 0.05 | 0.21 ± 0.08 | 0.81 ± 0.13 |
| | | Hall | 40.21 | 617.75 | 0.14 ± 0.06 | 0.22 ± 0.08 | 0.81 ± 0.14 |
| | | Crea | 21.69 | 591.74 | 0.13 ± 0.04 | 0.23 ± 0.08 | 0.86 ± 0.10 |
| | | Typi | 38.10 | 646.89 | 0.11 ± 0.05 | 0.20 ± 0.09 | 0.78 ± 0.12 |
| | DS-RL | ALL | 100.00 | 627.77 | 0.12 ± 0.05 | 0.22 ± 0.08 | 0.79 ± 0.13 |
| | | Hall | 36.84 | 621.67 | 0.13 ± 0.05 | 0.22 ± 0.08 | 0.79 ± 0.13 |
| | | Crea | 21.05 | 591.75 | 0.13 ± 0.04 | 0.24 ± 0.07 | 0.82 ± 0.09 |
| | | Typi | 42.11 | 652.57 | 0.10 ± 0.04 | 0.20 ± 0.08 | 0.77 ± 0.13 |
| Non-OC | DS-Base | ALL | 100.00 | 520.10 | 0.81 ± 0.35 | 0.46 ± 0.10 | 0.91 ± 0.06 |
| | | Hall | 68.44 | 529.72 | 0.83 ± 0.35 | 0.47 ± 0.10 | 0.90 ± 0.06 |
| | | Crea | 16.33 | 498.58 | 0.84 ± 0.39 | 0.45 ± 0.11 | 0.92 ± 0.06 |
| | | Typi | 15.23 | 500.89 | 0.69 ± 0.29 | 0.42 ± 0.10 | 0.91 ± 0.07 |
| | DS-Math | ALL | 100.00 | 516.34 | 0.78 ± 0.33 | 0.46 ± 0.10 | 1.00 ± 0.00 |
| | | Hall | 68.36 | 525.47 | 0.79 ± 0.33 | 0.47 ± 0.10 | 1.00 ± 0.00 |
| | | Crea | 16.28 | 494.45 | 0.82 ± 0.36 | 0.45 ± 0.10 | 1.00 ± 0.00 |
| | | Typi | 15.37 | 498.56 | 0.67 ± 0.27 | 0.42 ± 0.10 | 1.00 ± 0.00 |
| | DS-Inst | ALL | 100.00 | 521.97 | 0.51 ± 0.23 | 0.46 ± 0.10 | 0.85 ± 0.08 |
| | | Hall | 67.75 | 530.51 | 0.53 ± 0.23 | 0.48 ± 0.10 | 0.85 ± 0.08 |
| | | Crea | 16.05 | 504.72 | 0.52 ± 0.25 | 0.45 ± 0.11 | 0.86 ± 0.08 |
| | | Typi | 16.20 | 503.81 | 0.42 ± 0.19 | 0.42 ± 0.11 | 0.86 ± 0.08 |
| | DS-RL | ALL | 100.00 | 522.59 | 0.47 ± 0.22 | 0.46 ± 0.10 | 0.83 ± 0.09 |
| | | Hall | 67.81 | 530.84 | 0.48 ± 0.22 | 0.47 ± 0.10 | 0.83 ± 0.08 |
| | | Crea | 16.21 | 505.91 | 0.49 ± 0.24 | 0.45 ± 0.11 | 0.83 ± 0.09 |
| | | Typi | 15.98 | 505.47 | 0.39 ± 0.18 | 0.42 ± 0.11 | 0.84 ± 0.09 |

Table 19: Evaluation results of the DS family on responses generated by the DS-Math model ($q = 5\%, \theta = 0.35$).

| Type | Model | Label (%) | | Length | Entropy | HESR | Jaccard |
|------|-------|------|------|--------|---------|------|---------|
| OC | DS-Base | ALL | 100.00 | 615.36 | 0.19 ± 0.07 | 0.22 ± 0.07 | 0.87 ± 0.11 |
| | | Hall | 34.90 | 599.65 | 0.20 ± 0.07 | 0.22 ± 0.07 | 0.89 ± 0.09 |
| | | Crea | 22.15 | 604.82 | 0.21 ± 0.06 | 0.22 ± 0.06 | 0.91 ± 0.09 |
| | | Typi | 42.95 | 633.56 | 0.18 ± 0.07 | 0.21 ± 0.08 | 0.84 ± 0.11 |
| | DS-Math | ALL | 100.00 | 624.95 | 0.20 ± 0.08 | 0.24 ± 0.09 | 1.00 ± 0.00 |
| | | Hall | 39.39 | 622.14 | 0.22 ± 0.08 | 0.25 ± 0.10 | 1.00 ± 0.00 |
| | | Crea | 21.72 | 603.07 | 0.21 ± 0.06 | 0.23 ± 0.06 | 1.00 ± 0.00 |
| | | Typi | 38.89 | 640.03 | 0.18 ± 0.08 | 0.22 ± 0.09 | 1.00 ± 0.00 |
| | DS-Inst | ALL | 100.00 | 630.97 | 0.12 ± 0.05 | 0.20 ± 0.08 | 0.80 ± 0.13 |
| | | Hall | 36.23 | 624.96 | 0.13 ± 0.05 | 0.21 ± 0.08 | 0.79 ± 0.14 |
| | | Crea | 21.74 | 593.80 | 0.13 ± 0.04 | 0.21 ± 0.07 | 0.85 ± 0.10 |
| | | Typi | 42.03 | 655.38 | 0.10 ± 0.04 | 0.19 ± 0.08 | 0.78 ± 0.13 |
| | DS-RL | ALL | 100.00 | 629.32 | 0.11 ± 0.04 | 0.20 ± 0.07 | 0.78 ± 0.12 |
| | | Hall | 34.17 | 622.44 | 0.11 ± 0.05 | 0.21 ± 0.07 | 0.78 ± 0.12 |
| | | Crea | 22.50 | 585.85 | 0.12 ± 0.04 | 0.22 ± 0.06 | 0.83 ± 0.08 |
| | | Typi | 43.33 | 657.31 | 0.09 ± 0.04 | 0.18 ± 0.07 | 0.76 ± 0.13 |
| Non-OC | DS-Base | ALL | 100.00 | 523.80 | 0.79 ± 0.36 | 0.46 ± 0.10 | 0.90 ± 0.07 |
| | | Hall | 67.52 | 532.38 | 0.81 ± 0.36 | 0.47 ± 0.10 | 0.90 ± 0.07 |
| | | Crea | 16.16 | 503.55 | 0.82 ± 0.40 | 0.44 ± 0.11 | 0.92 ± 0.06 |
| | | Typi | 16.31 | 508.30 | 0.67 ± 0.29 | 0.41 ± 0.10 | 0.91 ± 0.07 |
| | DS-Math | ALL | 100.00 | 518.93 | 0.76 ± 0.33 | 0.46 ± 0.10 | 1.00 ± 0.00 |
| | | Hall | 68.06 | 528.46 | 0.78 ± 0.33 | 0.47 ± 0.10 | 1.00 ± 0.00 |
| | | Crea | 16.01 | 499.09 | 0.80 ± 0.36 | 0.45 ± 0.11 | 1.00 ± 0.00 |
| | | Typi | 15.93 | 498.12 | 0.65 ± 0.28 | 0.41 ± 0.10 | 1.00 ± 0.00 |
| | DS-Inst | ALL | 100.00 | 522.96 | 0.50 ± 0.24 | 0.46 ± 0.11 | 0.85 ± 0.08 |
| | | Hall | 67.13 | 531.16 | 0.52 ± 0.24 | 0.47 ± 0.10 | 0.85 ± 0.08 |
| | | Crea | 16.25 | 506.40 | 0.52 ± 0.26 | 0.45 ± 0.11 | 0.86 ± 0.08 |
| | | Typi | 16.62 | 506.04 | 0.42 ± 0.19 | 0.41 ± 0.11 | 0.86 ± 0.08 |
| | DS-RL | ALL | 100.00 | 524.50 | 0.46 ± 0.22 | 0.46 ± 0.11 | 0.83 ± 0.09 |
| | | Hall | 66.91 | 532.18 | 0.48 ± 0.22 | 0.47 ± 0.10 | 0.83 ± 0.09 |
| | | Crea | 16.26 | 508.51 | 0.48 ± 0.24 | 0.45 ± 0.11 | 0.83 ± 0.09 |
| | | Typi | 16.83 | 509.44 | 0.38 ± 0.18 | 0.42 ± 0.11 | 0.84 ± 0.09 |

Table 20: Evaluation results of the DS family on responses generated by the DS-Math model ($q = 5\%, \theta = 0.40$).

| Type | Model | Label (%) | | Length | Entropy | HESR | Jaccard |
|------|-------|------|------|--------|---------|------|---------|
| OC | DS-Base | ALL | 100.00 | 627.69 | 0.18 ± 0.07 | 0.21 ± 0.07 | 0.86 ± 0.11 |
| | | Hall | 34.38 | 633.70 | 0.18 ± 0.07 | 0.22 ± 0.07 | 0.88 ± 0.10 |
| | | Crea | 21.09 | 606.07 | 0.19 ± 0.06 | 0.21 ± 0.06 | 0.90 ± 0.10 |
| | | Typi | 44.53 | 633.28 | 0.16 ± 0.07 | 0.19 ± 0.08 | 0.83 ± 0.12 |
| | DS-Math | ALL | 100.00 | 627.96 | 0.18 ± 0.07 | 0.22 ± 0.08 | 1.00 ± 0.00 |
| | | Hall | 35.19 | 619.72 | 0.19 ± 0.07 | 0.23 ± 0.08 | 1.00 ± 0.00 |
| | | Crea | 23.46 | 628.76 | 0.20 ± 0.06 | 0.23 ± 0.07 | 1.00 ± 0.00 |
| | | Typi | 41.36 | 634.52 | 0.17 ± 0.07 | 0.21 ± 0.08 | 1.00 ± 0.00 |
| | DS-Inst | ALL | 100.00 | 628.37 | 0.11 ± 0.04 | 0.19 ± 0.07 | 0.79 ± 0.13 |
| | | Hall | 33.88 | 614.46 | 0.12 ± 0.05 | 0.20 ± 0.07 | 0.77 ± 0.14 |
| | | Crea | 22.31 | 597.52 | 0.12 ± 0.03 | 0.21 ± 0.07 | 0.85 ± 0.10 |
| | | Typi | 43.80 | 654.85 | 0.10 ± 0.04 | 0.17 ± 0.07 | 0.77 ± 0.13 |
| | DS-RL | ALL | 100.00 | 635.04 | 0.10 ± 0.04 | 0.19 ± 0.07 | 0.78 ± 0.13 |
| | | Hall | 33.33 | 629.61 | 0.11 ± 0.05 | 0.20 ± 0.07 | 0.78 ± 0.13 |
| | | Crea | 22.22 | 592.83 | 0.11 ± 0.03 | 0.22 ± 0.07 | 0.82 ± 0.09 |
| | | Typi | 44.44 | 660.21 | 0.09 ± 0.03 | 0.17 ± 0.06 | 0.76 ± 0.14 |
| Non-OC | DS-Base | ALL | 100.00 | 524.04 | 0.78 ± 0.36 | 0.45 ± 0.11 | 0.91 ± 0.07 |
| | | Hall | 67.08 | 531.35 | 0.81 ± 0.36 | 0.47 ± 0.10 | 0.90 ± 0.07 |
| | | Crea | 16.35 | 506.09 | 0.81 ± 0.40 | 0.44 ± 0.12 | 0.92 ± 0.06 |
| | | Typi | 16.57 | 512.20 | 0.66 ± 0.30 | 0.41 ± 0.10 | 0.91 ± 0.07 |
| | DS-Math | ALL | 100.00 | 521.40 | 0.75 ± 0.34 | 0.45 ± 0.10 | 1.00 ± 0.00 |
| | | Hall | 67.80 | 530.76 | 0.77 ± 0.34 | 0.47 ± 0.10 | 1.00 ± 0.00 |
| | | Crea | 15.95 | 496.97 | 0.79 ± 0.37 | 0.44 ± 0.11 | 1.00 ± 0.00 |
| | | Typi | 16.25 | 506.29 | 0.63 ± 0.28 | 0.41 ± 0.10 | 1.00 ± 0.00 |
| | DS-Inst | ALL | 100.00 | 524.51 | 0.50 ± 0.24 | 0.46 ± 0.11 | 0.85 ± 0.08 |
| | | Hall | 66.95 | 532.53 | 0.52 ± 0.24 | 0.47 ± 0.10 | 0.85 ± 0.08 |
| | | Crea | 16.27 | 507.12 | 0.51 ± 0.26 | 0.44 ± 0.11 | 0.86 ± 0.08 |
| | | Typi | 16.77 | 509.37 | 0.41 ± 0.19 | 0.41 ± 0.11 | 0.86 ± 0.09 |
| | DS-RL | ALL | 100.00 | 524.96 | 0.46 ± 0.22 | 0.46 ± 0.11 | 0.83 ± 0.09 |
| | | Hall | 66.69 | 532.38 | 0.47 ± 0.22 | 0.47 ± 0.10 | 0.82 ± 0.09 |
| | | Crea | 16.33 | 508.79 | 0.47 ± 0.24 | 0.45 ± 0.11 | 0.83 ± 0.09 |
| | | Typi | 16.98 | 511.34 | 0.38 ± 0.18 | 0.41 ± 0.11 | 0.84 ± 0.09 |

Table 21: Evaluation results of the DS family on responses generated by the DS-Math model ($q = 3\%, \theta = 0.35$).

| Type | Model | Label (%) | | Length | Entropy | HESR | Jaccard |
|------|-------|------|------|--------|---------|------|---------|
| OC | DS-Base | ALL | 100.00 | 639.48 | 0.12 ± 0.04 | 0.18 ± 0.06 | 0.83 ± 0.12 |
| | | Hall | 32.81 | 617.29 | 0.13 ± 0.05 | 0.19 ± 0.05 | 0.84 ± 0.10 |
| | | Crea | 20.31 | 592.62 | 0.15 ± 0.04 | 0.21 ± 0.05 | 0.90 ± 0.10 |
| | | Typi | 46.88 | 675.33 | 0.11 ± 0.03 | 0.15 ± 0.06 | 0.80 ± 0.12 |
| | DS-Math | ALL | 100.00 | 622.16 | 0.13 ± 0.05 | 0.19 ± 0.06 | 1.00 ± 0.00 |
| | | Hall | 33.75 | 614.52 | 0.13 ± 0.05 | 0.20 ± 0.06 | 1.00 ± 0.00 |
| | | Crea | 17.50 | 605.93 | 0.14 ± 0.04 | 0.20 ± 0.06 | 1.00 ± 0.00 |
| | | Typi | 48.75 | 633.28 | 0.12 ± 0.04 | 0.18 ± 0.07 | 1.00 ± 0.00 |
| | DS-Inst | ALL | 100.00 | 643.88 | 0.08 ± 0.03 | 0.16 ± 0.06 | 0.76 ± 0.12 |
| | | Hall | 33.90 | 612.90 | 0.08 ± 0.03 | 0.16 ± 0.05 | 0.75 ± 0.11 |
| | | Crea | 16.95 | 566.00 | 0.09 ± 0.02 | 0.20 ± 0.06 | 0.83 ± 0.09 |
| | | Typi | 49.15 | 692.10 | 0.07 ± 0.02 | 0.14 ± 0.05 | 0.75 ± 0.13 |
| | DS-RL | ALL | 100.00 | 651.73 | 0.07 ± 0.02 | 0.17 ± 0.06 | 0.74 ± 0.13 |
| | | Hall | 31.37 | 614.75 | 0.07 ± 0.03 | 0.17 ± 0.05 | 0.75 ± 0.11 |
| | | Crea | 15.69 | 589.50 | 0.08 ± 0.02 | 0.21 ± 0.06 | 0.80 ± 0.07 |
| | | Typi | 52.94 | 692.07 | 0.06 ± 0.02 | 0.15 ± 0.06 | 0.72 ± 0.14 |
| Non-OC | DS-Base | ALL | 100.00 | 528.11 | 0.76 ± 0.37 | 0.44 ± 0.11 | 0.90 ± 0.07 |
| | | Hall | 65.70 | 534.19 | 0.79 ± 0.36 | 0.46 ± 0.11 | 0.90 ± 0.07 |
| | | Crea | 16.60 | 512.65 | 0.78 ± 0.41 | 0.43 ± 0.12 | 0.92 ± 0.06 |
| | | Typi | 17.70 | 520.04 | 0.61 ± 0.31 | 0.39 ± 0.11 | 0.91 ± 0.07 |
| | DS-Math | ALL | 100.00 | 527.83 | 0.72 ± 0.35 | 0.44 ± 0.11 | 1.00 ± 0.00 |
| | | Hall | 66.01 | 533.74 | 0.75 ± 0.35 | 0.46 ± 0.11 | 1.00 ± 0.00 |
| | | Crea | 16.71 | 511.54 | 0.73 ± 0.39 | 0.42 ± 0.12 | 1.00 ± 0.00 |
| | | Typi | 17.27 | 521.02 | 0.59 ± 0.29 | 0.39 ± 0.11 | 1.00 ± 0.00 |
| | DS-Inst | ALL | 100.00 | 528.32 | 0.48 ± 0.24 | 0.45 ± 0.12 | 0.85 ± 0.08 |
| | | Hall | 65.54 | 534.37 | 0.51 ± 0.24 | 0.47 ± 0.11 | 0.85 ± 0.08 |
| | | Crea | 16.75 | 514.74 | 0.49 ± 0.27 | 0.43 ± 0.12 | 0.86 ± 0.08 |
| | | Typi | 17.71 | 518.75 | 0.38 ± 0.20 | 0.39 ± 0.12 | 0.85 ± 0.09 |
| | DS-RL | ALL | 100.00 | 528.67 | 0.44 ± 0.23 | 0.45 ± 0.12 | 0.83 ± 0.09 |
| | | Hall | 65.46 | 534.67 | 0.47 ± 0.22 | 0.47 ± 0.11 | 0.82 ± 0.09 |
| | | Crea | 16.79 | 514.39 | 0.45 ± 0.25 | 0.43 ± 0.12 | 0.83 ± 0.09 |
| | | Typi | 17.75 | 520.09 | 0.35 ± 0.19 | 0.40 ± 0.12 | 0.84 ± 0.10 |

Table 22: Evaluation results of the DS family on responses generated by the DS-Math model ($q = 7\%, \theta = 0.35$).

| Type | Model | Label (%) | | Length | Entropy | HESR | Jaccard |
|------|-------|------|------|--------|---------|------|---------|
| OC | DS-Base | ALL | 100.00 | 610.95 | 0.25 ± 0.09 | 0.25 ± 0.09 | 0.89 ± 0.10 |
| | | Hall | 41.98 | 609.88 | 0.27 ± 0.09 | 0.28 ± 0.10 | 0.89 ± 0.09 |
| | | Crea | 19.85 | 609.35 | 0.24 ± 0.07 | 0.24 ± 0.06 | 0.92 ± 0.08 |
| | | Typi | 38.17 | 612.96 | 0.23 ± 0.10 | 0.24 ± 0.09 | 0.86 ± 0.11 |
| | DS-Math | ALL | 100.00 | 610.31 | 0.26 ± 0.10 | 0.27 ± 0.10 | 1.00 ± 0.00 |
| | | Hall | 47.26 | 620.88 | 0.28 ± 0.09 | 0.30 ± 0.10 | 1.00 ± 0.00 |
| | | Crea | 18.90 | 595.45 | 0.25 ± 0.08 | 0.25 ± 0.08 | 1.00 ± 0.00 |
| | | Typi | 33.84 | 603.84 | 0.23 ± 0.10 | 0.25 ± 0.09 | 1.00 ± 0.00 |
| | DS-Inst | ALL | 100.00 | 623.13 | 0.15 ± 0.06 | 0.24 ± 0.09 | 0.83 ± 0.12 |
| | | Hall | 42.68 | 619.29 | 0.17 ± 0.06 | 0.26 ± 0.10 | 0.83 ± 0.13 |
| | | Crea | 20.73 | 613.25 | 0.15 ± 0.05 | 0.24 ± 0.08 | 0.87 ± 0.09 |
| | | Typi | 36.59 | 633.20 | 0.14 ± 0.06 | 0.22 ± 0.09 | 0.80 ± 0.12 |
| | DS-RL | ALL | 100.00 | 618.48 | 0.14 ± 0.05 | 0.24 ± 0.09 | 0.81 ± 0.12 |
| | | Hall | 39.38 | 613.78 | 0.15 ± 0.06 | 0.25 ± 0.09 | 0.81 ± 0.13 |
| | | Crea | 21.68 | 618.53 | 0.14 ± 0.04 | 0.25 ± 0.07 | 0.84 ± 0.08 |
| | | Typi | 38.94 | 623.22 | 0.13 ± 0.05 | 0.23 ± 0.09 | 0.79 ± 0.12 |
| Non-OC | DS-Base | ALL | 100.00 | 516.43 | 0.83 ± 0.34 | 0.47 ± 0.09 | 0.90 ± 0.06 |
| | | Hall | 68.99 | 526.55 | 0.85 ± 0.34 | 0.48 ± 0.09 | 0.90 ± 0.06 |
| | | Crea | 16.11 | 492.81 | 0.87 ± 0.38 | 0.46 ± 0.10 | 0.92 ± 0.06 |
| | | Typi | 14.90 | 495.13 | 0.74 ± 0.27 | 0.44 ± 0.09 | 0.91 ± 0.07 |
| | DS-Math | ALL | 100.00 | 511.33 | 0.81 ± 0.32 | 0.47 ± 0.09 | 1.00 ± 0.00 |
| | | Hall | 69.04 | 519.86 | 0.82 ± 0.32 | 0.48 ± 0.09 | 1.00 ± 0.00 |
| | | Crea | 16.16 | 491.22 | 0.85 ± 0.35 | 0.46 ± 0.10 | 1.00 ± 0.00 |
| | | Typi | 14.81 | 493.51 | 0.71 ± 0.26 | 0.44 ± 0.09 | 1.00 ± 0.00 |
| | DS-Inst | ALL | 100.00 | 515.26 | 0.53 ± 0.23 | 0.47 ± 0.10 | 0.85 ± 0.08 |
| | | Hall | 68.51 | 525.89 | 0.54 ± 0.23 | 0.49 ± 0.09 | 0.85 ± 0.08 |
| | | Crea | 15.98 | 492.40 | 0.55 ± 0.25 | 0.46 ± 0.10 | 0.86 ± 0.08 |
| | | Typi | 15.51 | 491.85 | 0.45 ± 0.18 | 0.44 ± 0.10 | 0.86 ± 0.08 |
| | DS-RL | ALL | 100.00 | 517.75 | 0.48 ± 0.21 | 0.47 ± 0.10 | 0.83 ± 0.09 |
| | | Hall | 68.69 | 528.14 | 0.49 ± 0.21 | 0.48 ± 0.09 | 0.82 ± 0.08 |
| | | Crea | 15.89 | 492.32 | 0.51 ± 0.23 | 0.47 ± 0.10 | 0.83 ± 0.09 |
| | | Typi | 15.42 | 497.71 | 0.42 ± 0.17 | 0.44 ± 0.10 | 0.84 ± 0.09 |

Table 23: Evaluation results on responses generated by the DS-RL model, categorized into two response types—Overconfidence (OC) and Non-Overconfidence (Non-OC)—across four DS family variants. The hyperparameters for determining response types are set to $q = 5\%$ and $\theta = 0.30$, as defined in Equation 2. Model denotes the evaluation model. ALL, Hall, Crea, and Typi refer to the full dataset, *Hallucinated_Solution*, *Creative_Solution*, and *Typical_Solution* samples, respectively. Length indicates the average token length of generated responses. Entropy, HESR, and Jaccard are reported as $\mu \pm \sigma$.

| Type | Model | Label (%) | | Length | Entropy | HESR | Jaccard |
|------|-------|------|------|--------|---------|------|---------|
| OC | DS-Base | ALL | 100.00 | 622.21 | 0.24 ± 0.07 | 0.24 ± 0.07 | 0.84 ± 0.13 |
| | | Hall | 35.71 | 705.00 | 0.23 ± 0.04 | 0.26 ± 0.07 | 0.82 ± 0.11 |
| | | Crea | - | - | - | - | - |
| | | Typi | 64.29 | 576.22 | 0.24 ± 0.08 | 0.23 ± 0.07 | 0.85 ± 0.14 |
| | DS-Math | ALL | 100.00 | 683.11 | 0.24 ± 0.07 | 0.31 ± 0.12 | 0.82 ± 0.10 |
| | | Hall | 50.00 | 706.22 | 0.26 ± 0.07 | 0.35 ± 0.13 | 0.82 ± 0.06 |
| | | Crea | - | - | - | - | - |
| | | Typi | 50.00 | 660.00 | 0.23 ± 0.07 | 0.27 ± 0.10 | 0.81 ± 0.14 |
| | DS-Inst | ALL | 100.00 | 628.89 | 0.21 ± 0.05 | 0.44 ± 0.09 | 0.87 ± 0.08 |
| | | Hall | 63.88 | 675.75 | 0.22 ± 0.05 | 0.46 ± 0.07 | 0.87 ± 0.07 |
| | | Crea | 7.93 | 615.56 | 0.21 ± 0.05 | 0.44 ± 0.08 | 0.85 ± 0.07 |
| | | Typi | 28.19 | 526.47 | 0.19 ± 0.05 | 0.39 ± 0.10 | 0.87 ± 0.10 |
| | DS-RL | ALL | 100.00 | 509.91 | 0.23 ± 0.05 | 0.47 ± 0.08 | 1.00 ± 0.00 |
| | | Hall | 56.07 | 587.27 | 0.23 ± 0.05 | 0.47 ± 0.07 | 1.00 ± 0.00 |
| | | Crea | 8.24 | 483.12 | 0.22 ± 0.06 | 0.48 ± 0.08 | 1.00 ± 0.00 |
| | | Typi | 35.69 | 394.56 | 0.23 ± 0.06 | 0.47 ± 0.11 | 1.00 ± 0.00 |
| Non-OC | DS-Base | ALL | 100.00 | 425.12 | 0.77 ± 0.26 | 0.52 ± 0.08 | 0.79 ± 0.09 |
| | | Hall | 56.14 | 484.88 | 0.77 ± 0.26 | 0.53 ± 0.07 | 0.80 ± 0.08 |
| | | Crea | 11.52 | 378.72 | 0.83 ± 0.27 | 0.53 ± 0.07 | 0.78 ± 0.10 |
| | | Typi | 32.34 | 337.89 | 0.76 ± 0.25 | 0.52 ± 0.08 | 0.79 ± 0.10 |
| | DS-Math | ALL | 100.00 | 423.91 | 0.68 ± 0.25 | 0.51 ± 0.08 | 0.81 ± 0.10 |
| | | Hall | 56.03 | 483.86 | 0.67 ± 0.25 | 0.51 ± 0.07 | 0.81 ± 0.08 |
| | | Crea | 11.55 | 378.72 | 0.73 ± 0.26 | 0.51 ± 0.08 | 0.80 ± 0.10 |
| | | Typi | 32.42 | 336.40 | 0.68 ± 0.24 | 0.49 ± 0.08 | 0.80 ± 0.11 |
| | DS-Inst | ALL | 100.00 | 393.09 | 0.40 ± 0.14 | 0.52 ± 0.07 | 0.86 ± 0.09 |
| | | Hall | 54.64 | 449.06 | 0.40 ± 0.15 | 0.52 ± 0.07 | 0.87 ± 0.07 |
| | | Crea | 12.00 | 352.56 | 0.42 ± 0.13 | 0.51 ± 0.07 | 0.86 ± 0.09 |
| | | Typi | 33.36 | 315.98 | 0.38 ± 0.13 | 0.51 ± 0.08 | 0.85 ± 0.10 |
| | DS-RL | ALL | 100.00 | 362.50 | 0.39 ± 0.13 | 0.53 ± 0.07 | 1.00 ± 0.00 |
| | | Hall | 55.88 | 407.47 | 0.39 ± 0.14 | 0.53 ± 0.07 | 1.00 ± 0.00 |
| | | Crea | 13.89 | 330.73 | 0.39 ± 0.12 | 0.52 ± 0.07 | 1.00 ± 0.00 |
| | | Typi | 30.24 | 293.99 | 0.39 ± 0.12 | 0.53 ± 0.08 | 1.00 ± 0.00 |

Table 24: Evaluation results of the DS family on responses generated by the DS-RL model ($q = 5\%, \theta = 0.35$).

| Type | Model | Label (%) | | Length | Entropy | HESR | Jaccard |
|---|---|---|---|---|---|---|---|
| OC | DS-Base | ALL | 100.00 | 571.00 | 0.22 ± 0.07 | 0.21 ± 0.06 | 0.82 ± 0.14 |
| | | Hall | 30.00 | 573.00 | 0.21 ± 0.03 | 0.21 ± 0.03 | 0.82 ± 0.14 |
| | | Crea | - | - | - | - | - |
| | | Typi | 70.00 | 570.14 | 0.22 ± 0.08 | 0.21 ± 0.07 | 0.82 ± 0.14 |
| | DS-Math | ALL | 100.00 | 671.25 | 0.21 ± 0.06 | 0.24 ± 0.08 | 0.83 ± 0.12 |
| | | Hall | 41.67 | 705.00 | 0.20 ± 0.02 | 0.26 ± 0.07 | 0.85 ± 0.05 |
| | | Crea | - | - | - | - | - |
| | | Typi | 58.33 | 647.14 | 0.21 ± 0.08 | 0.23 ± 0.08 | 0.83 ± 0.15 |
| | DS-Inst | ALL | 100.00 | 642.64 | 0.19 ± 0.05 | 0.40 ± 0.09 | 0.87 ± 0.08 |
| | | Hall | 55.00 | 711.74 | 0.19 ± 0.04 | 0.43 ± 0.07 | 0.88 ± 0.06 |
| | | Crea | 7.50 | 697.00 | 0.18 ± 0.04 | 0.41 ± 0.09 | 0.85 ± 0.07 |
| | | Typi | 37.50 | 530.42 | 0.18 ± 0.05 | 0.36 ± 0.10 | 0.88 ± 0.10 |
| | DS-RL | ALL | 100.00 | 526.78 | 0.21 ± 0.05 | 0.46 ± 0.09 | 1.00 ± 0.00 |
| | | Hall | 53.81 | 613.53 | 0.21 ± 0.05 | 0.46 ± 0.07 | 1.00 ± 0.00 |
| | | Crea | 8.71 | 489.58 | 0.20 ± 0.04 | 0.47 ± 0.08 | 1.00 ± 0.00 |
| | | Typi | 37.47 | 410.85 | 0.21 ± 0.05 | 0.46 ± 0.11 | 1.00 ± 0.00 |
| Non-OC | DS-Base | ALL | 100.00 | 425.94 | 0.77 ± 0.26 | 0.52 ± 0.08 | 0.79 ± 0.09 |
| | | Hall | 56.13 | 485.83 | 0.77 ± 0.26 | 0.53 ± 0.07 | 0.80 ± 0.08 |
| | | Crea | 11.49 | 378.72 | 0.83 ± 0.27 | 0.53 ± 0.07 | 0.78 ± 0.10 |
| | | Typi | 32.38 | 338.91 | 0.76 ± 0.25 | 0.51 ± 0.08 | 0.79 ± 0.10 |
| | DS-Math | ALL | 100.00 | 424.99 | 0.68 ± 0.25 | 0.51 ± 0.08 | 0.81 ± 0.10 |
| | | Hall | 56.07 | 484.88 | 0.67 ± 0.25 | 0.51 ± 0.07 | 0.81 ± 0.08 |
| | | Crea | 11.51 | 378.72 | 0.73 ± 0.26 | 0.51 ± 0.08 | 0.80 ± 0.10 |
| | | Typi | 32.42 | 337.85 | 0.68 ± 0.24 | 0.49 ± 0.08 | 0.80 ± 0.11 |
| | DS-Inst | ALL | 100.00 | 409.18 | 0.38 ± 0.14 | 0.51 ± 0.07 | 0.86 ± 0.08 |
| | | Hall | 56.04 | 467.98 | 0.38 ± 0.15 | 0.52 ± 0.07 | 0.87 ± 0.07 |
| | | Crea | 11.74 | 362.06 | 0.41 ± 0.13 | 0.51 ± 0.07 | 0.86 ± 0.09 |
| | | Typi | 32.22 | 324.08 | 0.37 ± 0.13 | 0.50 ± 0.08 | 0.85 ± 0.10 |
| | DS-RL | ALL | 100.00 | 386.13 | 0.36 ± 0.13 | 0.52 ± 0.07 | 1.00 ± 0.00 |
| | | Hall | 56.84 | 436.95 | 0.36 ± 0.14 | 0.52 ± 0.07 | 1.00 ± 0.00 |
| | | Crea | 12.52 | 347.27 | 0.38 ± 0.12 | 0.51 ± 0.07 | 1.00 ± 0.00 |
| | | Typi | 30.64 | 307.73 | 0.36 ± 0.12 | 0.52 ± 0.08 | 1.00 ± 0.00 |

Table 25: Evaluation results of the DS family on responses generated by the DS-RL model ($q = 5\%, \theta = 0.40$).

| Type | Model | Label (%) | | Length | Entropy | HESR | Jaccard |
|---|---|---|---|---|---|---|---|
| OC | DS-Base | ALL | 100.00 | 580.88 | 0.21 ± 0.07 | 0.22 ± 0.07 | 0.82 ± 0.16 |
| | | Hall | 37.50 | 573.00 | 0.21 ± 0.03 | 0.21 ± 0.03 | 0.82 ± 0.14 |
| | | Crea | - | - | - | - | - |
| | | Typi | 62.50 | 585.60 | 0.21 ± 0.08 | 0.22 ± 0.08 | 0.82 ± 0.17 |
| | DS-Math | ALL | 100.00 | 580.88 | 0.20 ± 0.07 | 0.22 ± 0.07 | 0.83 ± 0.14 |
| | | Hall | 37.50 | 573.00 | 0.19 ± 0.02 | 0.21 ± 0.04 | 0.86 ± 0.05 |
| | | Crea | - | - | - | - | - |
| | | Typi | 62.50 | 585.60 | 0.20 ± 0.08 | 0.22 ± 0.08 | 0.81 ± 0.17 |
| | DS-Inst | ALL | 100.00 | 655.22 | 0.16 ± 0.04 | 0.36 ± 0.10 | 0.88 ± 0.09 |
| | | Hall | 48.44 | 712.68 | 0.17 ± 0.04 | 0.40 ± 0.08 | 0.88 ± 0.07 |
| | | Crea | 7.81 | 724.00 | 0.16 ± 0.03 | 0.34 ± 0.05 | 0.84 ± 0.07 |
| | | Typi | 43.75 | 579.32 | 0.15 ± 0.04 | 0.32 ± 0.10 | 0.88 ± 0.12 |
| | DS-RL | ALL | 100.00 | 546.82 | 0.18 ± 0.04 | 0.45 ± 0.10 | 1.00 ± 0.00 |
| | | Hall | 48.67 | 655.48 | 0.18 ± 0.04 | 0.44 ± 0.07 | 1.00 ± 0.00 |
| | | Crea | 9.89 | 525.38 | 0.19 ± 0.04 | 0.46 ± 0.10 | 1.00 ± 0.00 |
| | | Typi | 41.44 | 424.33 | 0.19 ± 0.05 | 0.44 ± 0.12 | 1.00 ± 0.00 |
| Non-OC | DS-Base | ALL | 100.00 | 426.08 | 0.77 ± 0.26 | 0.52 ± 0.08 | 0.79 ± 0.09 |
| | | Hall | 56.06 | 485.83 | 0.77 ± 0.26 | 0.53 ± 0.07 | 0.80 ± 0.08 |
| | | Crea | 11.48 | 378.72 | 0.83 ± 0.27 | 0.53 ± 0.07 | 0.78 ± 0.10 |
| | | Typi | 32.47 | 339.66 | 0.76 ± 0.25 | 0.51 ± 0.09 | 0.80 ± 0.10 |
| | DS-Math | ALL | 100.00 | 426.08 | 0.68 ± 0.25 | 0.50 ± 0.08 | 0.81 ± 0.10 |
| | | Hall | 56.06 | 485.83 | 0.67 ± 0.25 | 0.51 ± 0.07 | 0.81 ± 0.08 |
| | | Crea | 11.48 | 378.72 | 0.73 ± 0.26 | 0.51 ± 0.08 | 0.80 ± 0.10 |
| | | Typi | 32.47 | 339.66 | 0.68 ± 0.24 | 0.49 ± 0.08 | 0.80 ± 0.11 |
| | DS-Inst | ALL | 100.00 | 417.25 | 0.38 ± 0.14 | 0.51 ± 0.07 | 0.86 ± 0.08 |
| | | Hall | 56.28 | 477.92 | 0.38 ± 0.15 | 0.52 ± 0.07 | 0.87 ± 0.07 |
| | | Crea | 11.57 | 368.91 | 0.40 ± 0.14 | 0.51 ± 0.07 | 0.86 ± 0.09 |
| | | Typi | 32.15 | 328.45 | 0.37 ± 0.13 | 0.50 ± 0.08 | 0.85 ± 0.10 |
| | DS-RL | ALL | 100.00 | 402.99 | 0.34 ± 0.13 | 0.52 ± 0.07 | 1.00 ± 0.00 |
| | | Hall | 57.41 | 457.56 | 0.34 ± 0.13 | 0.52 ± 0.07 | 1.00 ± 0.00 |
| | | Crea | 11.72 | 354.12 | 0.36 ± 0.12 | 0.51 ± 0.07 | 1.00 ± 0.00 |
| | | Typi | 30.86 | 320.05 | 0.34 ± 0.12 | 0.51 ± 0.08 | 1.00 ± 0.00 |

Table 26: Evaluation results of the DS family on responses generated by the DS-RL model ($q = 3\%, \theta = 0.35$).

| Type | Model | Label (%) | | Length | Entropy | HESR | Jaccard |
|------|-------|-----------|------|--------|---------|------|---------|
| OC | DS-Base | ALL | 100.00 | 617.50 | 0.12 ± 0.03 | 0.23 ± 0.08 | 0.62 ± 0.00 |
| | | Hall | - | - | - | - | - |
| | | Crea | - | - | - | - | - |
| | | Typi | 100.00 | 617.50 | 0.12 ± 0.03 | 0.23 ± 0.08 | 0.62 ± 0.00 |
| | DS-Math | ALL | 100.00 | 733.33 | 0.14 ± 0.04 | 0.22 ± 0.07 | 0.72 ± 0.15 |
| | | Hall | 33.33 | 965.00 | 0.17 ± 0.00 | 0.21 ± 0.00 | 0.93 ± 0.00 |
| | | Crea | - | - | - | - | - |
| | | Typi | 66.67 | 617.50 | 0.12 ± 0.04 | 0.23 ± 0.08 | 0.62 ± 0.04 |
| | DS-Inst | ALL | 100.00 | 653.71 | 0.11 ± 0.03 | 0.29 ± 0.11 | 0.89 ± 0.11 |
| | | Hall | 47.06 | 707.38 | 0.12 ± 0.02 | 0.33 ± 0.11 | 0.85 ± 0.10 |
| | | Crea | - | - | - | - | - |
| | | Typi | 52.94 | 606.00 | 0.11 ± 0.03 | 0.25 ± 0.10 | 0.91 ± 0.10 |
| | DS-RL | ALL | 100.00 | 530.72 | 0.15 ± 0.05 | 0.42 ± 0.12 | 1.00 ± 0.00 |
| | | Hall | 40.28 | 688.45 | 0.14 ± 0.04 | 0.40 ± 0.08 | 1.00 ± 0.00 |
| | | Crea | 4.17 | 340.33 | 0.18 ± 0.04 | 0.43 ± 0.07 | 1.00 ± 0.00 |
| | | Typi | 55.56 | 430.65 | 0.16 ± 0.05 | 0.43 ± 0.14 | 1.00 ± 0.00 |
| Non-OC | DS-Base | ALL | 100.00 | 426.62 | 0.77 ± 0.26 | 0.52 ± 0.08 | 0.79 ± 0.09 |
| | | Hall | 56.03 | 486.12 | 0.76 ± 0.26 | 0.53 ± 0.07 | 0.80 ± 0.08 |
| | | Crea | 11.43 | 378.72 | 0.83 ± 0.27 | 0.53 ± 0.07 | 0.78 ± 0.10 |
| | | Typi | 32.53 | 340.97 | 0.75 ± 0.25 | 0.51 ± 0.09 | 0.80 ± 0.10 |
| | DS-Math | ALL | 100.00 | 426.28 | 0.68 ± 0.25 | 0.50 ± 0.08 | 0.81 ± 0.10 |
| | | Hall | 56.01 | 485.58 | 0.67 ± 0.25 | 0.51 ± 0.07 | 0.81 ± 0.08 |
| | | Crea | 11.44 | 378.72 | 0.73 ± 0.26 | 0.51 ± 0.08 | 0.80 ± 0.10 |
| | | Typi | 32.55 | 340.97 | 0.67 ± 0.24 | 0.49 ± 0.09 | 0.80 ± 0.11 |
| | DS-Inst | ALL | 100.00 | 424.40 | 0.37 ± 0.14 | 0.51 ± 0.08 | 0.86 ± 0.08 |
| | | Hall | 56.06 | 484.11 | 0.37 ± 0.15 | 0.51 ± 0.07 | 0.87 ± 0.07 |
| | | Crea | 11.54 | 378.72 | 0.40 ± 0.14 | 0.50 ± 0.08 | 0.86 ± 0.09 |
| | | Typi | 32.40 | 337.36 | 0.36 ± 0.14 | 0.50 ± 0.09 | 0.85 ± 0.10 |
| | DS-RL | ALL | 100.00 | 421.92 | 0.33 ± 0.13 | 0.51 ± 0.08 | 1.00 ± 0.00 |
| | | Hall | 56.71 | 479.28 | 0.32 ± 0.14 | 0.51 ± 0.07 | 1.00 ± 0.00 |
| | | Crea | 11.76 | 379.37 | 0.34 ± 0.13 | 0.51 ± 0.08 | 1.00 ± 0.00 |
| | | Typi | 31.53 | 334.61 | 0.32 ± 0.12 | 0.51 ± 0.09 | 1.00 ± 0.00 |

Table 27: Evaluation results of the DS family on responses generated by the DS-RL model ($q = 7\%, \theta = 0.35$).

| Type | Model | Label (%) | | Length | Entropy | HESR | Jaccard |
|------|-------|------|------|--------|---------|------|---------|
| OC | DS-Base | ALL | 100.00 | 633.13 | $0.27 \pm 0.07$ | $0.26 \pm 0.08$ | $0.83 \pm 0.11$ |
| | | Hall | 29.17 | 719.14 | $0.26 \pm 0.05$ | $0.28 \pm 0.08$ | $0.80 \pm 0.09$ |
| | | Crea | 4.17 | 773.00 | $0.26 \pm 0.00$ | $0.27 \pm 0.00$ | $0.86 \pm 0.00$ |
| | | Typi | 66.67 | 586.75 | $0.28 \pm 0.07$ | $0.25 \pm 0.07$ | $0.85 \pm 0.12$ |
| | DS-Math | ALL | 100.00 | 639.60 | $0.29 \pm 0.06$ | $0.35 \pm 0.12$ | $0.82 \pm 0.10$ |
| | | Hall | 47.92 | 688.35 | $0.29 \pm 0.06$ | $0.40 \pm 0.10$ | $0.82 \pm 0.06$ |
| | | Crea | 6.25 | 641.33 | $0.27 \pm 0.03$ | $0.35 \pm 0.13$ | $0.81 \pm 0.06$ |
| | | Typi | 45.83 | 588.41 | $0.29 \pm 0.07$ | $0.29 \pm 0.10$ | $0.82 \pm 0.13$ |
| | DS-Inst | ALL | 100.00 | 586.45 | $0.24 \pm 0.05$ | $0.45 \pm 0.08$ | $0.87 \pm 0.07$ |
| | | Hall | 61.84 | 642.04 | $0.24 \pm 0.05$ | $0.47 \pm 0.07$ | $0.88 \pm 0.06$ |
| | | Crea | 7.97 | 545.30 | $0.24 \pm 0.06$ | $0.45 \pm 0.06$ | $0.86 \pm 0.07$ |
| | | Typi | 30.19 | 483.49 | $0.22 \pm 0.05$ | $0.42 \pm 0.10$ | $0.87 \pm 0.09$ |
| | DS-RL | ALL | 100.00 | 494.37 | $0.24 \pm 0.06$ | $0.48 \pm 0.08$ | $1.00 \pm 0.00$ |
| | | Hall | 55.32 | 566.59 | $0.24 \pm 0.05$ | $0.48 \pm 0.07$ | $1.00 \pm 0.00$ |
| | | Crea | 10.43 | 452.57 | $0.25 \pm 0.06$ | $0.49 \pm 0.07$ | $1.00 \pm 0.00$ |
| | | Typi | 34.25 | 390.44 | $0.24 \pm 0.06$ | $0.47 \pm 0.10$ | $1.00 \pm 0.00$ |
| Non-OC | DS-Base | ALL | 100.00 | 423.69 | $0.78 \pm 0.26$ | $0.53 \pm 0.07$ | $0.79 \pm 0.09$ |
| | | Hall | 56.37 | 484.27 | $0.77 \pm 0.26$ | $0.53 \pm 0.07$ | $0.80 \pm 0.08$ |
| | | Crea | 11.53 | 376.53 | $0.84 \pm 0.27$ | $0.53 \pm 0.07$ | $0.78 \pm 0.10$ |
| | | Typi | 32.09 | 334.22 | $0.77 \pm 0.25$ | $0.52 \pm 0.08$ | $0.79 \pm 0.10$ |
| | DS-Math | ALL | 100.00 | 420.21 | $0.69 \pm 0.24$ | $0.51 \pm 0.07$ | $0.81 \pm 0.10$ |
| | | Hall | 56.21 | 480.74 | $0.68 \pm 0.24$ | $0.51 \pm 0.07$ | $0.81 \pm 0.09$ |
| | | Crea | 11.58 | 374.29 | $0.74 \pm 0.25$ | $0.51 \pm 0.07$ | $0.80 \pm 0.10$ |
| | | Typi | 32.21 | 331.09 | $0.69 \pm 0.23$ | $0.50 \pm 0.08$ | $0.80 \pm 0.11$ |
| | DS-Inst | ALL | 100.00 | 370.43 | $0.42 \pm 0.14$ | $0.52 \pm 0.07$ | $0.86 \pm 0.09$ |
| | | Hall | 53.89 | 422.87 | $0.42 \pm 0.15$ | $0.53 \pm 0.07$ | $0.87 \pm 0.07$ |
| | | Crea | 12.64 | 341.57 | $0.43 \pm 0.13$ | $0.52 \pm 0.08$ | $0.85 \pm 0.09$ |
| | | Typi | 33.48 | 296.93 | $0.40 \pm 0.13$ | $0.51 \pm 0.08$ | $0.84 \pm 0.10$ |
| | DS-RL | ALL | 100.00 | 335.61 | $0.42 \pm 0.14$ | $0.54 \pm 0.07$ | $1.00 \pm 0.00$ |
| | | Hall | 56.82 | 380.24 | $0.42 \pm 0.15$ | $0.54 \pm 0.07$ | $1.00 \pm 0.00$ |
| | | Crea | 12.76 | 297.14 | $0.43 \pm 0.12$ | $0.52 \pm 0.08$ | $1.00 \pm 0.00$ |
| | | Typi | 30.42 | 268.38 | $0.42 \pm 0.12$ | $0.54 \pm 0.07$ | $1.00 \pm 0.00$ |

