# OpenReview forum: "Can Reasoning Language Models Think More Creatively? A Study of Reasoning Ability and Overconfidence"
_ICLR.cc/2026/Conference — Submitted to ICLR 2026_

### Official Review · Reviewer_Abau · 2025-10-26

**Soundness:** 3
**Presentation:** 2
**Contribution:** 2
**Rating:** 4
**Confidence:** 3

**Summary:**

In this analysis paper, the authors analyze the creativity of Qwen and Deepseek after different training stages using a model-based evaluation approach. They find that further post-training can harm their creativity. The authors also quantify token entropy and finds that models exhibit lower entropy after post-training.

**Strengths:**

1. The investigation into the diversity of language model outputs is important, as it is closely related to the generalization of language models.
2. Overall, the analysis is comprehensive and multi-faceted.

**Weaknesses:**

1. The fact that entropy can be lower after post-training has been investigated and is a well-known fact in the field [1-3], which makes the analysis less novel.
2. The creativity evaluation is a bit subjective as it is model-based and reference-based. The authors do not clarify why the reference solution is less creative. Also, the lack of qualitative studies makes it hard for the reader to understand.
3. Besides the analysis, guidance on how to help design better language model training methodologies is also lacking
4. The use of the terms "reasoning model" and "math model" is a bit strange, as a reasoning model can also perform well on math tasks. I understand the naming habit of open-source models before o1 was released in 2024/09, but it would be better not to treat them as parallel categories.

References

[1] The Entropy Mechanism of Reinforcement Learning for Reasoning Language Models, in arxiv 2025

[2] One-shot Entropy Minimization, in arxiv 2025

[3] Beyond the 80/20 Rule: High-Entropy Minority Tokens Drive Effective Reinforcement Learning for LLM Reasoning, in arxiv 2025

**Questions:**

See the weaknesses

---

> ### Author Response · Authors · 2025-11-18
> **Response to the Concerns [1/2]**
>
> First, we sincerely appreciate your thoughtful and high-quality review. We truly respect your insightful presentation of cutting-edge research trends and in-depth insights, and we are honored to discuss our research. Below is our response to the concerns raised in the Weakness section.
>
> **A1. On the Novelty of Entropy Reduction After Post-Training**
>
> We agree that the decrease in entropy after post-training has been consistently reported in prior studies. As described in the Introduction, our motivation arises from empirical patterns obscured by this average decrease. In our work, the reduction in average entropy is treated only as a basic assumption grounded in existing studies and used solely as a baseline observation rather than a core contribution.
>
> The actual contributions of this study—distinct from the commonly reported reduction in average entropy—are twofold:
>
> 1. We define Overconfidence (OC) as an independent quantitative metric and provide a numerical comparison and analysis of the excessive certainty exhibited by reasoning models.
> 2. We introduce a new perspective through the High Entropy Segment (HES), which enables an examination of local uncertainty patterns that were previously masked by the decrease in average entropy and thus remained unreported.
>
> Empirically, we observe that although overall entropy within the same model family decreases after post-training, the HES Ratio (HESR) remains constant or very similar. This pattern cannot be explained solely by the reduction in entropy, has not been captured in prior works, and constitutes the first independent empirical finding of its kind.
>
> Therefore, the core message of our study is not to restate the widely known fact that “entropy decreases after post-training,” but rather to shed light on distributional and structural changes in reasoning models that were previously hidden by the reduction in average entropy—using the newly introduced metrics OC and HES/HESR.
>
> We sincerely appreciate the reviewer’s insightful comments informed by a deep understanding of recent research. We will revise the paper to ensure that the independent contributions of our work are clearly conveyed. Thank you again for your valuable feedback.
>
> **A2-1. Creativity Evaluation Is a Bit Subjective**
>
> First, we agree with your comment that creativity evaluation can be somewhat subjective. Since creativity is not a concept that can be cleanly categorized as correct or incorrect, a certain degree of subjectivity is indeed unavoidable. To minimize this limitation, our study strictly follows the formal evaluation criteria proposed in CreativeMath (AAAI 2025) [1], which was established through rigorous peer review.
>
> **A2-2. Why the Reference Solution Is Considered Less Creative**
>
> We instructed the model to generate a novel solution that differs from the reference solution based on the criteria specified in the input prompt. This prompt was also adopted from [1] without any modification. In the CreativeMath dataset, the reference solution is intentionally designed to function as the standard solution, and creativity is evaluated based on how novel a generated solution is compared to this standard. Therefore, treating the reference solution as relatively less creative is not a definition introduced by our study, but rather part of the evaluation framework established in the dataset and prior work. This does not imply that the reference solution itself lacks creativity.
>
> **A2-3. Regarding the Lack of Qualitative Studies**
>
> Qualitative evidence supporting the creativity assessment is already included in the Appendix. The Appendix provides the LLM judge's rationale—both for why a solution is judged creative and why it is not—with examples. The creativity criteria are also described there in detail.
>
> We acknowledge that the main text did not sufficiently guide readers to this qualitative evidence. In the revised version, we will explicitly reference the relevant Appendix sections to ensure that readers can easily access the qualitative explanations.
>
> Furthermore, as noted in Appendix B.2.4, conducting large-scale human evaluations for math competition–level problem solving was infeasible due to practical constraints (time and cost). Thus, we had no option but to rely on LLM-based judgments. (In fact, we reached out to several organizations, including AMT, to conduct human evaluations, but were unable to proceed due to the limited availability of experts and high associated costs.)
>
> Finally, we emphasize that our study does not propose a new creativity evaluation method nor does it perform subjective assessments of its own. Rather, it builds on an existing, rigorously evaluated framework and reports new empirical observations grounded in that established methodology. Thank you.
>
> ---
>
> **References**
>
> [1] Ye, et al. "Assessing the Creativity of LLMs in Proposing Novel Solutions to Mathematical Problems." AAAI 2025.

---

> > ### Author Response · Authors · 2025-11-18
> > **Response to the Concerns [2/2]**
> >
> > **A3. On Providing Guidance for Better Training Methodologies**
> >
> > We fully agree with the points you raised. However, as you know, training language models larger than 7B requires enormous computational resources, and with our current resources, it was difficult to conduct more specific experiments or propose a concrete methodology. Proposing and analyzing techniques such as GRPO is research that only a small number of companies or research institutes can realistically pursue. Therefore, we kindly ask for your understanding regarding our difficulty in presenting a more aggressive direction in this paper. In addition, it was challenging to precisely target the GRPO technique (including detailed parameter configurations) for the Qwen-Inst and DeepSeek-RL models.
> >
> > With these considerations in mind, we would like to cautiously describe the solution guidance we have in mind. We believe that multi-objective RL could help alleviate this issue. Prior studies have already explored reinforcement learning methods with multiple objectives [1, 2]. However, these studies mainly focus on ensemble techniques for handling multi-objective variables, with an emphasis on stability and efficiency. We believe that such multi-objective RL methods could benefit from our findings, especially if the evaluation emphasizes not only accuracy but both accuracy and creativity.
> >
> > Your deep insights have encouraged us to think in new directions. Thank you sincerely.
> >
> > **A4. Clarification on the Use of “Math Model” and “Reasoning Model”**
> >
> > First, we apologize for any confusion caused by the terminology. As you pointed out, the terms “math model” and “reasoning model” may be somewhat inappropriate. We will replace these terms with more appropriate terminology in the revised paper to avoid ambiguity.
> >
> > To clarify, our work does not claim that reasoning models perform poorly on mathematical tasks. Reasoning models indeed demonstrate strong mathematical capabilities, but they struggled specifically when asked to generate novel solutions that differ from existing ones. To investigate the underlying causes, we conducted several experiments and found that (1) the overconfidence phenomenon was significantly higher in reasoning models, and (2) through HESR, these models exhibited a heterogeneous entropy range compared to other models.
> >
> > We sincerely thank you once again for taking the time to provide such thoughtful feedback and for helping us substantially improve our paper.
> >
> > We truly hope that our responses have addressed your concerns, and please feel free to reach out if you have any further questions or points for discussion.
> >
> > ---
> >
> > **References**
> >
> > [1]https://arxiv.org/pdf/2310.03708
> >
> > [2]https://aclanthology.org/2025.sigdial-1.33/

---

> > > ### Comment · Reviewer_Abau · 2025-11-18
> > > **Response to the authors**
> > >
> > > Thanks for your response. Can authors upload a revised manuscript during the discussion period? For example, regarding your statement, 'We will replace these terms with more appropriate terminology in the revised paper to avoid ambiguity.' It would be much clearer for our discussions if you could show the actual revision.

---

> > > > ### Author Response · Authors · 2025-11-18
> > > >
> > > > Thank you for your message.
> > > >
> > > > We are currently preparing a revised version of the manuscript that incorporates your comment as well as feedback from the other reviewers. We will upload the revised version as soon as possible during the discussion period so that you can clearly see the changes, including the updated terminology mentioned earlier.

---

### Official Review · Reviewer_LKxK · 2025-10-27

**Soundness:** 2
**Presentation:** 1
**Contribution:** 2
**Rating:** 2
**Confidence:** 3

**Summary:**

This paper focuses on the trade-off between reasoning ability and creative problem-solving in Large Language Models, comparing two types of models: reasoning models (trained via SFT/RL, including DeepSeek-RL, Qwen-Inst) and math models (only pre-trained on mathematical corpora, including DeepSeek-Math, Qwen-Math). The core findings suggest that reasoning models exhibit higher accuracy in typical mathematical tasks but significantly lower creativity than math models.

**Strengths:**

- The study addresses a critical yet underexplored gap in LLM research—whether the pursuit of advanced reasoning ability (via SFT/RL) compromises creative thinking. This is highly relevant given the widespread application of reasoning models in fields like mathematics and coding, where both accuracy and novel problem-solving are valuable.
- The cross-model family design (DeepSeek + Qwen) enhances the generalizability of findings, while the introduction of HES and OAR adds mechanistic depth beyond simple OC ratio measurements. Comparing entropy, token length, and OC across model types (reasoning vs. math) and evaluation scenarios (self-evaluation vs. external evaluation) provides a comprehensive view of the relationship between model training paradigms and creative output.
- The identification of OC as a barrier to creativity, along with the potential of distillation models (e.g., Qwen-Distill) to balance reasoning and creativity, offers actionable insights for improving LLM design

**Weaknesses:**

- Severe Writing and Table Clarity Issues. e.g.,
1. Inconsistent Model Naming: Section 3.1 clearly defines model aliases, but Table 1 uses inconsistent full names. This forces readers to repeatedly check Section 3.1 to map model types (reasoning vs. math), creating unnecessary confusion.​
2. Unreadable Table Structures: Table 2 is hard to read and we can't have a straightforward comparison between different models in Table 3.
- Insufficient Control and Baseline Design: The inclusion of InternLM2-Math-20B in Table 1 lacks justification (no explanation of its training paradigm in the main text) and creates a baseline mismatch. Unlike the DeepSeek/Qwen models, which have clear "math vs. reasoning" pairs, InternLM2-Math-20B’s classification as a "reasoning model" is unsubstantiated, weakening the paper’s core comparison between model types.​
- Incomplete Explanations: While the paper attributes low creativity to OC, it does not fully address why OC arises during SFT/RL. For example, it mentions "overly aggressive probability adjustments for certain tokens" but provides no analysis of which tokens (e.g., fork tokens vs. common tokens) are affected or how RL loss functions (e.g., GRPO) drive this bias.​

**Questions:**

- Confusion in Section 5.1: Why is InternLM2-Math-20B classified as a reasoning model? What is the purpose of including InternLM2-Math-20B?
- Confusion in Section 5.2: Why is 28.9% considered a "high" OC ratio? How to explain OC ratio drops in non-self-evaluation?
- Confusion in Section 5.3: Why link token length to creativity?
- Priority of Creativity vs. Accuracy: Why is overconfidence criticized if it boosts accuracy?
- Consequences of Overconfidence: What harms does OC cause beyond low creativity?

---

> ### Author Response · Authors · 2025-11-19
> **Response to the Concerns [1/3]**
>
> First, we sincerely appreciate your insightful and constructive comments. Your detailed feedback has been extremely helpful in improving the clarity and precision of our manuscript. Below, we provide point-by-point responses to the issues raised in the Weaknesses and Questions sections.
>
> **A1. Clarification on Model Naming and Baseline Selection (Weakness 1, 2-1 & Question 1)**
>
> We agree with your point regarding the inconsistency in model naming. Although aliases were clearly defined in Section 3.1, we acknowledge that the mixed use of full names in Table 1 could have caused unnecessary confusion when mapping reasoning models and math models. In the revised paper, we will ensure consistency by using only the terminology defined in Section 3.1 across all tables—including Table 1—and throughout the text.
>
> We also believe that the confusion surrounding InternLM2-Math-20B arose from the same issue. We included InternLM2-Math-20B because the initial observation motivating this study was that “creative generation rates do not necessarily increase as model size and accuracy increase.” Table 1 was intended not only to present the baseline performance of the DeepSeek and Qwen models (both 7B) used in our analysis, but also to provide readers with a general performance range for larger models as a point of reference for understanding creative generation capabilities.
>
> However, this intention was not sufficiently explained in the paper, and we recognize that including InternLM2-Math-20B—despite it not being part of the main analysis model set—could mislead readers and weaken the connection between our baselines and analysis. Based on your feedback, we will remove InternLM2-Math-20B from Table 1 in the revised paper to maintain consistency with the analyzed model set.
>
> Thank you for your helpful and constructive feedback.
>
> **A2. Incomplete Explanations Regarding Why OC Arises During SFT/RL (Weakness 2-2)**
>
> First, thank you for pointing out the part of our paper where the explanation was insufficient. As you know, extensively observing the SFT and RL processes of billion-scale language models requires an enormous amount of computational resources, and we do not have the resources to monitor these processes in detail. Therefore, we reasonably relied on the observational results reported in prior studies.
>
> Research teams with sufficient resources have conducted large-scale experiments examining which tokens are primarily being adjusted during post-training. For example, the Qwen team performed extensive analyses to identify which tokens are most frequently adjusted during RL training [1]. There are also relevant studies conducted by Google DeepMind and UC Berkeley [2]. Based on the experimental results reported in these works, we newly defined and measured the concepts of OC and HESR.
>
> There have been very few attempts to measure OC using internal parameter information rather than model responses, and HESR is a completely novel approach introduced in our study. Although we were unable to directly observe token-level adjustments during the SFT/RL process, we would greatly appreciate your understanding that our work builds upon prior research to introduce these new concepts.
>
> **A3. Unreadable Table Structures (Weakness 2)**
>
> As you mentioned, we agree that the readability of Table 2 is suboptimal. This issue arose because we attempted to present a large amount of information within a limited space. In the revised paper, we will reorganize the table structure to improve clarity and readability. Thank you for pointing this out.
>
> ---
>
> **References**
>
> [1] Beyond the 80/20 Rule: High-Entropy Minority Tokens Drive Effective Reinforcement Learning for LLM Reasoning, NeurIPS, 2025
>
> [2] SFT Memorizes, RL Generalizes: A Comparative Study of Foundation Model Post-training, PMLR 267, 2025

---

> ### Author Response · Authors · 2025-11-19
> **Response to the Concerns [2/3]**
>
> **A4-1. Why Is 28.9% Considered “High” OC? (Question 2)**
>
> We did not evaluate OC based on its absolute value, but rather based on a relative comparison within the same model family. As shown in Table 2, when the generation model is DS-RL, the OC values across different evaluation models are:
> - DS-Base: 0.6%
> - DS-Math: 0.7%
> - DS-Inst: 7.4%
> - DS-RL: 28.9%
>
> Within the DeepSeek family, 28.9% is clearly much higher than the other values (0.6%, 0.7%, 7.4%), and therefore we describe it as “high” OC. This study does not compare absolute OC values across different model families (e.g., Qwen vs. DeepSeek). Instead, we analyze the relative magnitude of OC within each family under its own conditions. Thus, the values in the Qwen family and those in the DeepSeek family should be interpreted and analyzed separately.
>
> **A4-2. Why Does OC Decrease in Non-Self-Evaluation? (Question 2)**
>
> The decrease in OC during non-self-evaluation arises from the post-training characteristics of RL-based reasoning models. Reinforcement learning (e.g., RL, GRPO) strengthens specific reasoning trajectories while suppressing others, which narrows the model’s reasoning space and concentrates confidence on a limited set of trajectories. Recent studies [1,2] also report that RL models do not create entirely new reasoning paths; rather, they selectively sharpen a subset of trajectories already present in the base model.
>
> This structural property naturally explains the OC difference observed between self-evaluation and non-self-evaluation.
> OC results for DS-Math generation are as follows:
> - DS-Base: 9.9%
> - DS-Math: 13.2%
> - DS-Inst: 9.3%
> - DS-RL: 8.1%
>
> The Math model is trained only through SFT on a math corpus, so none of its reasoning paths are selectively reinforced. As a result, in self-evaluation, DS-Math evaluates familiar reasoning and shows the highest OC (13.2%), and other models also show similar OC values (9.9%, 9.3%, 8.1%) when evaluating Math-generated reasoning. This indicates that the Math model’s reasoning is not a “selectively reinforced path” for any model, but rather belongs to a general and familiar reasoning space, which naturally leads to small OC differences across models.
>
> For the RL model, this interpretation also holds. RL does not learn entirely new reasoning paths [1]; instead, it selectively reinforces certain trajectories within the same reasoning space that the Math model can generate. Therefore, the reasoning produced by the Math model is not a reinforced trajectory, but it remains a familiar and standard trajectory within the shared reasoning space. Consequently, the RL model also assigns an OC level similar to that of the other models.
>
> In contrast, the reasoning path generated by the RL model is a trajectory selectively strengthened through RL training. Accordingly, in self-evaluation, the RL model exhibits excessive confidence in this path, producing a very high OC (28.9%). However, for other models (Base, Math, Inst), the RL-enhanced trajectory is merely one of many possible reasoning paths and was not reinforced during their training. Therefore, they do not assign high confidence to it, resulting in very low OC in non-self-evaluation.
>
> This phenomenon aligns with patterns commonly reported in recent RL reasoning studies [1,2], where RL models tend to narrow the reasoning space for efficient sampling, leading to confidence being overly concentrated on specific trajectories.
>
> ---
>
> **References**
>
> [1] Does Reinforcement Learning Really Incentivize Reasoning Capacity in LLMs Beyond the Base Model?, NeurIPS 2025
>
> [2] Beyond the 80/20 Rule: High-Entropy Minority Tokens Drive Effective Reinforcement Learning for LLM Reasoning, NeurIPS 2025

---

> > ### Author Response · Authors · 2025-11-19
> > **Response to the Concerns [3/3]**
> >
> > **A5. Clarifying the Relationship Between Token Length and Creativity (Question 3)**
> >
> > We would like to clarify your concern regarding the question, “Why link token length to creativity?” This part of our analysis was never intended to suggest a causal relationship. It simply reports a descriptive observation: different type of models (math model or reasoning model), reexhibit different output length distributions during the generation process.
> >
> > We did not analyze token length as a factor that explains or determines creativity. Rather, we only reported the measured differences in generation lengths between the reasoning model and the math model under identical conditions, including the differences observed across classes. We included this analysis because, while the distributional characteristics of hallucination vs. non-hallucination have been widely studied, the characteristics of creative generations are still not well studied. For this reason, we presented these general observational results without implying any causal interpretation.
> >
> > In our study, the key indicators directly related to creativity are OC and HES/HESR, and the token-length analysis serves only as a supplementary observation that does not influence our conclusions. However, considering your concern that this point may lead to confusion, we will adjust the phrasing in the revised paper to ensure the distinction is clearer and to avoid any misinterpretation of a causal relationship. Thank you.
> >
> > **A6. Addressing Why OC Is Problematic (Question 4, 5)**
> >
> > I would like to address both of your questions: “Why is overconfidence criticized if it increases accuracy?” and “What harms does OC cause beyond the reduction of creativity?”
> >
> > First, the OC examined in our study is not related to accuracy but to creativity (novelty). In our experiment setup, the model is explicitly instructed through the input prompt to “generate a novel solution different from the reference solution.” However, in many cases, the model was confident that it had generated a novel solution, even though the solution it actually generated was not novel.  Thus, regardless of whether the final answer was correct, it failed to follow the intent of the input prompt, yet still expressed high confidence.
> >
> > Beyond creativity loss, recent RL models exhibit OC that leads to broader issues, including degradation in generation diversity and difficulty in detecting hallucinations [1, 2, 3]. Many RL models are known to collapse into extremely low-entropy distributions or to become excessively confident in their own outputs. Research on Expected Calibration Error (ECE) has been actively conducted to address these calibration problems [4, 5, 6]. We discussed these issues in Section 2.1, “Hallucinations from overconfident generations,” and we additionally reported comparative results using existing ECE techniques in Appendix D, “Compared with self-assigned confidence and overconfidence.”
> >
> > You have accurately highlighted the fundamental problems that OC introduces into model behavior. We will emphasize these points more clearly in the revised paper to ensure that the implications of OC are more explicitly conveyed. Thank you.
> >
> > We sincerely thank you once again for taking the time to provide such thoughtful feedback and for helping us substantially improve our paper.
> >
> > We truly hope that our responses have addressed your concerns, and please feel free to reach out if you have any further questions or points for discussion.
> >
> > ---
> >
> > **References**
> >
> > [1] Survey of hallucination in natural language generation. ACM computing surveys, 2023
> >
> > [2] Towards mitigating llm hallucination via self reflection. EMNLP findings, 2023
> >
> > [3] A survey on hallucination in large language models: Principles, taxonomy, challenges, and open questions. ACM Transactions on Information Systems, 2025
> >
> > [4] Calibration in deep learning: A survey of the state-of-the-art,  arXiv:2308.01222, 2023
> >
> > [5] Full-ECE: A Metric For Token-level Calibration on Large Language Models, arXiv:2406.1134, 2025
> >
> > [6] Can We Trust LLMs? Mitigate Overconfidence Bias in LLMs through Knowledge Transfer, CoRR, 2024

---

### Official Review · Reviewer_zFCN · 2025-10-28

**Soundness:** 3
**Presentation:** 3
**Contribution:** 2
**Rating:** 4
**Confidence:** 4

**Summary:**

This paper investigates the internal behavioral patterns of reasoning-oriented large language models (LLMs) by introducing two quantitative metrics: token-level entropy and overconfidence (OC). Through comparative experiments between math-pretrained models (e.g., Qwen-Math, DeepSeek-Math) and reasoning-enhanced models (e.g., Qwen-Inst, DeepSeek-RL), the authors find that reasoning models exhibit lower overall entropy and higher OC, indicating excessive certainty and reduced creative variability. The study further introduces High Entropy Segments (HES) to analyze local uncertainty spikes, suggesting that these segments may correspond to superficial or unstable creative attempts rather than genuine reasoning diversity.

**Strengths:**

- The paper provides well-defined and interpretable quantitative metrics (entropy and OC) to assess model uncertainty and reasoning confidence, addressing a relatively underexplored aspect of LLM interpretability.

- The experimental analysis is systematic and statistically grounded, using proper correlation metrics and consistent evaluations across two model families (DeepSeek and Qwen).

**Weaknesses:**

- While the results are internally coherent, the implications for training methodologies (SFT and RL) are not fully articulated. It remains unclear what practical insight the findings offer for improving reasoning model design beyond observational diagnosis.

- The definition and evaluation of “creativity” rely entirely on LLM-based judges. The paper does not convincingly justify that current LLMs possess the reliability or semantic sensitivity to distinguish truly creative reasoning from surface-level diversity.

-  Model selection is somewhat confusing. The DeepSeek “RL” and Qwen “Inst” variants are presented as reasoning models, but they lack clear long-chain-of-thought (LongCoT) abilities typically associated with advanced reasoning architectures. The experimental lineup therefore might not represent the reasoning paradigm in its full sense.

- There is limited discussion of causality: are high OC and low entropy truly caused by RL fine-tuning, or simply correlated with model style and decoding parameters?

**Questions:**

Minor stylistic issue: line 141 refers to DeepSeek-Coder with a typo.

---

> ### Author Response · Authors · 2025-11-20
> **Response to the Concerns [1/2]**
>
> We sincerely appreciate your careful and constructive evaluation. Your insightful feedback has been instrumental in enhancing the clarity and overall contribution of our manuscript. Below, we provide detailed, point-by-point responses to the concerns raised in the Weaknesses section.
>
> **A1. On the Practical Implications for Improving SFT/RL Training**
>
> We fully agree with the points you raised. However, as you know, training language models larger than 7B requires enormous computational resources, and with our current resources, it was difficult to conduct more detailed experiments or propose a concrete methodology. Proposing and analyzing RL method such as GRPO is realistically feasible only for a small number of companies or research institutes. Therefore, we kindly ask for your understanding regarding our inability to provide a more aggressive methodological direction in this paper.
>
> With these limitations in mind, we would like to cautiously describe the solution guidance we have in mind. We believe that multi-objective RL could help mitigate this issue. Prior studies have already explored RL with multiple objectives [1, 2]. However, these works primarily focus on ensemble techniques for handling multi-objective variables, emphasizing stability and efficiency. We believe that such multi-objective RL approaches could benefit from our findings if future RL methods are proposed and evaluated with attention not only to accuracy but to both accuracy and creativity.
>
> Your insightful comments encouraged us to think in new directions. Thank you sincerely.
>
> **A2. Creativity Definition and Evaluation Reliant on LLM Judgment**
>
> As you pointed out, we also acknowledge the limitations of relying on LLMs for creativity evaluation. However, we would like to emphasize that the definition of creativity used in our study is not determined arbitrarily by the LLM. Instead, it strictly follows the official criteria proposed in CreativeMath (AAAI 2025) [3].
>
> The CreativeMath framework does not judge creativity based on superficial differences in wording or expression. Rather, it defines creativity through structural and substantive differences in reasoning, including changes in solution structure, logical strategies, mathematical tools used, and the scope of applicability. A solution is considered creative if it satisfies at least one of the following:
> - The overall approach is fundamentally different.
> - The intermediate reasoning steps differ in a meaningful way.
> - Different conditions, theorems, or assumptions are employed.
> - The applicable problem domain changes.
> - The solution demonstrates a strategic difference, such as being significantly simpler or more complex.
>
> These criteria are designed to evaluate deep, structural differences in reasoning—not surface-level variation. Our prompts directly incorporate these criteria, meaning that the LLM judge is not making arbitrary decisions but is evaluating according to clearly defined creativity requirements.
>
> We fully agree that human evaluation is valuable. However, applying these criteria requires far more than verifying correctness; it demands substantial mathematical expertise capable of analyzing techniques, theorems, and logical structures used in competition-level problem solving. As noted in Appendix B.2.4, this makes large-scale human annotation practically infeasible.
>
> Regarding whether the LLM judge can meaningfully distinguish “genuinely creative reasoning,” Appendix B.2.3 provides qualitative examples of the rationales behind the LLM’s decisions. These examples show that the LLM bases its judgments on structural differences—such as theorems used, logical strategies, and new mathematical perspectives—rather than superficial stylistic differences.
>
> Finally, we emphasize that our study does not propose a new creativity evaluation method. Rather, it analyzes the distributional and structural changes of LLMs within an established, peer-reviewed evaluation framework.
>
> ---
>
> **References**
>
> [1] Beyond One-Preference-Fits-All Alignment: Multi-Objective Direct Preference Optimization, ACL, 2024
>
> [2] EMORL: Ensemble Multi-Objective Reinforcement Learning for Efficient and Flexible LLM Fine-Tuning, SIGDIAL, 2025
>
> [3] Assessing the Creativity of LLMs in Proposing Novel Solutions to Mathematical Problems, AAAI, 2025

---

> > ### Author Response · Authors · 2025-11-20
> > **Response to the Concerns [2/2]**
> >
> > **A3. Clarifying the Terminology and Model Categorization**
> >
> > We apologize for the confusion caused by our use of the terms “math model” and “reasoning model,” which overlapped with terminology used in prior works and may have made it difficult for reviewers to follow our categorization. In the revised paper, we will replace these terms with alternative terminology accompanied by clear definitions. For example, as suggested in prior work [1], we plan to use the term “RLVR model” in place of “reasoning model.”
> >
> > We sincerely appreciate your careful review and will ensure that these changes are reflected in the revised paper. Thank you once again for your constructive feedback.
> >
> > **A4. Clarifying the Causal Interpretation of OC and Entropy Changes in RL**
> >
> > Another reviewer raised similar concerns, and we acknowledge that our paper did not sufficiently cite or explain the relevant prior research. We sincerely appreciate your meticulous review. However, recent prior studies have already provided extensive empirical evidence showing that RL leads to simplified reasoning paths and lower entropy. Therefore, our explanations were based in these prior observations, and we cited them accordingly.
> >
> > Prior works [1, 2, 3] conducted large-scale experiments examining how token entropy changes during RL. We relied on the experimental findings from these studies as the basis for our hypotheses, but our explanation and emphasis on this connection were not sufficiently clear. We will make sure to clarify these citations in the revised paper. These studies reported the following findings, which served as the theoretical foundation for our work:
> > 1. Studying how entropy patterns evolve during RLVR training reveals that RLVR largely adheres to the base model’s entropy patterns, primarily adjusting the entropy of high-entropy tokens. [1]
> > 2. SFT stabilizes the model’s output format, enabling subsequent RL to achieve its performance gains. [2]
> > 3. The distribution of Perplexity of RL model closely matches the lower portion of the Perplexity of base model distribution, corresponding to responses that the base model tends to generate. [3]
> > 4. After RLVR training, the model often exhibits narrower reasoning coverage compared to its base model. [3]
> >
> > Building on these findings, we newly defined and measured the concepts of OC and HESR. Few prior works have attempted to measure OC using internal parameter information rather than output-based metrics, and HESR represents a completely novel approach introduced in our study. Although we were unable to directly observe token-level adjustments throughout the SFT/RL process, we would be sincerely grateful if this point—namely, that our work introduces new conceptual measures built upon prior research—could be taken into consideration.
> >
> > **Additional Comment**
> >
> > Thank you for pointing out the minor typographical issue regarding “DeepSeek-Coder” in line 141. We have corrected this typo in the revised version. Thank you again for your careful attention to detail.
> >
> > We sincerely thank you once again for taking the time to provide such thoughtful feedback and for helping us substantially improve our paper.
> >
> > We truly hope that our responses have addressed your concerns, and please feel free to reach out if you have any further questions or points for discussion.
> >
> > ---
> >
> > **References**
> >
> > [1] Beyond the 80/20 Rule: High-Entropy Minority Tokens Drive Effective Reinforcement Learning for LLM Reasoning, NeurIPS, 2025
> >
> > [2] SFT Memorizes, RL Generalizes: A Comparative Study of Foundation Model Post-training, ICML, 2025
> >
> > [3] Does Reinforcement Learning Really Incentivize Reasoning Capacity in LLMs Beyond the Base Model?, NeurIPS, 2025

---

> > > ### Comment · Reviewer_zFCN · 2025-11-28
> > >
> > > Thank you very much for the authors’ careful and detailed response. I fully understand the practical limitations caused by insufficient computational resources, especially when conducting large-scale post-training experiments on LM models, and I appreciate your candid explanation.
> > >
> > > That said, I still feel that the overall insights provided by this work remain somewhat limited. More importantly, the issue of “creativity” in large language models is not only about whether an LLM judge is reliable or well-designed. In practice, what counts as a “creative” solution is itself highly subjective, even among experienced human experts or mathematics educators with strong domain knowledge. This inherent subjectivity makes it difficult for “creativity” to serve as a solid central pillar for analyzing and drawing conclusions about reasoning models.
> > >
> > > Moreover, regarding the discussion on RL and entropy, while your explanations are clearer now and better grounded in prior work, I still feel that the current manuscript does not provide sufficient theoretical novelty beyond confirming and reorganizing existing empirical observations.
> > >
> > > For these reasons, I will maintain my original score. I hope you can understand my position, and I sincerely appreciate the efforts you made to improve the clarity and framing of your work.

---

> > > > ### Author Response · Authors · 2025-11-28
> > > >
> > > > Thank you once again for reviewing our paper and sharing your thoughts. We sincerely appreciate the detailed feedback and constructive suggestions you have provided throughout this process, and we respect the perspective you have shared.
> > > >
> > > > We would have welcomed the opportunity to address any further questions or continue the discussion; however, due to the current situation, this is unfortunately no longer possible, which we find quite regrettable. Nonetheless, we are truly grateful for the valuable comments you provided despite your busy schedule, as they have meaningfully contributed to improving the clarity and direction of our work.
> > > >
> > > > Thank you.

---

### Official Review · Reviewer_hhFo · 2025-11-01

**Soundness:** 2
**Presentation:** 2
**Contribution:** 3
**Rating:** 4
**Confidence:** 4

**Summary:**

This paper focus on the diversity loss or homogeneity of reasoning models compared to pre-trained models in terms of math problem solving.
The author introduced a concept of **High Entropy Segment** to measure the ratio of the mean TE (token-level entropy) of a window compared to  the mean of the entire generation.  The take-away is that, although reasoning models often exhibited (over-)confidence in
the creativity of their generations, the actual proportion of creative solutions was relatively low.

**Strengths:**

1. The theme to investigate diversity loss or homogeneity of reasoning models is imo very pertinent and deserves more attention to the LLM community.
2. The author introduced a concept of **High Entropy Segment** to measure the ratio of the mean TE (token-level entropy) of a window compared to  the mean of the entire generation. I can envision future works report the time-sensitive heterogeneity of their model using this metric

**Weaknesses:**

Two major weaknesses:
1. The overall observation is that reasoning models is less creative in math problem solving than a base model pre-trained on math corpus.
A very important hypothesis of the paper is that this is because of **overly aggressive probability adjustments for certain tokens -- referred to as fork tokens**. The authors introduced High Entropy Segment (HES) only to corroborate such observation, yet, no quantitative or qualitative analysis is conducted on these certain tokens to showcase or support the hypothesis.
The claim *This is largely because their distributions contain a substantially greater share of tokens whose probabilities exceed 80% at each step.* needs more support either by quantitative or qualitative analysis.
I am afraid the take-away message is `reasoning models tend to mysteriously exhibit greater heterogeneity` with an `unverified hypothesis`. This make the contribution less convincing to me.

2. Paper presentation is largely unclear. I found the following confusing:
* Experimental models mentioned in Section 3.1 does match the results shown in Table 1, Section 5. No results for qwen-distill. Where does InternLM2-Math-20B come from

* The results of jaccard, HESR are presented in Table 3 without definition of these terms. (later they were explained in section 6 but the authors need to provide definitions the first time these terms appear. otherwise it is surprising and confusing to the readers when the first time encountering these terms are in the results table.

* The paper seem to not emphasize its effort in LLM as a judge, but a decent proportion of the results, analysis, and discussions are pertinent to the variance of math&reasoning models as a judge.

* Same as previous point, def of math model is presented in Figure 1, but abstract and intro did not provide a crystal clear definition

**Questions:**

N/A

---

> ### Author Response · Authors · 2025-11-21
> **Response to the Concerns [1/2]**
>
> First, we sincerely appreciate your thoughtful and high-quality review. We truly respect your insightful presentation of cutting-edge research trends and in-depth insights, and we are honored to discuss our research. Below is our response to the concerns raised in the Weakness section.
>
> **A1. Providing Quantitative and Qualitative Evidence**
>
> Thank you sincerely for the sharp and thoughtful comments. As you pointed out, both “overly aggressive probability adjustments for certain tokens—referred to as fork tokens” and “This is largely because their distributions contain a substantially greater share of tokens whose probabilities exceed 80% at each step” require explicit supporting evidence. The supporting evidence is as follows.
>
> *“Theoretical Basis”*
>
> Previous studies [1, 2, 3] conducted extensive experiments examining how token entropy changes during the RL process. We relied on the experimental findings from these works as the justification for our hypotheses, but our explanation and emphasis on this point were insufficient. We will make sure to clarify these citations more explicitly in the revised paper.  These prior works revealed the following facts, and we used them as the theoretical basis for our study:
> 1. Studying how entropy patterns evolve during RLVR training reveals that RLVR largely adheres to the base model’s entropy patterns, primarily adjusting the entropy of high-entropy tokens. [1]
> 2. SFT stabilizes the model’s output format, enabling subsequent RL to achieve its performance gains. [2]
> 3. The distribution of Perplexity of RL model closely matches the lower portion of the Perplexity of base model distribution, corresponding to responses that the base model tends to generate. [3]
> 4. After RLVR training, the model often exhibits narrower reasoning coverage compared to its base model. [3]
>
> Based on these findings, we formulated reasonable hypotheses grounded in prior research and analyzed our experiments accordingly. We acknowledge that these prior research results were not sufficiently emphasized, and we will reflect this more clearly in the revised paper.
>
> *"Quantitative Analysis"*
>
> Although we do not have the resource to perform SFT or RL on Billion-scale models and therefore cannot present changes during training, we performed a quantitative analysis of the generated outputs based on the finding from [1] that "the top 20% highest-entropy tokens undergo strong adjustments, while the remaining 80% experience little change." The following results show, for each model, whether the top 20% high-entropy tokens in its generations remain high-entropy when evaluated under other models.
>
> | Generation Model / Split | Inc (%) | Dec (%) | Norm Δ Avg | Raw Δ Avg |
> |---|:---:|:---:|:---:|:---:|
> | Deepseek-Math (Top 20%) | 37.69 | 62.31 | -0.04 | -0.79 |
> | Deepseek-Math (Bottom 80%) | 19.84 | 80.12 | -0.00 | -0.09 |
> | Qwen-Math (Top 20%) | 26.52 | 73.48 | -0.09 | 1.04 |
> | Qwen-Math (Bottom 80%) | 68.41 | 31.58 | 0.02 | 0.43 |
>
> In the Math model generations, the normalized values of the top 20% entropy tokens showed far more decreases when evaluated by the RL model. Specifically, 62.31% of tokens generated by the DeepSeek-Math model and 73.48% of tokens generated by the Qwen-Math model exhibited reduced normalized entropy. Furthermore, the bottom 80% of tokens recorded much smaller values in Raw ΔAvg compared to the top 20% tokens, showing a relatively limited degree of change—strongly consistent with the findings of prior work [1].
>
> Additionally, Figure 10 in Appendix E visualizes the changes in normalized entropy for the top 20% and bottom 80% of tokens. As with the quantitative results above, the top 20% tokens show substantially larger fluctuations, whereas the bottom 80% tokens exhibit almost no change.
>
> *"Qualitative Analysis"*
>
> We present qualitative observations of HES in Figure 2 of the main text, through which one can observe examples of how the reasoning model adjusts token entropy. Figure 11 in Appendix E illustrates how the entropy of the math model's generation differs when measured by the math model itself or by the RL model. For the same generated response, the RL models showed much lower entropy—indicating higher confidence—on most tokens compared to the math model. This observation is highly consistent with previous research [1, 2, 3] and our findings.
>
> These results could serve as key evidence supporting our takeaway message: 'reasoning models tend to mysteriously exhibit greater heterogeneity.' We sincerely appreciate your insigh into this critical aspect of our work and thank you for highlighting it.
>
> ---
>
> **References**
>
> [1] Beyond the 80/20 Rule: High-Entropy Minority Tokens Drive Effective Reinforcement Learning for LLM Reasoning, NeurIPS, 2025
>
> [2] SFT Memorizes, RL Generalizes: A Comparative Study of Foundation Model Post-training, ICML, 2025
>
> [3] Does Reinforcement Learning Really Incentivize Reasoning Capacity in LLMs Beyond the Base Model?, NeurIPS, 2025

---

> > ### Author Response · Authors · 2025-11-21
> > **Response to the Concerns [2/2]**
> >
> > **A2-1. Clarifying Model Naming and the Inclusion of InternLM2-Math-20B**
> >
> > We sincerely apologize for the confusion. We agree with the reviewer's comment regarding the inconsistency in model naming. Although alias were clearly defined in Section 3.1, mixing full names in Table 1 may have caused unnecessary confusion for readers when mapping the reasoning models to the math models. In the revised paper, we will ensure that Table1 as well as other tables and the main text consistently follow the naming conventions defined in Section 3.1.
> >
> > We also believe that the confusion surrounding InternLM2-Math-20B arose from the same issue. The reason we initially included InternLM2-Math-20B is that this study was motivated by the observation that "even as model size and accuracy increase, the rate of creative generation does not necessarily improve.” Table 1 was designed not only to present the baseline performance of the DeepSeek and Qwen models (both 7B) used in our analysis, but also to provide readers with a reference point for the general performance range of larger models.
> >
> > However, this intention was not sufficiently explained in the paper, and, as the reviewer correctly pointed out, including InternLM2-Math-20B in Table 1 despite it not being part of the main analysis model set could lead to unnecessary confusion and weaken the clarity of our baseline presentation. Accordingly, in the revised paper, we will remove InternLM2-Math-20B from Table 1 to maintain consistency with the set of models analyzed. In addition, we are currently evaluating the performance of Qwen-Distill. Because the generation and labeling processes require substantial time, we will update the main paper with the results once the evaluation is complete.
> >
> > **A2-2. Clarifying Early Definition for HESR**
> >
> > We appreciate your accurate comment, and we apologize for any inconvenience caused. We will move the HESR definition from Section 6 to Section 4 and reflect this change in the revised paper.
> >
> > **A2-3. Clarifying Our Use of Math and Reasoning Models as Evaluators**
> >
> > Thank you for the insightful comment. As you noted, both the math model and the reasoning model serve crucial roles as evaluators in our study. Through this experimental design, we aim to clarify the following:
> > 1. Based on prior research suggesting that certain reasoning paths are selectively reinforced during the RL process, we prompted the reasoning model to generate creative solutions; however, the results were not even better than those of the math model.
> > 2. To understand the cause, we observed the heterogeneity in the generation of each model across models trained using the same base model.
> > 3. To this end, we selected a math model and a reasoning model built on the same base model and measured various experimental results.
> > 4. We proposed the novel metrics OC and HES to identify this phenomenon, and demonstrated that the reasoning model shows higher OC and HE values than the math model.
> >
> > We acknowledge that our explanation of these efforts was insufficient, and we have included quantitative and qualitative analyses of these differences in Appendix E and uploaded a revised paper. We sincerely appreciate your thorough examination of the shortcomings in our paper.
> >
> > **A2-4. Clarifying and Revising Model Terminology**
> > We noted that multiple several reviewers requested clearer definitions and refined terminology for the "math model" and the "reasoning model". We appreciate these insightful comments, and we will revise the definitions and terminology accordingly in the updated version of the paper.
> >
> > We sincerely thank you once again for taking the time to provide such thoughtful feedback and for helping us substantially improve our paper.
> >
> > We truly hope that our responses have addressed your concerns, and please feel free to reach out if you have any further questions or points for discussion.

---

### Author Response · Authors · 2025-11-21
**Summary of Revisions in the Updated Manuscript**

First, we would like to express our sincere gratitude to all reviewers and the AC for their hard work in reviewing our paper. We have uploaded the revised version, which incorporates all reviewer feedback. We understand that you have busy schedules, but we kindly ask that you review the updated manuscript.

Here is a summary of the revisions:
1. We added quantitative and qualitative analyses of token probability adjustments in Figures 10 and 11 and Table 7 in Appendix E.
2. We no longer use the previously introduced terms “math model” and “reasoning model” in the Abstract or Introduction. Beginning in Section 3, we instead define the terms “math model” and “RL model” clearly and use them throughout the rest of the paper to prevent confusion.
3. The experimental results for InternLM in Table 1 were removed, as they were unnecessary.
4. Table 2 has been revised to improve readability.
5. The definition of HESR has been moved from Section 6 to Section 4 so that it appears before the abbreviation is used in the text.
6. We clearly articulated the basis for our research assumptions by accurately citing findings from prior studies, such as token-probability adjustments during the SFT/RL process. We also clarified our novel contributions and the specific aspects on which our analysis focuses.

We were able to improve the presentation and clarity of our paper thanks to the many thoughtful comments from the reviewers. We sincerely appreciate everyone who provided us with this valuable opportunity to strengthen our work, and we are fully prepared for any further discussion.

Thank you sincerely.

Authors

---

### Meta-Review · Area_Chair_Lf5U · 2026-01-04

**Summary:**

While the reviewers acknowledged the submission's strengths, particularly the introduction of HES and the insightful analysis, they raised significant concerns regarding the study's novelty, methodological contribution, and the subjectivity of the evaluations. Although the authors made a commendable effort to address these issues during the rebuttal, the necessary revisions are unfortunately too extensive to be adequately reviewed within the constraints of the current ICLR cycle.

**Reviewer Concerns:**

1. The novelty of the proposed notion HES and the findings.

2. Lack of methodological contribution.

3. Subjectivity of the evaluations.

**Reviewer Scores:**

4/4/2/6

---

### Decision · Program_Chairs · 2026-01-26

Reject